# On Penalty Methods for Nonconvex Bilevel Optimization and First-Order Stochastic Approximation

## Abstract

In this work, we study first-order algorithms for solving Bilevel Optimization (BO) where the objective functions are smooth but possibly nonconvex in both levels and the variables are restricted to closed convex sets. As a first step, we study the landscape of BO through the lens of penalty methods, in which the upper- and lower-level objectives are combined in a weighted sum with penalty parameter $\sigma > 0$. In particular, we establish a strong connection between the penalty function and the hyper-objective by explicitly characterizing the conditions under which the values and derivatives of the two must be $O(\sigma)$-close. A by-product of our analysis is the explicit formula for the gradient of hyper-objective when the lower-level problem has multiple solutions under mild regularity conditions, which could be of independent interest. Next, viewing the penalty formulation as $O(\sigma)$-approximation of the original BO, we propose first-order algorithms that find an $\epsilon$-stationary solution by optimizing the penalty formulation with $\sigma = O(\epsilon)$. When the perturbed lower-level problem uniformly satisfies the *small-error* proximal error-bound (EB) condition, we propose a first-order algorithm that converges to an $\epsilon$-stationary point of the penalty function using in total $O(\epsilon^{-7})$ accesses to first-order stochastic gradient oracles. Under an additional assumption on stochastic oracles, we show that the algorithm can be implemented in a fully *single-loop* manner, *i.e.,* with $O(1)$ samples per iteration, and achieves the improved oracle-complexity of $O(\epsilon^{-5})$.

## 1 Introduction

Bilevel Optimization (BO) Dempe (2003); Colson et al. (2007); Dempe et al. (2015) is a versatile framework for optimization problems in many applications arising in economics, transportation, operations research, and machine learning, among others Sinha et al. (2017). In this work, we consider the following formulation of BO:

$$\min_{x \in \mathcal{X}, y^* \in \mathcal{Y}} \quad f(x, y^*) := \mathbb{E}_\zeta[f(x, y^*; \zeta)]$$
$$\text{s.t.} \quad y^* \in \arg\min_{y \in \mathcal{Y}} g(x, y) := \mathbb{E}_\xi[g(x, y; \xi)], \tag{P}$$

where $f$ and $g$ are continuously differentiable and smooth functions, $\mathcal{X} \subseteq \mathbb{R}^{d_x}$ and $\mathcal{Y} \subseteq \mathbb{R}^{d_y}$ are closed convex sets, and $\zeta$ and $\xi$ are random variables (*e.g.,* indexes of batched samples in empirical risk minimization). That is, (P) minimizes $f$ over $x \in \mathcal{X}$ and $y \in \mathcal{Y}$ (the upper-level problem) when $y$ must be one of the minimizers of $g(x, \cdot)$ over $y \in \mathcal{Y}$ (the lower-level problem).

Scalable optimization methods for solving (P) are in high demand to handle increasingly large-scale applications in machine-learning, including meta-learning Rajeswaran et al. (2019), hyper-parameter optimization Franceschi et al. (2018); Bao et al. (2021), model selection Kunapuli et al. (2008); Giovannelli et al. (2021), adversarial networks Goodfellow et al. (2020); Gidel et al. (2018), game theory Stackelberg et al. (1952), and reinforcement learning Konda & Tsitsiklis (1999); Sutton & Barto (2018). There is particular interest in developing (stochastic) gradient-descent-based methods due to their simplicity and scalability to large-scale problems Ghadimi & Wang (2018); Chen et al. (2021); Hong et al. (2023); Khanduri et al. (2021); Chen et al. (2022); Dagréou et al. (2022); Guo et al. (2021); Sow et al. (2022); Ji et al. (2021); Yang et al. (2021). One popular approach is to perform a direct gradient-descent on the hyper-objective $\psi(x)$ defined as follows:

$$\psi(x) := \min_{y \in \mathcal{S}(x)} f(x, y), \quad \text{where } \mathcal{S}(x) = \arg\min_{y \in \mathcal{Y}} g(x, y), \tag{1}$$

but this requires the estimation of $\nabla\psi(x)$, which we refer to as an *implicit* gradient. Existing works obtain this gradient under the assumptions that $g(x, y)$ is strongly convex in $y$ and the lower-level problem is unconstrained, *i.e.,* $\mathcal{Y} = \mathbb{R}^{d_y}$. There is no straightforward extension of these approaches to nonconvex and/or constrained lower-level problems (*i.e.,* $\mathcal{Y} \subset \mathbb{R}^{d_y}$ is defined by some constraints) due to the difficulty in estimating implicit gradients; see Section A.1 for more discussion in detail.

The goal of this paper is to extend our knowledge of solving BO with unconstrained strongly-convex lower-level problems to a broader class of BO with possibly constrained and nonconvex lower-level problems. In general, however, when there is not enough curvature around the lower-level solution, the problem can be highly ill-conditioned and no known algorithms can handle it even for the simpler case of min-max optimization Chen et al. (2023b); Jin et al. (2020) (see also Example 1). In such cases, it could be fundamentally hard even to identify *tractable* algorithms Daskalakis et al. (2021).

To circumvent the fundamental hardness, there have been many recent works on BO with nonconvex lower-level problems (nonconvex inner-level maximization in min-max optimization literature) under the uniform Polyak-Łojasiewicz (PL)-condition Karimi et al. (2016); Yang et al. (2020; 2022); Xiao et al. (2023a); Huang (2023); Li et al. (2022a). While this assumption does not cover all interesting cases of BO, it can cover situations in which the lower-level problem is nonconvex and where it does not have a unique solution. It can thus be viewed as a significant generalization of the strong convexity condition. Furthermore, several recent results show that the uniform PL condition can be satisfied by practical and complicated functions such as over-parameterized neural networks Frei & Gu (2021); Song et al. (2021); Liu et al. (2022).

Nevertheless, to our best knowledge, no algorithm is known to reach the stationary point of $\psi(x)$ (*i.e.,* to find $x$ where $\nabla\psi(x) = 0$) under PL conditions alone. In fact, even the landscape of $\psi(x)$ has not been studied precisely when the lower-level problem can have multiple solutions and constraints. We take a step forward in this direction under the proximal error-bound (EB) condition that is analogous to PL but more suited for constrained problems[1].

## 1.1 Overview of Main Results

Since it is difficult to work directly with implicit gradients when the lower-level problem is nonconvex, we consider a common alternative that converts the BO (**P**) into an equivalent constrained single-level problem, namely,

$$\min_{x \in \mathcal{X}, y \in \mathcal{Y}} f(x, y) \qquad \text{s.t.} \quad g(x, y) \leq \min_{z \in \mathcal{Y}} g(x, z), \qquad (\mathbf{P}_{\text{con}})$$

and finds a stationary solution of this formulation, also known as an (approximate)-KKT solution Lu & Mei (2023); Liu et al. (2023); Ye et al. (2022). ($\mathbf{P}_{\text{con}}$) suggests the penalty formulation

$$\min_{x \in \mathcal{X}, y \in \mathcal{Y}} \sigma f(x, y) + (g(x, y) - \min_{z \in \mathcal{Y}} g(x, z)), \qquad (\mathbf{P}_{\text{pen}})$$

with some sufficiently small $\sigma > 0$. Our fundamental goal in this paper is to describe algorithms for finding approximate stationary solutions of ($\mathbf{P}_{\text{pen}}$), as explored in several previous works White & Anandalingam (1993); Ye et al. (1997); Shen & Chen (2023); Kwon et al. (2023).

In pursuing our goal, there are two important questions to be addressed. The first one is a landscape question: since ($\mathbf{P}_{\text{pen}}$) is merely an approximation of (**P**), it is important to understand the relationship between their respective landscapes, which have remained elusive in the literature Chen et al. (2023b). The second one is an algorithmic question: solving ($\mathbf{P}_{\text{pen}}$) still requires care since it involves a nested optimization structure (solving an inner minimization problem over $z$), and typically with a very small value of the penalty parameter $\sigma > 0$.

**Landscape Analysis.** Our first goal is to bridge the gap between landscapes of the two problems (**P**) and ($\mathbf{P}_{\text{pen}}$). By scaling ($\mathbf{P}_{\text{pen}}$), we define the penalized hyper-objective $\psi_\sigma(x)$:

$$\psi_\sigma(x) := \frac{1}{\sigma} \left( \min_{y \in \mathcal{Y}} (\sigma f(x, y) + g(x, y)) - \min_{z \in \mathcal{Y}} g(x, z) \right). \qquad (2)$$

As mentioned earlier, without any assumptions on the lower-level problem, it is not possible to make any meaningful connections between $\psi_\sigma(x)$ and $\psi(x)$, since the original landscape $\psi(x)$ itself may

---

[1] In the unconstrained setting, EB and PL are known to be equivalent conditions, see *e.g.,* Karimi et al. (2016)

not be well-defined. Proximal operators are key to defining assumptions that guarantee nice behavior of the lower-level problem.

**Definition 1** *The proximal operator with parameter $\rho$ and function $f(\theta)$ over $\Theta$ is defined as*

$$\mathbf{prox}_{\rho f}(\theta) := \arg\min_{z \in \Theta} \left\{ \rho f(z) + \tfrac{1}{2}\|z - \theta\|^2 \right\}. \tag{3}$$

We now state the proximal-EB assumption, which is crucial to our approach in this paper.

**Assumption 1 (Proximal-EB)** *Let* $h_\sigma(x, \cdot) := \sigma f(x, \cdot) + g(x, \cdot)$ *and* $T(x, \sigma) := \arg\min_{y \in \mathcal{Y}} h_\sigma(x, y)$. *We assume that for all* $x \in \mathcal{X}$ *and* $\sigma \in [0, \sigma_0]$, $h_\sigma(x, \cdot)$ *satisfies the* $(\mu, \delta)$-*proximal error bound:*

$$\rho^{-1} \cdot \left\| y - \mathbf{prox}_{\rho h_\sigma(x, \cdot)}(y) \right\| \geq \mu \cdot \mathbf{dist}(y, T(x, \sigma)), \tag{4}$$

*for all* $y \in \mathcal{Y}$ *that satisfies* $\rho^{-1} \cdot \|y - \mathbf{prox}_{\rho h_\sigma(x, \cdot)}(y)\| \leq \delta$ *with some positive parameters* $\mu, \delta, \sigma_0 > 0$.

As we discuss in detail in Section 3, the crux of Assumption 1 is the guaranteed (Lipschitz) continuity of solution sets, under which we prove our key landscape analysis result:

**Theorem 1.1 (Informal)** *Under Assumption 1 (with additional smoothness assumptions), for all* $x \in \mathcal{X}$ *such that at least one sufficiently regular solution path* $y^*(x, \sigma)$ *exists for* $\sigma \in [0, \sigma_0]$, *we have*

$$|\psi_\sigma(x) - \psi(x)| = O(\sigma/\mu),$$
$$\|\nabla\psi_\sigma(x) - \nabla\psi(x)\| = O(\sigma/\mu^3).$$

As a corollary, our result implies global $O(\sigma)$-approximability of $\psi_\sigma(x)$ for the special case studied in several previous works (*e.g.,* Ghadimi & Wang (2018); Chen et al. (2021); Ye et al. (2022); Kwon et al. (2023)), where $g(x, \cdot)$ is (locally) strongly-convex and the lower-level problem is unconstrained. To our best knowledge, such a connection, and even the differentiability of $\psi(x)$, is not fully understood for BO with nonconvex lower-level problems with possibly multiple solutions and constraints. In particular, the case with possibly multiple solutions (Theorem 3.1) is discussed under the mild assumptions of solution-set continuity and additional regularities of lower-level constraints.

**Algorithm.** Once we show that $\nabla\psi_\sigma(x)$ is an $O(\sigma)$-approximation of $\nabla\psi(x)$ in most desirable circumstances, it suffices to find an $\epsilon$-stationary solution of $\psi_\epsilon(x)$. However, still directly optimizing $\psi_\sigma(x)$ is not possible since the exact minimizers (in $y$) of $h_\sigma(x, y) := \sigma f(x, y) + g(x, y)$ and $g(x, y)$ are unknown. Thus, we use the alternative min-max formulation:

$$\min_{x \in \mathcal{X}, y \in \mathcal{Y}} \max_{z \in \mathcal{Y}} \psi_\sigma(x, y, z) := \frac{h_\sigma(x, y) - g(x, z)}{\sigma}. \tag{$\mathbf{P}_{\text{saddle}}$}$$

Once we reduce the problem to finding an $\epsilon$-stationary point of the saddle-point problem ($\mathbf{P}_{\text{saddle}}$), we may invoke the rich literature on min-max optimization. However, even when we assume that $g(x, \cdot)$ satisfies the PL conditions *globally* for all $y \in \mathbb{R}^{d_y}$, a plug-in min-max optimization method (*e.g.,* Yang et al. (2022)) yields an oracle-complexity that cannot be better than $O\left(\sigma^{-4}\epsilon^{-4}\right)$ with stochastic oracles Li et al. (2021), resulting in an overall $O(\epsilon^{-8})$ complexity bound when $\sigma = O(\epsilon)$.

As we pursue an $\epsilon$-saddle point specifically in the form of ($\mathbf{P}_{\text{saddle}}$), we show that we can achieve a better complexity bound under Assumption 1. We list below our algorithmic contributions.

- In contrast to previous work on bilevel or min-max optimization, (*e.g.,* Ghadimi & Wang (2018); Ye et al. (2022); Yang et al. (2020)), Assumption 1 holds only in a neighborhood of the lower-level solution. In fact, we show that we only need a nice lower-level landscape within the neighborhood of solutions with $O(\delta)$-proximal error.

- While we eventually set $\sigma = O(\epsilon)$, it can be overly conservative to choose such a small value of $\sigma$ from the first iteration, resulting in a slower convergence rate. By gradually decreasing penalty parameters $\{\sigma_k\}$ polynomially as $k^{-s}$ for some $s > 0$, we save an $O(\epsilon^{-1})$-order of oracle-complexity, improving the overall complexity to $O(\epsilon^{-7})$ with stochastic oracles.

- If stochastic oracles satisfy the mean-squared smoothness condition, *i.e.,* the stochastic gradient is Lipschitz in expectation, then we show a version of our algorithm can be implemented in a fully single-loop manner (*i.e.,* only $O(1)$ calls to stochastic oracles before updating the outer variables $x$) with an improved oracle-complexity of $O(\epsilon^{-5})$ with stochastic oracles.

In conclusion, we show the following result in this paper:

**Theorem 1.2 (Informal)** *There exists an iterative algorithm which finds an $\epsilon$-stationary point $x_\epsilon$ of $\psi_\sigma(x)$ within a total $O(\epsilon^{-7})$ access to stochastic gradient oracles under Assumption 1 with $\sigma = O(\epsilon)$. If the stochastic gradient oracle is mean-squared smoothness, then the algorithm can be implemented in a fully single-loop manner with improved complexity of $O(\epsilon^{-5})$. Furthermore, if the condition required in Theorem 1.1 holds at $x_\epsilon$, then $x_\epsilon$ is an $O(\epsilon)$-stationary point of $\psi(x)$.*

Due to space constraints, in the main text, we only present our landscape analysis of penalty and original formulations, and defer the algorithm specification to appendix.

## 2 PRELIMINARIES

We state several assumptions on (**P**) to specify the problem class of interest. Our focus is on smooth objectives whose values are bounded below:

**Assumption 2** *$f$ and $g$ are twice continuously-differentiable and $l_{f,1}$, $l_{g,1}$-smooth jointly in $(x, y)$ respectively, i.e., $\|\nabla^2 f(x,y)\| \leq l_{f,1}$ and $\|\nabla^2 g(x,y)\| \leq l_{g,1}$ for all $x \in \mathcal{X}$, $y \in \mathcal{Y}$.*

**Assumption 3** *The following conditions hold for objective functions $f$ and $g$:*

1. *$f$ and $g$ are bounded below and coercive, i.e., for all $x \in \mathcal{X}$, $f(x,y), g(x,y) > -\infty$ for all $y \in \mathcal{Y}$, and $f(x,y), g(x,y) \to +\infty$ as $\|y\| \to \infty$.*

2. *$\|\nabla_y f(x,y)\| \leq l_{f,0}$ for all $x \in \mathcal{X}, y \in \mathcal{Y}$.*

We also make technical assumptions on the domains:

**Assumption 4** *The following conditions hold for domains $\mathcal{X}$ and $\mathcal{Y}$:*

1. *$\mathcal{X}, \mathcal{Y}$ are convex and closed.*

2. *$\mathcal{Y}$ is bounded, i.e., $\max_{y \in \mathcal{Y}} \|y\| \leq D_{\mathcal{Y}}$ for some $D_{\mathcal{Y}} = O(1)$. Furthermore, we assume that $C_f := \max_{x \in \mathcal{X}, y \in \mathcal{Y}} |f(x,y)| = O(1)$ is bounded.*

3. *The domain $\mathcal{Y}$ can be compactly expressed with at most $m_1 \geq 0$ inequality constraints $\{g_i(y) \leq 0\}_{i \in [m_1]}$ with convex and twice continuously-differentiable $g_i$, and at most $m_2 \geq 0$ equality constraints $\{h_i(y) = 0\}_{i \in [m_2]}$ with linear functions $h_i$.*

We note here that the expressiveness of inner domain constraints $\mathcal{Y}$ is only required for the analysis, and not required in our algorithms as long as there exist efficient projection operators. While there could be many possible representations of constraints, only the most compact representation would matter in our analysis. We denote by $\Pi_{\mathcal{X}}$ and $\Pi_{\mathcal{Y}}$ the projection operators onto sets $\mathcal{X}$ and $\mathcal{Y}$, respectively. $\mathcal{N}_{\mathcal{X}}(x)$ denotes the normal cone of $\mathcal{X}$ at a point $x \in \mathcal{X}$.

Next, we define the distance measure between sets:

**Definition 2 (Hausdorff Distance)** *Let $S_1$ and $S_2$ be two sets in $\mathbb{R}^d$. The Hausdorff distance between $S_1$ and $S_2$ is given as*

$$\mathbf{dist}(S_1, S_2) = \max\left\{\sup_{\theta_1 \in S_1} \inf_{\theta_2 \in S_2} \|\theta_1 - \theta_2\|, \sup_{\theta_2 \in S_2} \inf_{\theta_1 \in S_1} \|\theta_1 - \theta_2\|\right\}.$$

*For distance between a point $\theta$ and a set $S$, we denote $\mathbf{dist}(\theta, S) := \mathbf{dist}(\{\theta\}, S)$.*

Throughout the paper, we use Definition 2 as a measure of distances between sets. We define the notion of (local) Lipschitz continuity of solution sets introduced in Chen et al. (2023b).

**Definition 3 (Lipschitz Continuity of Solution Sets Chen et al. (2023b))** *For a differentiable and smooth function $f(w, \theta)$ on $\mathcal{W} \times \Theta$, we say the solution set $S(w) := \arg\min_{\theta \in \Theta} f(w, \theta)$ is locally Lipschitz continuous at $w \in \mathcal{W}$ if there exists an open-ball of radius $\delta > 0$ and a constant $L_S < \infty$ such that for any $w' \in \mathbb{B}(w, \delta)$, we have $\mathbf{dist}(S(w), S(w')) \leq L_S \|w - w'\|$.*

**Constrained Optimization**   We introduce some standard notions of regularities from nonlinear constrained optimization Bertsekas (1997). For a general constrained optimization problem $\mathbf{Q}$ : $\min_{\theta \in \Theta} f(\theta)$, suppose $\Theta$ can be compactly expressed with $m_1 \geq 0$ inequality constraints $\{g_i(\theta) \leq 0\}_{i \in [m_1]}$ with convex and twice continuously differentiable $g_i$, and $m_2 \geq 0$ equality constraints $\{h_i(\theta) = 0\}_{i \in [m_2]}$ with linear functions $h_i$.

**Definition 4 (Active Constraints)** *We denote $\mathcal{I}(\theta) \subseteq [m_1]$ the index of active inequality constraints of $\mathbf{Q}$ at $\theta \in \Theta$, i.e., $\mathcal{I}(\theta) := \{i \in [m_1] \, : \, g_i(\theta) = 0\}$.*

**Definition 5 (Linear Independence Constraint Qualification (LICQ))** *We say $\mathbf{Q}$ is regular at a feasible point $\theta \in \Theta$ if the set of vectors consisting of all equality constraint gradients $\nabla h_i(\theta)$, $\forall i \in [m_2]$ and the active inequality constraint gradients $\nabla g_i(\theta)$, $\forall i \in \mathcal{I}(\theta)$ is a linearly independent set.*

A solution $\theta^*$ of $\mathbf{Q}$ satisfies the so-called KKT conditions when LICQ holds at $\theta^*$: the KKT conditions are that there exist unique Lagrangian multipliers $\lambda_i^* \geq 0$ for $i \in \mathcal{I}(\theta^*)$ and $\nu_i^* \in \mathbb{R}$ for $i \in [m_2]$ such that

$$\theta^* \in \Theta \text{ and } \nabla f(\theta^*) + \textstyle\sum_{i \in \mathcal{I}(\theta^*)} \lambda_i^* \nabla g_i(\theta^*) + \sum_{i \in [m_2]} \nu_i^* \nabla h_i(\theta^*) = 0. \tag{5}$$

For such a solution, we define the *strict* complementary slackness condition:

**Definition 6 (Strict Complementarity)** *Let $\theta^*$ be a solution of $\mathbf{Q}$ satisfying LICQ and the KKT condition above. We say that the strict complementary condition is satisfied at $\theta^*$ if there exist multipliers $\lambda^*, \nu^*$ that satisfy (5), and further $\lambda_i^* > 0$ for all $i \in \mathcal{I}(\theta^*)$.*

**Other Notation**   We say $a_k \asymp b_k$ if $a_k$ and $b_k$ decreases (or increases) in the same rate as $k \to \infty$, i.e., $\lim_{k \to \infty} a_k / b_k = \Theta(1)$. Throughout the paper, $\|\cdot\|$ denotes the Euclidean norm for vectors, and operator norm for matrices. $[n]$ with a natural number $n \in \mathbb{N}_+$ denotes a set $\{1, 2, ..., n\}$. Let $\Pi_{\mathcal{S}}(\theta)$ be the projection of a point $\theta$ onto a convex set $\mathcal{S}$. We denote $\mathbf{Ker}(M)$ and $\mathbf{Im}(M)$ to mean the kernel (nullspace) and the image (range) of a matrix $M$ respectively. For a symmetric matrix $M$, we define the pseudo-inverse of $M$ as $M^\dagger := U(U^\top M U)^{-1} U^\top$ where the columns of $U$ consist of eigenvectors corresponding to all non-zero eigenvalues of $M$.

## 3   Landscape Analysis and Penalty Method

In this section, we establish the relationship between the landscapes of the penalty formulation ($\mathbf{P}_{\text{pen}}$) and the original problem ($\mathbf{P}$). Recalling the definition of the perturbed lower-level problem $h_\sigma(x, y) := \sigma f(x, y) + g(x, y)$ from Assumption 1, we introduce the following notation for its solution set: For $\sigma \in [0, \sigma_0]$ with sufficiently small $\sigma_0 > 0$, we define

$$l(x, \sigma) := \min_{y \in \mathcal{Y}} h_\sigma(x, y), \quad T(x, \sigma) := \arg\min_{y \in \mathcal{Y}} h_\sigma(x, y). \tag{6}$$

We call $l(x, \sigma)$ the value function, and $T(x, \sigma)$ the corresponding solution set. Then, the minimization problem ($\mathbf{P}_{\text{pen}}$) over the penalty function and $\psi_\sigma$ defined in (2) can be rewritten as

$$\min_{x \in \mathcal{X}} \psi_\sigma(x) \text{ where } \psi_\sigma(x) = \frac{l(x, \sigma) - l(x, 0)}{\sigma}.$$

We can view $\psi_\sigma(x)$ as a sensitivity measure of how the optimal value $\min_{y \in \mathcal{Y}} g(x, y)$ changes when we impose a perturbation of $\sigma f(x, y)$ in the objective. In fact, it can be easily shown that

$$\lim_{\sigma \to 0} \psi_\sigma(x) = \frac{\partial}{\partial \sigma} l(x, \sigma)|_{\sigma = 0^+} = \psi(x).$$

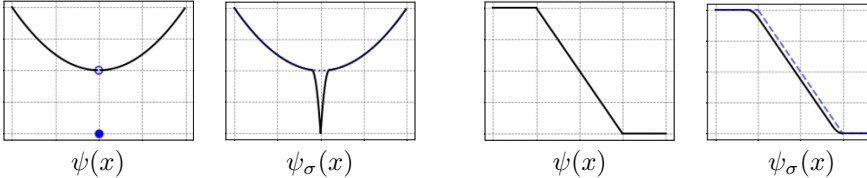

**Figure 1:** $\psi(x)$ and $\psi_\sigma(x)$ in Examples: **(left)** Example 1, **(right)** Example 2. Blue dashed lines compare $\psi_\sigma(x)$ to the original hyper-objective $\psi(x)$.

However, this formula provides only a pointwise asymptotic equivalence of two functions and does not imply the equivalence of the *gradients* $\nabla\psi_\sigma(x)$ and $\nabla\psi(x)$ of the two hyper-objectives. In the limit setting, we check whether

$$\nabla\psi(x) = \frac{\partial^2}{\partial x \partial\sigma}l(x,\sigma)|_{\sigma=0^+} \overset{?}{=} \frac{\partial^2}{\partial\sigma\partial x}l(x,\sigma)|_{\sigma=0^+} = \lim_{\sigma\to 0}\nabla\psi_\sigma(x). \tag{7}$$

Unfortunately, it is not always true that the above relation (7) holds, and the gradient of $\nabla\psi(x)$ may not even be well-defined, as illustrated in Examples 1 and 2 in the following section.

In what follows, we derive two assumptions: one concerning solution-set continuity (Assumption 5) and another addressing the regularities of lower-level constraints (Assumption 6). Under these assumptions, we establish the main theorem of this section, Theorem 3.1. This, in turn, leads to the derivation of the second inequality in our landscape analysis result, as presented in Theorem 1.1 and shown in Theorem 3.7.

### 3.1 Sufficient Conditions for Differentiability

The following two examples illustrate the obstacles encountered when claiming $\lim_{\sigma\to 0}\nabla\psi_\sigma(x) \to \nabla\psi(x)$ or even when simply ensuring the existence of $\nabla\psi(x)$.

**Example 1** *Consider the bilevel problem with $\mathcal{X} = \mathcal{Y} = [-1,1]$, $f(x,y) = x^2 + y^2$, and $g(x,y) = xy$. Note that $\lim_{\sigma\to 0}\psi_\sigma(x) \to x^2 + 1$ for all $x \in [-1,1]\setminus\{0\}$, where we have $\lim_{\sigma\to 0}\psi_\sigma(0) \to 0$. See Figure 1. Therefore $\psi(x)$ is a pointwise convergent point of $\psi_\sigma(x)$. However, neither $\psi(x)$ nor $\psi_\sigma(x)$ is differentiable at $x = 0$.*

This example also implies that the order of (partial) differentiation may not be swapped in general. Even when the lower-level objective $g(x,y)$ is strongly convex in $y$, the inclusion of a constraint set $\mathcal{Y}$, even when compact, can lead to a nondifferentiable $\psi$, as we see next.

**Example 2** *Consider an example with $\mathcal{X} = [-2,2]$, $\mathcal{Y} = [-1,1]$, $f(x,y) = -y$, and $g(x,y) = (y-x)^2$. In this example, $\psi(x) = 1$ if $x < -1$, $\psi(x) = -x$ if $x \in [-1,1]$, and $\psi(x) = -1$ otherwise. Thus $\psi(x)$ is not differentiable at $x = -1$ and $1$, while $\nabla\psi_\sigma(1) = 0$ and $\nabla\psi_\sigma(-1) = -1$ for all $\sigma > 0$.*

There are two reasons for the poor behavior of $\nabla\psi(x)$. First, in Example 1, the solution set moves discontinuously at $x = 0$. When the solution abruptly changes due to a small perturbation in $x$, the problem is highly ill-conditioned — a fundamental difficulty Daskalakis et al. (2021). Second, in Example 2, even though the solution set is continuous thanks to the strong convexity, the solution can move nonsmoothly when the set of active constraints for the lower-level problem changes. The result is frequently nonsmoothness of $\psi(x)$, so $\nabla\psi(x)$ may not be defined at such $x$. In summary, we need regularity conditions to ensure that these two cases do not happen.

We first consider assumptions that obviate solution-set discontinuity so situations like those appearing in Example 1 will not occur.

**Assumption 5** *The solution set $T(x,\sigma) := \arg\min_{y\in\mathcal{Y}} h_\sigma(x,y)$ is locally Lipschitz continuous, i.e., $T(x,\sigma)$ satisfies Definition 3 at $(x,\sigma)$.*

We note here that the differentiability of the value function $l(x, \sigma)$ requires Lipschitz continuity of solution sets (not just continuity). When the solution set is locally Lipschitz continuous, it is known (see Lemma D.2) that the value function $l(x, \sigma)$ is (locally) differentiable and smooth.

As we have seen in Example 2, however, the Lipschitz-continuity of the solution set alone may not be sufficient in the constrained setting. We need additional regularity assumptions for constrained lower-level problems. Recalling the algebraic definition of $\mathcal{Y}$ in Assumption 4, we define the Lagrangian of the constrained ($\sigma$-perturbed) lower-level optimization problem:

**Definition 7** *Given the lower-level feasible set $\mathcal{Y}$ satisfying Assumption 4, let $\{\lambda_i\}_{i=1}^{m_1} \subset \mathbb{R}_+$ and $\{\nu_i\}_{i=1}^{m_2} \subset \mathbb{R}$ be some Lagrangian multipliers. The Lagrangian function $\mathcal{L}(\cdot, \cdot, \cdot | x, \sigma) : \mathbb{R}_+^{m_1} \times \mathbb{R}^{m_2} \times \mathbb{R}^{d_y} \to \mathbb{R}$ at $(x, \sigma)$ is defined by*

$$\mathcal{L}(\lambda, \nu, y | x, \sigma) = \sigma f(x, y) + g(x, y) + \sum_{i \in [m_1]} \lambda_i g_i(y) + \sum_{i \in [m_2]} \nu_i h_i(y).$$

*We also define the Lagrangian $\mathcal{L}_{\mathcal{I}}(\cdot | x, \sigma) : \mathbb{R}_+^{|\mathcal{I}|} \times \mathbb{R}^{m_2} \times \mathbb{R}^{d_y} \to \mathbb{R}$ restricted to a set of constraints $\mathcal{I} \subseteq [m_1]$:*

$$\mathcal{L}_{\mathcal{I}}(\lambda_{\mathcal{I}}, \nu, y | x, \sigma) = \sigma f(x, y) + g(x, y) + \sum_{i \in \mathcal{I}} \lambda_i g_i(y) + \sum_{i \in [m_2]} \nu_i h_i(y). \tag{8}$$

When the context is clear, we always let $\mathcal{I}$ in (8) be $\mathcal{I}(y)$ at a given $y$. The required assumption is the existence of a regular and stable solution that satisfies the following:

**Assumption 6** *The solution set $T(x, \sigma)$ contains at least one $y^* \in T(x, \sigma)$ such that LICQ (Definition 5) and strict complementary condition (Definition 6) hold at $y^*$, and $\nabla^2 \mathcal{L}_{\mathcal{I}}(\cdot, \cdot, \cdot | x, \sigma)$ (the matrix of second derivatives with respect to all variables $\lambda_{\mathcal{I}}, \nu, y$) is continuous at $(\lambda_{\mathcal{I}}^*, \nu^*, y^*)$.*

Assumption 6 helps ensure that the active set $\mathcal{I}(y^*)$ given in Definition 4 does not change when $x$ or $\sigma$ is perturbed slightly.

### 3.2 Asymptotic Landscape

We show that Assumptions 5 and 6 are nearly minimal to ensure the twice-differentiability of the value function $l(x, \sigma)$ which, in turn, guarantees asymptotic equivalence of $\nabla \psi_\sigma(x)$ and $\nabla \psi(x)$. In the sequel, we state our main (local) landscape analysis result only under the two assumptions at a given point $(x, \sigma)$, which is given in the following theorem:

**Theorem 3.1** *Suppose $T(\cdot, \cdot)$ satisfies Assumption 5 in a neighborhood of $(x, \sigma)$. If there exists at least one $y^* \in T(x, \sigma)$ that satisfies Assumption 6, then $\frac{\partial^2}{\partial x \partial \sigma} l(x, \sigma)$ exists and can be given explicitly by*

$$\frac{\partial^2}{\partial \sigma \partial x} l(x, \sigma) = \frac{\partial^2}{\partial x \partial \sigma} l(x, \sigma) \tag{9}$$

$$= \nabla_x f(x, y^*) - \begin{bmatrix} 0 & \nabla_{xy}^2 h_\sigma(x, y^*) \end{bmatrix} (\nabla^2 \mathcal{L}_{\mathcal{I}}(\lambda_{\mathcal{I}}^*, \nu^*, y^* | x, \sigma))^\dagger \begin{bmatrix} 0 \\ \nabla_y f(x, y^*) \end{bmatrix}.$$

*If this equality holds at $\sigma = 0^+$, then $\psi(x)$ is differentiable at $x$, and $\lim_{\sigma \to 0} \nabla \psi_\sigma(x) = \nabla \psi(x)$.*

Theorem 3.1 generalizes the expression of $\nabla \psi(x)$ from the case of a unique solution to the one with multiple solutions, significantly enlarging the scope of tractable instances of BO. Up to our best knowledge, there are no previous results that provide an explicit formula of $\nabla \psi(x)$, even when the solution set is Lipschitz continuous, though conjectures have been made in the literature Xiao et al. (2023a); Arbel & Mairal (2022) under similar conditions.

**Remark 3.2 (Set Lipschitz Continuity)** *While we require the entire solution set $T(x, \sigma)$ to be Lipschitz continuous, the proof indicates that we need only Lipschitz continuity of solution paths passing through $y^*$ (that defines the first-order derivative of $l(x, \sigma)$ and satisfies other regularity conditions) in all possible perturbation directions of $(x, \sigma)$. Nonetheless, we stick to a stronger requirement of Definition 3, since our algorithm requires a stronger condition that implies the continuity of entire solution sets.*

**Remark 3.3 (Lipschitz Continuity in $\sigma$)** *While we require $T(x, \sigma)$ to be Lipschitz continuous in both $x$ and $\sigma$, well definedness of $\nabla \psi(x)$ requires only Lipschitz continuity of $T(x, 0^+)$ in $x$ (which sometimes can be implied by the PL condition only on $g$ as in Chen et al. (2023b); Shen & Chen (2023); Xiao et al. (2023a)). Still, for implementing a stable and efficient algorithm with stochastic oracles, we conjecture that it is essential to have the Lipschitz continuity assumption on the additional axis $\sigma$.*

We conclude our asymptotic landscape analysis with a high-level discussion on Theorem 3.1. The key step in our proof is to prove the following proposition:

**Proposition 3.4 (Necessary Condition for Lipschitz Continuity)** *Suppose $T(x, \sigma)$ satisfies Assumption 5 at $(x, \sigma)$. For any $y^* \in T(x, \sigma)$ that satisfies Assumption 6, the following must hold:*

$$\forall v \in \text{span}(\mathbf{Im}(\nabla_{yx}^2 h_\sigma(x, y^*)), \nabla_y f(x, y^*)) : \begin{bmatrix} 0 \\ v \end{bmatrix} \in \mathbf{Im}(\nabla^2 \mathcal{L}_\mathcal{I}(\lambda_\mathcal{I}^*, \nu^*, y^* | x, \sigma)). \quad (10)$$

Formally, perturbations in $(x, \sigma)$ must not tilt the flat directions of the lower-level landscape for $T(x, \sigma)$ to be continuous. While it is easy to make up examples that do not meet the condition (*e.g.* Example 1), several recent results show that the solution landscape may be stabilized for complicated functions such as over-parameterized neural networks Frei & Gu (2021); Song et al. (2021); Liu et al. (2022). In Appendix E.5, we prove a more general version of Theorem 3.1 (see Theorem E.4), of possible broader interest, concerning the Hessian of $l(x, \sigma)$.

**Remark 3.5** *With a more standard assumption on the uniqueness of the lower-level solution and the invertibility of the Lagrangian Hessian, we can provide a sufficiency guarantee for the Lipschitz continuity of solution sets stronger than Theorem 3.1. We refer the readers to Appendix E.1.*

### 3.3 Landscape Approximation with $\sigma > 0$

We can view $\nabla \psi_\sigma(x)$ as an approximation of $\frac{\partial}{\partial \sigma} \left( \frac{\partial}{\partial x} l(x, \sigma) \right) |_{\sigma = 0^+}$ via finite differentiation with respect to $\sigma$. Assuming that $\frac{\partial^2}{\partial \sigma \partial x} l(x, \sigma)$ exists and is continuous for all small values of $\sigma$, we can apply the mean-value theorem and conclude that $\nabla \psi_\sigma(x) = \frac{\partial^2}{\partial \sigma \partial x} l(x, \sigma')$ for some $\sigma' \in [0, \sigma]$. Thus, $\|\nabla \psi_\sigma(x) - \nabla \psi(x)\|$ is $O(\sigma)$ whenever $\frac{\partial^2}{\partial x \partial \sigma} l(x, \sigma)$ is well-defined and uniformly Lipschitz continuous over $[0, \sigma]$.

To work with nonzero constant $\sigma > 0$, we need the regularity assumptions to hold in significantly larger regions. The crux of Assumption 1 is the guaranteed Lipschitz continuity of solution sets (see also Assumption 5) for every given $x$ and $\sigma$, which is also crucial for the tractability of lower-level solutions by local search algorithms whenever upper-level variable changes:

**Lemma 3.6** *Under Assumption 1, $T(x, \sigma)$ is $(l_{g,1}/\mu)$-Lipschitz continuous in $x$ and $(l_{f,0}/\mu)$-Lipschitz continuous in $\sigma$ for all $x \in \mathcal{X}, \sigma \in [0, \delta/C_f]$.*

An additional consequence of Lemma 3.6 is that, by Lemma D.2, $l(x, \sigma)$ is continuously differentiable and smooth for all $x \in \mathcal{X}$ and $\sigma \in [0, \delta/C_f]$. This fact guarantees in turn that $\psi_\sigma(x)$ is differentiable and smooth (though $\psi(x)$ does not necessarily have these properties).

While Assumption 1 is sufficient to ensure $\psi_\sigma(x)$ is well-behaved, we need additional regularity conditions to ensure that $\psi(x)$ is also well-behaved. Therefore when we connect $\psi(x)$ and $\psi_\sigma(x)$, we make two more *local* assumptions that are non-asymptotic versions of Assumption 5 and 6. The first concerns Hessian-Lipschiztness and regularity of solutions.

**Assumption 7** *For a given $x$, there exists at least one $y^* \in T(x, 0)$ such that if we follow the solution path $y^*(\sigma)$ along the interval $\sigma \in [0, \sigma_0]$,*

*(1) all $y^*(\sigma)$ satisfies Assumption 6 with active constraint indices $\mathcal{I}$ and Lagrangian multipliers $\lambda_\mathcal{I}^*(\sigma), \nu^*(\sigma)$ of size $O(1)$ and*

*(2) $\nabla^2 f, \nabla^2 g, \{\nabla^2 g_i\}_{i=1}^{m_1}$ are $l_{h,2}$-Lipschitz continuous at all $(x, y^*(\sigma))$.*

In the unconstrained settings with Hessian-Lipschitz objectives, Assumption 7 is implied by Assumption 1 for all $x \in \mathcal{X}, \sigma \in [0, \delta/C_f]$.

The second assumption is on the minimum nonzero singular value of active constraint gradients.

**Assumption 8** *For a given $x$, there exists at least one $y^* \in T(x, 0)$ such that if we follow the solution path $y^*(\sigma)$ along the interval $\sigma \in [0, \sigma_0]$, all solutions $y^*(\sigma)$ satisfy Definition 5 with minimum singular value $s_{\min} > 0$. That is, for all $\sigma \in [0, \sigma_0]$, we have*

$$\min_{v:\|v\|_2=1} \|[\nabla g_i(y^*(\sigma)), \forall i \in \mathcal{I} \quad | \nabla h_i(y^*(\sigma)), \forall i \in [m_2]] v\| \geq s_{\min}.$$

Note that $s_{\min}$ in the constrained setting depends purely on the LICQ condition, Definition 5.

**Theorem 3.7** *Under Assumptions 1 - 4, we have*

$$|\psi_\sigma(x) - \psi(x)| \leq O\left(l_{f,0}^2/\mu\right) \cdot \sigma,$$

*for all $x \in \mathcal{X}$ and $\sigma \in [0, \frac{\delta}{2C_f}]$. If, in addition, Assumptions 7 and 8 hold at a given $x$, then*

$$\|\nabla\psi_\sigma(x) - \nabla\psi(x)\| \leq O\left(\frac{l_{g,1}^4 l_{f,0}^3}{\mu^3 s_{\min}^3} + \frac{l_{h,2} l_{g,1}^2 l_{f,0}^3}{\mu^3 s_{\min}^2}\right) \cdot \sigma.$$

The proof of Theorem 3.7 is given in Appendix E.6.

**Remark 3.8 (Change in Active Sets)** *A slightly unsatisfactory conclusion of Theorem 3.7 is that when $\nabla\psi(x)$ is not well-defined due to the nonsmooth movement of the solution set as in Example 2, it does not relate $\nabla\psi_\sigma(x)$ to any alternative measure for $\nabla\psi(x)$. Around the point where $\psi(x)$ is non-smooth, some concurrent work attempts to find a so-called $(\epsilon, \delta)$-Goldstein stationary point Chen et al. (2023b), which can be seen as an approximation of gradients via localized smoothing (but only in the upper-level variables $x$). While this is an interesting direction, we do not pursue it here. Instead, we conclude this section by stating that an $\epsilon$-stationary solution of $\psi_\sigma(x)$ is an $O(\epsilon + \sigma)$-KKT point of ($P_{\text{con}}$) (this claim is fairly straightforward to check, see for example, Theorem E.5 in Appendix E.7).*

### 3.4 FINDING STATIONARY POINT OF $\psi_\sigma(x)$

Theorem 3.7 explains the conditions and mechanisms under which the penalty methods can yield an approximate solution of the original Bilevel optimization problem (P), providing a rationale for pursuing a weaker criterion of $\|\nabla\psi_\sigma(x)\| \leq \epsilon$ with $\sigma = O(\epsilon)$. In Appendix B, we present algorithms that rely solely on access to first-order (stochastic) gradient oracles and find a stationary point of the penalty function $\psi_\sigma(x)$ with $\sigma = O(\epsilon)$. Then we conclude the paper by providing formal versions of Theorem 1.2 in Appendix C.

## 4 CONCLUSION

This paper studies a first-order algorithm for solving Bilevel Optimization when the lower-level problem and perturbed versions of it satisfy a proximal error bound condition when the errors are small. We establish an $O(\sigma)$-closeness relationship between the penalty formulation $\psi_\sigma(x)$ and the hyper-objective $\psi(x)$ under the proximal-error bound condition, and then we develop a fully first-order stochastic approximation scheme for finding a stationary point of $\psi_\sigma(x)$, and study its non-asymptotic performance guarantees. We believe our algorithm to be simple and general, and useful in many large-scale scenarios that involve nested optimization problems. In Appendix A.2, we discuss several issues not addressed in this paper, that may become the subjects of fruitful future research.

ACKNOWLEDGMENTS

This project was partially supported by AFOSR/AFRL grant FA9550-18-1-0166. DK was partially supported by the National Research Foundation of Korea (NRF) grant funded by the Korea government (MSIT) (No. RS-2023-00252516) and the POSCO Science Fellowship of POSCO TJ Park Foundation. SW was supported in part by NSF Awards DMS-2023239 and CCF-2224213 and AFOSR Award FA9550-21-1-0084.

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

## APPENDIX A    DEFERRED DISCUSSIONS

### A.1    RELATED WORK

Since its introduction in Bracken & McGill (1973), Bilevel optimization has been an important research topic in many scientific disciplines. Classical results tend to focus on the asymptotic properties of algorithms once in neighborhoods of global/local minimizers (see *e.g.,* White & Anandalingam (1993); Vicente et al. (1994); Colson et al. (2007)). In contrast, recent results are more focused on studying numerical optimization methods and non-asymptotic analysis to obtain an approximate stationary solution of Bilevel problems (see *e.g.,* Ghadimi & Wang (2018); Chen et al. (2021)). Our work falls into this category. Due to the vast volume of literature on Bilevel optimization, we only discuss some relevant lines of work.

**Implicit-Gradient Descent**    As mentioned earlier, initiated by Ghadimi & Wang (2018), a flurry of recent works (see *e.g.,* Chen et al. (2021); Hong et al. (2023); Khanduri et al. (2021); Chen et al. (2022); Dagréou et al. (2022); Guo et al. (2021)) study stochastic-gradient-descent (SGD)-based iterative procedures and their finite-time performance for solving (**P**) when the lower-level problem is strongly-convex and unconstrained. In such cases, the implicit gradient of hyper-objective $\psi(x)$ is given by

$$\nabla_x f(x, y^*(x)) - \nabla^2_{xy} g(x, y^*(x)) \left( \nabla^2_{yy} g(x, y^*(x)) \right)^{-1} \nabla_y f(x, y^*(x)),$$

where $y^*(x) = \arg\min_{w \in \mathbb{R}^{d_y}} g(x, w)$. Two main challenges in performing (implicit) gradient descent are (a) to evaluate lower-level solution $y^*(x)$, and (b) to estimate Hessian inverse $(\nabla^2_{yy} g(x, y^*(x)))^{-1}$. For (a), it is now well-understood that instead of exactly solving for $y^*(x_k)$ for every $k^{th}$ iteration, we can incrementally solve for $y^*(x_k)$, *e.g.,* run a few more gradient steps on the current estimate $y_k$, and use it as a proxy for the lower-level solution Chen et al. (2021). As long as the contraction toward the true solution (with the strong convexity of $g(x_k, \cdot)$) is large enough to compensate for the change of lower-level solution (due to the movement in $x_k$), $y_k$ will eventually stably stay around $y^*(x_k)$, and can be used as a proxy for the lower-level solution to compute the implicit gradient. Then for (b), with the Hessian of $g$ being invertible for all given $y$, we can exploit the Neumann series approximation Ghadimi & Wang (2018) to estimate the true Hessian inverse using $y_k$ as a proxy for $y^*(x_k)$.

Unfortunately, the above results are not easily extendable to nonconvex lower-level objectives with potential constraints $\mathcal{Y}$. One obstacle is, again, to estimate the Hessian-inverse: now that for some $y \in \mathcal{Y}$, the Hessian of $g$ may not be invertible even if $\nabla^2_{yy} g(x, y^*(x))$ is invertible at the exact solution. Therefore, in order to use an approximate $y_k$ as a proxy to $y^*(x_k)$, we need a certain high-probability guarantee (or some other complicated arguments) to ensure that the algorithm remains stable with the inversion operation. The other obstacle, which is more complicated to resolve, is that the implicit gradient formula may no longer be the same if the solution is found at the boundary of $\mathcal{Y}$. In such a case, explicitly estimating $\nabla \psi(x)$ would not only require an approximate solution but also require the optimal dual variables for the lower-level solutions which are unknown (see Theorem 3.1 for the exact formula). However, computing the optimal solution as well as the optimal dual variables is even more challenging when we only access objective functions through stochastic oracles. Therefore, it is essential to develop a first-order method that does not rely on the explicit estimation of the implicit gradient to solve a broader class of BO, aside from the cost of using second-order derivatives in large-scale applications.

**Nonconvex Lower-Level Objectives**    In general, BO with nonconvex lower-level objectives is not computationally tractable without further assumptions, even for the special case of min-max optimization Daskalakis et al. (2021). Therefore, additional assumptions on the lower-level problem are necessary. Arguably, the minimal assumption would be the continuity of lower-level solution sets, otherwise, any local-search algorithms are likely to fail due to the hardness of (approximately) tracking the lower-level problem. The work in Chen et al. (2023b) considers several growth conditions for the lower-level objectives (including PL), which guarantee Lipschitz continuity of lower-level solution sets, and proposes a zeroth-order method for solving (**P**). The work in Shen & Chen (2023) assumes the PL condition, and studies the complexity of the penalty method in deterministic settings. The work in Arbel & Mairal (2022) introduces a notion of parameteric Morse-Bott functions, and studies some asymptotic properties of their proposed gradient flow under the proposed condition. In

all these works as well as in our work, underlying assumptions involve some growth conditions of the lower-level problem, which is essential for the continuity of lower-level solution sets.

**Penalty Methods** Studies on penalty methods date back to 90s Marcotte & Zhu (1996); White & Anandalingam (1993); Anandalingam & White (1990); Ye et al. (1997); Ishizuka & Aiyoshi (1992), when the equivalence is established between two formulations (**P**) and (**P**$_{\text{pen}}$) for sufficiently small $\sigma > 0$. However, these results are often limited to relations within *infinitesimally* small neighborhoods of global/local minimizers. As we aim to obtain a stationary solution from arbitrary initial points, we need a comprehensive understanding of approximating the *global* landscape, rather than only around infinitesimally small neighborhoods of global/local minimizers.

The most closely related work to ours is a recent work in Shen & Chen (2023), where the authors study the penalty method under lower-level PL-like conditions with constraints $\mathcal{Y}$ as in ours. However, the connection established in Shen & Chen (2023) only relates (**P**$_{\text{pen}}$) and $\epsilon$-relaxed version of (**P**$_{\text{con}}$), and only concerns infinitesimally small neighborhoods of global/local minimizers as in older works. Furthermore, their analysis is restricted to the double-loop implementation of algorithms in deterministic settings. In contrast, we establish a direct connection between $\nabla\psi(x)$ and $\nabla\psi_\sigma(x)$ when they are well-defined, and our analysis can be applied to both double-loop and single-loop algorithms with explicit oracle-complexity bounds in stochastic settings.

**Implicit Differentiation Methods** Another popular approach to side-step the computation of implicit gradients is to construct a chain of lower-level variables via gradient descents, a technique often called automatic implicit differentiation (AID), or iterative differentiation (ITD) Pedregosa (2016); Yang et al. (2021); Li et al. (2022b); Ji et al. (2021); Grazzi et al. (2023). The benefit of this technique is that now we do not require the estimation of Hessian-inverse. In fact, this construction can be seen as one constructive way of approximating the Hessian-inverse. However, when the lower-level problem is constrained by compact $\mathcal{Y}$, more complicated operations such as the projection may prevent the use of the implicit differentiation technique.

### A.2 FUTURE DIRECTIONS

**Tightness of Results.** Can our complexity result can be improved in terms of its dependence on $\epsilon$ while using only first-order oracles? Recent work in Chen et al. (2023a) shows that when the lower-level problem is unconstrained and strongly convex, oracle complexity can be improved to $O(\epsilon^{-2})$ with deterministic first-order gradient oracles. Can similar improvements be found in the complexity when stochastic oracles and constraints are present in the formulation?

**Lower Level $x$-Dependent Constraints.** When the lower-level constraints depend on $x$, it is also possible to derive an implicit gradient formula when the lower level problem is non-degenerate. For instance, Xiao et al. (2023b) has studied the case in which the lower-level objective is strongly convex and there are lower-level linear equality constraints that depend on $x$. In general, with $x$-dependent constraints, we cannot avoid estimating Lagrangian multipliers, as they are needed in the implicit gradient formula. Even to find the stationary point of penalty functions, $\nabla\psi_\sigma(x)$ requires Lagrangian multipliers (see the Envelope Theorem Milgrom & Segal (2002)). An interesting future direction would be to develop an efficient first-order algorithm for this case.

**General Convex Lower-Level.** One interesting special case is when $g(x,\cdot)$ is merely convex, not necessarily strongly convex. There have been recent advances in min-max optimization for nonconvex-concave problems; see for example Boroun et al. (2023); Thekumparampil et al. (2019); Kovalev & Gasnikov (2022); Kong & Monteiro (2021); Ostrovskii et al. (2021); Zhang et al. (2020). We note that when $g(x,\cdot)$ is convex, an $\epsilon$-stationary point of (**P**$_{\text{saddle}}$) is also an $\epsilon$-KKT solution of (**P**$_{\text{con}}$). The first paper to investigate this direction in deterministic settings is Lu & Mei (2023), to our knowledge. An important future direction would be to extend their results to stochastic settings.

**Nonsmooth Objectives.** We could also consider nonsmooth objectives in both levels where efficient proximal operators are available for handling the nonsmoothness. It would also be interesting in future work to see whether the analysis in this paper needs to be changed significantly in order to handle nonsmooth objectives.

## APPENDIX B    ALGORITHM

We have the following assumptions on the first-order (stochastic) oracles and efficient projection operators required to develop our algorithms.

**Assumption 9** *The projection operations $\Pi_{\mathcal{X}}, \Pi_{\mathcal{Y}}$ onto sets $\mathcal{X}, \mathcal{Y}$, respectively, can be implemented efficiently.*

**Assumption 10** *We access first-order information about the objective functions via unbiased estimators $\nabla f(x, y; \zeta), \nabla g(x, y; \xi)$, where $\mathbb{E}[\nabla f(x, y; \zeta)] = \nabla f(x, y)$ and $\mathbb{E}[\nabla g(x, y; \xi)] = \nabla g(x, y)$. The variances of stochastic gradient estimators are bounded as follows:*

$$\mathbb{E}[\|\nabla f(x, y; \zeta) - \nabla f(x, y)\|^2] \le \sigma_f^2, \quad \mathbb{E}[\|\nabla g(x, y; \xi) - \nabla g(x, y)\|^2] \le \sigma_g^2,$$

*for some universal constants $\sigma_f^2, \sigma_g^2 \ge 0$.*

Throughout the rest of the paper, we assume that Assumption 1 holds.

### B.1    STATIONARITY MEASURES

Since we showed in the previous section that $\nabla \psi_\sigma(x)$ is an $O(\sigma)$-approximation of $\nabla \psi(x)$ in most desirable circumstances, now we consider finding a stationary point $(x^*, y^*, z^*)$ of $\nabla \psi_\sigma(x)$. Under Assumption 1, we can show that this is equivalent to finding the stationary point of ($\mathbf{P}_{\text{saddle}}$) defined as the following:

$$y^* = \mathbf{prox}_{\rho h_\sigma(x^*, \cdot)}(y^*), \quad z^* = \mathbf{prox}_{\rho g(x^*, \cdot)}(z^*), \quad -(\nabla_x h_\sigma(x^*, y^*) - \nabla_x g(x^*, z^*)) \in \mathcal{N}_{\mathcal{X}}(x^*), \tag{11}$$

where $\mathcal{N}_{\mathcal{X}}(x^*)$ is the normal cone of $\mathcal{X}$ at $x^*$. We define a notion of *approximate* stationary points as follows.

**Definition 8** *We say $(x, y, z)$ is an $\epsilon$-stationary point of ($\mathbf{P}_{\text{saddle}}$) if it satisfies the following:*

$$\frac{1}{\rho}\|y - \mathbf{prox}_{\rho h_\sigma(x, \cdot)}(y)\| \le \sigma\epsilon, \quad \frac{1}{\rho}\|z - \mathbf{prox}_{\rho g(x, \cdot)}(z)\| \le \sigma\epsilon,$$

$$\frac{1}{\rho}\|x - \Pi_{\mathcal{X}}\{x - \rho(\nabla_x h_\sigma(x, y) - \nabla_x g(x, z))\}\| \le \sigma\epsilon.$$

The lemma below relates the $\epsilon$-stationarity of ($\mathbf{P}_{\text{saddle}}$) to the landscape of $\nabla \psi_\sigma(x)$:

**Lemma B.1** *Let $(x^*, y^*, z^*)$ be an $\epsilon$-stationary point of ($\mathbf{P}_{\text{saddle}}$).*

1. *For all $x \in \mathcal{X}$, $\nabla \psi_\sigma(x)$ is well-defined, and $x^*$ is a $(1 + l_{g,1}/\mu)\epsilon$-stationary point of $\psi_\sigma(x)$.*

2. *Supposing in addition that Assumptions 7 and 8 hold at $x^*$, then $x^*$ is a $((1+l_{g,1}/\mu)\epsilon+L_\sigma\sigma)$-stationary point of $\psi(x)$, where $L_\sigma = O\left(l_{g,1}^4 l_{f,0}^3/(\mu^3 s_{\min}^3)\right)$.*

The first part of the lemma is a consequence of Lemma 3.6, while the second part is from Theorem 3.7. Henceforth, we aim to find a saddle point of formulation ($\mathbf{P}_{\text{saddle}}$).

### B.2    FIRST-ORDER METHOD WITH LARGE BATCHES

We first consider solving a stochastic saddle-point problem by applying (projected) stochastic gradient descent-ascent, alternating between upper-level and lower-level variables, with multiple iterations for the lower-level variables $(y, z)$ per single iteration in the upper-level variables $(x)$. There are two technical challenges that we aim to tackle specifically for the form ($\mathbf{P}_{\text{saddle}}$) with Assumption 1.

---

**Algorithm 1** Double-Loop Algorithm with Large Batches

---

**Input:** total outer-loop iterations: $K$, step sizes: $\{\alpha_k, \gamma_k\}$, proximal-smoothing parameters: $\{\beta_k : \beta_k \in (0,1]\}$, inner-loop iteration counts: $\{T_k\}$, outer-loop batch size: $\{M_k\}$, penalty parameters: $\{\sigma_k\}$, proximal parameter: $\rho$, initializations: $x_0 \in \mathcal{X}, y_0, z_0 \in \mathcal{Y}$

1: Initialize $w_{y,0} = y_0$, $w_{z,0} = z_0$
2: **for** $k = 0...K - 1$ **do**
3:    # Inner-Loop Proximal-Operation Solvers
4:    $u_0 \leftarrow w_{y,k}, v_0 \leftarrow w_{z,k}$
5:    **for** $t = 0, ..., T_k - 1$ **do**
6:       $u_{t+1} \leftarrow \Pi_{\mathcal{Y}} \left\{ u_t - \gamma_k (\sigma_k f_{wy}^{k,t} + g_{wy}^{k,t} + \rho^{-1}(u_t - y_k)) \right\}$
7:       $v_{t+1} \leftarrow \Pi_{\mathcal{Y}} \left\{ v_t - \gamma_k (g_{wz}^{k,t} + \rho^{-1}(v_t - z_k)) \right\}$
8:    **end for**
9:    $w_{y,k+1} \leftarrow u_T, w_{z,k+1} \leftarrow v_T$
10:   # Proximal-Smoothing on Lower-Level Variables
11:   $y_{k+1} \leftarrow (1 - \beta_k)y_k + \beta_k w_{y,k+1}$
12:   $z_{k+1} \leftarrow (1 - \beta_k)z_k + \beta_k w_{z,k+1}$
13:   # (Projected) Gradient Descent on Upper-Level Variables
14:   $x_{k+1} \leftarrow \Pi_{\mathcal{X}} \left\{ x_k - \frac{\alpha_k}{M_k} \sum_{m=1}^{M_k} (\sigma_k f_x^{k,m} + g_{xy}^{k,m} - g_{xz}^{k,m}) \right\}$
15: **end for**

---

1. Technically speaking, the main difference from many previous works (*e.g.,* Hong et al. (2023); Chen et al. (2021; 2022)) is that now we no longer have a global contraction property of inner iterations toward solution sets. To be more specific, when the lower-level objective the PL-condition for all $y \in \mathbb{R}^{d_y}$, the distance between the current (lower-level) iterates and solution-sets contracts globally after applying inner gradient steps, *i.e.,* if updating $z_{k+1} \leftarrow z_k - \beta_k \nabla_y g(x_k, z_k)$ at the $k^{th}$ iteration before updating $x_k$, we get

$$\mathbb{E}[\mathbf{dist}(z_{k+1}, T(x_k, 0))|\mathcal{F}_k] \leq (1 - \lambda_k) \cdot \mathbf{dist}(z_k, T(x_k, 0)),$$

   for some $\lambda_k \in (0, 1]$. However, we assume that the error-bound condition only holds at points with $O(\delta)$ proximal error. That is, unless $y$ and $z$ remain close to the solution set (with high probability if gradient oracles are stochastic), we cannot guarantee that $\mathbf{dist}(z_k, T(x_k, 0))$ is improved (in expectation) as the outer-iteration $k$ proceeds.

2. Eventually, we want $\sigma = O(\epsilon)$ since $\psi_\sigma(x)$ is ideally an $O(\sigma)$-approximation of $\psi(x)$ up to first-order. However, to set $\sigma = O(\epsilon)$ from the first iteration is overly conservative, resulting in an overall slowdown of convergence. We decrease the penalty parameters $\{\sigma_k\}$ gradually, to improve the overall convergence rates and the gradient oracle complexity.

To address issue 1, we propose a smoothed surrogate of $\psi_\sigma(x, y, z)$ via proximal envelope (often referred to as Moreau Envelope Moreau (1965)) with sufficiently small $\rho \ll 1/l_{g,1}$:

$$h_{\sigma,\rho}^*(x, y) = \min_{w \in \mathcal{Y}} \left( h_\sigma(x, y, w) := h_\sigma(x, w) + \frac{1}{2\rho}\|w - y\|^2 \right),$$

$$g_\rho^*(x, z) = \min_{w \in \mathcal{Y}} \left( g(x, y, w) := g(x, w) + \frac{1}{2\rho}\|w - z\|^2 \right), \tag{12}$$

and consider the following alternative saddle-point problem with proximal envelopes:

$$\min_{x \in \mathcal{X}, y \in \mathcal{Y}} \max_{z \in \mathcal{Y}} \psi_{\sigma,\rho}(x, y, z) := \frac{h_{\sigma,\rho}^*(x, y) - g_\rho^*(x, z)}{\sigma}. \tag{13}$$

This formulation is convenient because the inner-minimization problem is strongly convex, so we always have a unique and well-defined lower-level optimizer to chase.

Note that $\nabla h_{\sigma,\rho}^*(x, y) = \nabla h_\sigma(x, w_y^*)$ where $w_y^* = \mathbf{prox}_{\rho h_\sigma(x,\cdot)}(y)$, and similarly, $\nabla g_\rho^*(x, z) = \nabla g(x, w_z^*)$ where $w_z^* = \mathbf{prox}_{\rho g(x,\cdot)}(z)$. That is, to apply gradient descent-ascent on $\psi_{\sigma,\rho}(x, y, z)$, we need only solve for proximal operations associated with $h_\sigma(x, \cdot)$ and $g(x, \cdot)$. While we may not be able to compute the proximal operators exactly, we can introduce intermediate variables $w_{y,k}, w_{z,k}$

---

**Algorithm 2** Single Loop Algorithm with Momentum Assistance

---

**Input:** total outer-loop iterations: $K$, step sizes: $\{\alpha_k, \gamma_k\}$, proximal-smoothing parameters: $\{\beta_k : \beta_k \in (0,1]\}$ penalty parameters: $\{\sigma_k\}$, momentum schedulers: $\{\eta_k : \eta_k \in (0,1]\}$, proximal parameter: $\rho$, initializations: $x_0 \in \mathcal{X}, y_0, z_0 \in \mathcal{Y}$

1: Initialize $w_{y,0} = y_0$, $w_{z,0} = z_0$
2: **for** $k = 0...K-1$ **do**
3:     # Proximal-Operation Solvers
4:     $w_{y,k+1} \leftarrow \Pi_{\mathcal{Y}} \left\{ w_{y,k} - \gamma_k(\sigma_k \widetilde{f}_{wy}^k + \widetilde{g}_{wy}^k + \rho^{-1}(w_{y,k} - y_k)) \right\}$
5:     $w_{z,k+1} \leftarrow \Pi_{\mathcal{Y}} \left\{ w_{z,k} - \gamma_k(\widetilde{g}_{wz}^k + \rho^{-1}(w_{z,k} - z_k)) \right\}$
6:     # Proximal-Smoothing on Lower-Level Variables
7:     $y_{k+1} \leftarrow (1-\beta_k)y_k + \beta_k w_{y,k+1}$
8:     $z_{k+1} \leftarrow (1-\beta_k)z_k + \beta_k w_{z,k+1}$
9:     # (Projected) Gradient Descent on Upper-Level Variables
10:     $x_{k+1} \leftarrow \Pi_{\mathcal{X}} \left\{ x_k - \alpha_k \left( \sigma_k \widetilde{f}_x^k + \widetilde{g}_{xy}^k - \widetilde{g}_{xz}^k \right) \right\}$
11: **end for**

---

that chase the solution of proximal envelopes. We then design the inner loop of the algorithm to solve the proximal operation using $T_k$ inner iterations. Later, we make particular choices of the number of inner iterations $T_k$ to achieve the best oracle complexity and convergence rates.

To address issue 2 above, we simply choose $\sigma_k = k^{-s}$ for some chosen constant $s > 0$. This rate of decrease of $\sigma_k$ is optimized to achieve the best oracle complexity and convergence rates to reach an $\epsilon$-stationary point of $\psi_{\epsilon,\rho}(x,y,z)$. We summarize the overall double-loop implementation in Algorithm 1, where we define:

$$f_{wy}^{k,t} = \nabla_y f(x_k, u_t; \zeta_{wy}^{k,t}), \qquad g_{wy}^{k,t} = \nabla_y g(x_k, u_t; \xi_{wy}^{k,t}), \qquad g_{wz}^{k,t} = \nabla_y g(x_k, v_t; \xi_{wz}^{k,t}),$$
$$f_x^{k,m} = \nabla_x f(x_k, w_{y,k+1}; \zeta_x^{k,m}), \quad g_{xy}^{k,m} = \nabla_x g(x_k, w_{y,k+1}; \xi_{xy}^{k,m}), \quad g_{xz}^{k,m} = \nabla_x g(x_k, w_{z,k+1}; \xi_{xz}^{k,m}).$$

We mention here that one may try $T_k = M_k = O(1)$, in which case Algorithm 1 becomes a single-loop algorithm. However, as we see in the analysis, the optimal scheduling of $T_k$ and $M_k$ should increase with $k$ (see also Remark C.3).

### B.3  A Fully Single-Loop First-Order Algorithm

A drawback of double-loop implementation is that we have to wait for an increasingly large number of samples (since we design $T_k$ or $M_k$ to be increased in $k$) to be collected before we can improve the objective. A natural question is whether we can keep incrementally updating upper-level variables $x$ without waiting for too many inner iterations or for the evaluation of large batches. If the stochastic oracle satisfies the mean-squared smoothness assumption and allow two points to be queried simultaneously, then we can implement the algorithm in single-loop (that replace the inner loops with a single step, and avoid the use of large $\text{poly}(\epsilon^{-1})$ batches):

**Assumption 11** *Stochastic oracles allow 2-simultaneous query: the algorithm can observe unbiased estimators of $\nabla f(x,y), \nabla g(x,y)$ at two different points $(x_1, y_1), (x_2, y_2)$ for a shared random seed $\zeta$ and $\xi$. Furthermore, gradient estimators satisfy the mean-squared smoothness condition:*

$$\mathbb{E}[\|\nabla f(x_1, y_1; \zeta) - \nabla f(x_2, y_2; \zeta)\|^2] \le l_{f,1}^2(\|x_1 - x_2\|^2 + \|y_1 - y_2\|^2),$$
$$\mathbb{E}[\|\nabla g(x_1, y_1; \xi) - \nabla g(x_2, y_2; \xi)\|^2] \le l_{g,1}^2(\|x_1 - x_2\|^2 + \|y_1 - y_2\|^2).$$

We define momentum-assisted gradient estimators recursively for the inner loop proximal-solvers as follows:

$$\widetilde{g}_{wz}^k := \nabla_y g(x_k, w_{z,k}; \xi_{wz}^k) + (1-\eta_k)\left(\widetilde{g}_{wz}^{k-1} - \nabla_y g(x_{k-1}, w_{z,k-1}; \xi_{wz}^k)\right),$$
$$\widetilde{f}_{wy}^k := \nabla_y f(x_k, w_{y,k}; \zeta_{wy}^k) + (1-\eta_k)\left(\widetilde{f}_{wy}^{k-1} - \nabla_y f(x_{k-1}, w_{y,k-1}; \zeta_{wy}^k)\right),$$
$$\widetilde{g}_{wy}^k := \nabla_y g(x_k, w_{y,k}; \xi_{wy}^k) + (1-\eta_k)\left(\widetilde{g}_{wy}^{k-1} - \nabla_y g(x_{k-1}, w_{y,k-1}; \xi_{wy}^k)\right),$$

where $\eta_k \in (0, 1]$, and $\eta_0 = 1$ (and thus, ignores $(k-1)^{th}$ terms at $k = 0$). Formulas for the update to upper-level variables $x$ are defined similarly. A single-loop alternative to Algorithm 1 can be defined as in Algorithm 2. Our analysis shows that the momentum-assisted technique leads to improvement in sample-complexity upper bounds.

## APPENDIX C   ANALYSIS OVERVIEW

In this section, we provide our main convergence results for Algorithm 1 and Algorithm 2.

### C.1   ANALYSIS OF ALGORITHM 1

We first define the proximal error of $y$ and $z$ at the $k^{th}$ iteration as:

$$\Delta_k^y := \rho^{-1} \cdot (y_k - \mathbf{prox}_{\rho h_{\sigma_k}(x_k, \cdot)}(y_k)), \quad \Delta_k^z := \rho^{-1} \cdot (z_k - \mathbf{prox}_{\rho g(x_k, \cdot)}(z_k)).$$

For measuring the error in $x$, we define [2]

$$\hat{x}_k := \Pi_{\mathcal{X}} \left\{ x_k - \alpha_k \left( \nabla_x h_{\sigma_k}(x_k, \mathbf{prox}_{\rho h_{\sigma_k}(x_k, \cdot)}(y_k)) - \nabla_x g(x_k, \mathbf{prox}_{\rho g(x_k, \cdot)}(z_k)) \right) \right\},$$
$$\Delta_k^x := \alpha_k^{-1}(x_k - \hat{x}_k).$$

Next, we define

$$\Phi_{\sigma,\rho}(x, y, z) := \frac{h_{\sigma,\rho}^*(x, y) - g_\rho^*(x, z)}{\sigma} + \frac{C}{\sigma}(g_\rho^*(x, z) - g^*(x)), \tag{14}$$

with some universal constant $C \geq 4$, and finallly we define the potential function as

$$\mathbb{V}_k := \Phi_{\sigma_k, \rho}(x_k, y_k, z_k) + \frac{C_w \lambda_k}{\sigma_k \rho} \left( \|w_{y,k} - \mathbf{prox}_{\rho h_{\sigma_k}(x_k, \cdot)}(y_k)\|^2 + \|w_{z,k} - \mathbf{prox}_{\rho g(x_k, \cdot)}(z_k)\|^2 \right), \tag{15}$$

where $C_w > 0$ is some sufficiently large universal constant, and $\lambda_k := T_k \gamma_k / (4\rho)$ is a target improvement rate for chasing proximal operators per outer-iteration. We are now ready to state our main convergence theorem.

**Theorem C.1** *Suppose that Assumptions 1-4 and 9-10 hold, with parameters and stepsizes satisfying the following bounds, for all $k \geq 0$:*

$$\rho < c_2/l_{g,1}, \quad \sigma_k < c_1 l_{g,1}/l_{f,1}, \quad T_k \gamma_k < c_3 \rho, \quad \beta_k \leq c_4 \ll 1, \quad \alpha_k \leq c_5 \rho(1 + l_{g,1}/\mu)^{-1},$$
$$\alpha_k \leq c_6 \rho^3 \min(\mu^2, \delta^2/D_{\mathcal{Y}}^2) \cdot \beta_k, \quad \frac{\sigma_k - \sigma_{k+1}}{\sigma_{k+1}} \leq c_7 \rho^2 \min(\mu^2, \delta^2/D_{\mathcal{Y}}^2) \cdot \beta_k, \tag{16}$$

*with some universal constants $c_1, c_2, c_3, c_4, c_5, c_6, c_7 > 0$ as well as the following:*

$$\rho \beta_k + \alpha_k \leq c_8 T_k^2 \gamma_k^2, \quad \forall k, \tag{17}$$

*with some universal constant $c_8 > 0$. Then the iterates of Algorithm 1 satisfy*

$$\mathbb{E} \left[ \sum_{k=0}^{K-1} \frac{\alpha_k}{16\sigma_k} \|\Delta_k^x\|^2 + \frac{\rho \beta_k}{16\sigma_k} (\|\Delta_k^y\|^2 + \|\Delta_k^z\|^2) \right] \leq \mathbb{E}[\mathbb{V}_0 - \mathbb{V}_K] \tag{18}$$

$$+ O(C_f) \cdot \sum_{k=0}^{K-1} \left( \frac{\sigma_k - \sigma_{k+1}}{\sigma_{k+1}} \right) + \frac{O(l_{g,1}/\mu + C_w)}{\rho} \left( \sum_{k=0}^{K-1} \sigma_k^{-1} \left( \frac{\alpha_k}{M_k} + \rho^{-1} T_k^2 \gamma_k^3 \right) (\sigma_k^2 \sigma_f^2 + \sigma_g^2) \right).$$

---

[2]Note that $\Delta_k^x$ is a stricter stationarity measure on $x$ than Definition 8 as long as $\alpha_k \leq \rho$, since the function $g : [0, \infty) \to \mathbb{R}$ defined by $g(s) := \|x - \Pi_{\mathcal{X}} \{x + sw\}\|/s$ with any $w \in \mathbb{R}^{d_x}$ is monotonically nonincreasing (see *e.g.,* Lemma 2.3.1 in Bertsekas (1999))

The proof of Theorem C.1 is given in Appendix F. We mention here that problem-dependent constants may not be fully optimized and could be improved with more careful analysis. Still, there are two major considerations for the stepsizes: (i) the relations between (i) $\beta_k$ and $\alpha_k$ and (ii) the relations between $\alpha_k$ (or $\rho\beta_k$) and $T_k\gamma_k$. Regarding (i), the conditions (16) require $\alpha_k/\beta_k \asymp \rho^2 \min(\mu, \delta/D_\mathcal{Y})^2$. Effectively, this relation determines the number of updates of the $y_k$ and $z_k$ variables for each update of $x_k$. The condition is necessary to ensure that $y_k$ and $z_k$ always remain relatively close to the solution-set $T(x_k, \sigma_k)$ and $T(x_k, 0)$ *in expectation*, which is crucial to convergence to a stationary point of the saddle-point problem (13). Regarding (ii), the relation between $\alpha_k$ and $T_k\gamma_k$ in (17) is required for approximately evaluating the proximal operators without solving from scratch at every outer iteration.

As a corollary, with proper design of step-sizes, we can give a finite-time convergence guarantee for reaching an approximate stationary point of $\psi_\sigma(x)$. To simplify the statement, we treat all problem-dependent parameters as $O(1)$ quantities.

**Corollary C.2** *Let $\alpha_k = c_\alpha\rho(k+k_0)^{-a}$, $\beta_k = c_\beta(k+k_0)^{-b}$, $\gamma_k = c_\gamma(k+k_0)^{-c}$, and $\sigma_k = c_\sigma(k+k_0)^{-s}$, $T_k = (k+k_0)^t$, $M_k = (k+k_0)^m$ with some proper problem-dependent constants $c_\alpha, c_\beta, c_\gamma, c_\sigma,$ and $k_0$. Let $R$ be a random variable drawn from a uniform distribution over $\{0, ..., K-1\}$, and let $\epsilon = \sigma_K$. Under the same conditions in Theorem C.1, the following holds after $K$ iterations of Algorithm 1: for the optimal design of rates, we set $a = b = 0, s = 1/3$, and*

   *(a) if stochastic noises are present in both upper-level objective $f$ and lower-level objective $g$ (i.e., $\sigma_f^2, \sigma_g^2 > 0$), then let $c = t = m = 4/3$.*

   *(b) If stochastic noises are present only in $f$ (i.e., $\sigma_f^2 > 0$, $\sigma_g^2 = 0$), then let $c = t = m = 2/3$.*

   *(c) If we have access to exact information about $f$ and $g$ (i.e., $\sigma_f^2 = \sigma_g^2 = 0$), then let $c = t = m = 0$.*

*Then, we have $\|\nabla\psi_\epsilon(x_R)\| \asymp \frac{\log K}{K^{1/3}}$ with probability at least $2/3$. If Assumption 7 and 8 additionally hold at $x_R$, then we also have $\|\nabla\psi(x_R)\| \asymp \frac{\log K}{K^{1/3}}$.*

Note that the overall gradient oracle complexity (or simply sample complexity) to have $\mathbb{E}[\|\nabla\psi_\epsilon(x_R)\|] = O(\epsilon)$ is given by $O(K \cdot (M_K + T_K))$ with $K = O(\epsilon^{-1/s})$ and $M_K = T_K = O(\epsilon^{t/s})$. Thus, we have $O(\epsilon^{-7})$, $O(\epsilon^{-5})$, and $O(\epsilon^{-3})$ sample-complexity upper-bounds for fully-stochastic, only upper-level stochastic, and deterministic cases respectively.

**Remark C.3 (Single-Loop Implementation with Algorithm 1)** *While we design $T_K = M_K = O(\epsilon^{-4})$ to achieve the best complexity bound in stochastic scenarios, we can also find different rate scheduling for which $T_K = M_K = O(1)$. For instance, when $\mathcal{X} = \mathbb{R}^{d_x}$, we can change the coefficients of noise-variance terms from $O(\alpha_k/M_k)$ to $O(\alpha_k^2)$, and schedule the rates of step-sizes such that left-hand side of (18) converges. However, we found that such a single-loop design may result in overall worse complexity bounds unless momentum-assistance techniques are deployed.*

## C.2 ANALYSIS OF ALGORITHM 2

In addition to quantities defined before, we also should track the noise-variance terms in momentum-assisted gradient estimators. We first define the expected gradients $G_{wy}^k, G_{wz}^k, G_x^k$ as follows:

$$G_{wz}^k := \nabla_y g(x_k, w_{z,k}), \quad G_{wy}^k := \sigma_k \nabla_y f(x_k, w_{y,k}) + \nabla_y g(x_k, w_{y,k}),$$
$$G_x^k := \sigma_k \nabla_x f(x_k, w_{y,k+1}) + \nabla_x g(x_k, w_{y,k+1}) - \nabla_x g(x_k, w_{z,k+1}).$$

Next, we define error terms $e_{wz}^k, e_{wy}^k, e_x^k$ in these gradient estimators as follows:

$$e_{wz}^k := \widetilde{g}_{wz}^k - G_{wz}^k, \quad e_{wy}^k := \sigma_k\widetilde{f}_{wy}^k + \widetilde{g}_{wy}^k - G_{wy}^k, \quad e_x^k := \sigma_k\widetilde{f}_x^k + (\widetilde{g}_{xy}^k - \widetilde{g}_{xz}^k) - G_x^k.$$

Finally, we we redefine the potential function:

$$\mathbb{V}_k := \Phi_{\sigma_k,\rho}(x_k, y_k, z_k) + \frac{C_w}{\sigma_k\rho}\left(\|w_{y,k} - \mathbf{prox}_{\rho h_{\sigma_k}(x_k,\cdot)}(y_k)\|^2 + \|w_{z,k} - \mathbf{prox}_{\rho g(x_k,\cdot)}(z_k)\|^2\right)$$

$$+ \frac{C_\eta \rho^2}{\sigma_k \gamma_{k-1}} \left( \|e_x^{k-1}\|^2 + \|e_{wy}^k\|^2 + \|e_{wz}^k\|^2 \right), \tag{19}$$

with some properly set universal constants $C_w, C_\eta > 0$. For technical reasons, we require here one additional assumption on the boundedness of the movement in $w_{y,k}$.

**Assumption 12** *For all $x \in \mathcal{X}$ and $y, z \in \mathcal{Y}$, let $w_y^* := \mathbf{prox}_{\rho h_\sigma(x,\cdot)}(y) = \arg\min_{w \in \mathcal{Y}} h_\sigma(x,y,w)$ and $w_z^* := \mathbf{prox}_{\rho g(x,\cdot)}(z) = \arg\min_{w \in \mathcal{Y}} g(x,z,w)$ where $h_\sigma(x,y,w)$ and $g(x,z,w)$ are defined in* (12). *We assume that*

$$\|\nabla_w h_\sigma(x,y,w_y^*)\| \le M_w, \quad \|\nabla_w g(x,z,w_z^*)\| \le M_w,$$

*for some (problem-dependent) constant $M_w = O(1)$.*

We are now ready to state the convergence guarantee for the momentum-assisted fully-single loop implementation.

**Theorem C.4** *Suppose that Assumptions 1-4, 9-12 hold, with parameters and step-sizes satisfying*(16) *as well as the following relations for all $k \ge 0$:*

$$\rho \beta_k + \alpha_k \le c_8 \gamma_k, \quad \eta_{k+1} \ge c_9 \rho^{-2} \cdot \max\left( (l_{g,1}/\mu) \alpha_k \gamma_k, \gamma_k^2 \right). \tag{20}$$

*Then the iterates of Algorithm 2 satisfy the following inequality:*

$$
\mathbb{E}\left[ \sum_{k=0}^{K-1} \frac{\alpha_k}{16\sigma_k} \|\Delta_k^x\|^2 + \frac{\beta_k}{16\sigma_k \rho} \left( \|y_k - w_{y,k}^*\|^2 + \|z_k - w_{z,k}^*\|^2 \right) \right] \le \mathbb{E}[\mathbb{V}_0 - \mathbb{V}_K]
$$
$$
+ \sum_{k=0}^{K-1} \left( \left( \frac{\sigma_k - \sigma_{k+1}}{\sigma_k} \right) \cdot O(C_f) + \frac{O(M_w^2)\rho^2}{C_w \sigma_k \gamma_k} \left( \frac{\sigma_k - \sigma_{k+1}}{\sigma_{k+1}} \right)^2 + C_\eta \rho^2 \frac{O(\eta_{k+1}^2)}{\sigma_k \gamma_k} (\sigma_k^2 \sigma_f^2 + \sigma_g^2) \right)
$$
$$
+ C_\eta O(\rho^2 l_{g,1}^2) \left( \frac{h_{\sigma_0}(x_0, y_0, w_{y,0}) - h_{\sigma_0,\rho}^*(x_0, y_0)}{\sigma_0} + \frac{g(x_0, z_0, w_{z,0}) - g_\rho^*(x_0, z_0)}{\sigma_0} \right).
$$

We then give a corollary analogous to Corollary C.2, with proper design of step-sizes. As before, to simplify the statement, we treat all problem-dependent parameters as $O(1)$ quantities.

**Corollary C.5** *Let $\alpha_k = c_\alpha \rho(k+k_0)^{-a}$, $\beta_k = c_\beta(k+k_0)^{-b}$, $\gamma_k = c_\gamma(k+k_0)^{-c}$, $\sigma_k = c_\sigma(k+k_0)^{-s}$ and $\eta_k = (k+k_0)^{-n}$ with some proper problem-dependent constants $c_\alpha, c_\beta, c_\gamma, c_\sigma$, and $k_0$. Let $R$ be a random variable drawn from a uniform distribution over $\{0, ..., K-1\}$, and let $\epsilon = \sigma_K$. Under the same conditions in Theorem C.4, the following claims hold after $K$ iterations of Algorithm 2.*

(a) *If stochastic noise is present in both upper-level objective $f$ and lower-level objective $g$ (i.e., $\sigma_f^2, \sigma_g^2 > 0$), then let $a = b = c = 2/5$, $s = 1/5$, and $n = 4/5$. Then $\|\nabla \psi_\epsilon(x_R)\| \asymp \frac{\log K}{K^{1/5}}$ with probability at least $2/3$.*

(b) *If stochastic noises are present only in $f$, let $a = b = c = 1/4$, $s = 1/4$, and $n = 1/2$. Then $\|\nabla \psi_\epsilon(x_R)\| \asymp \frac{\log K}{K^{1/4}}$ with probability at least $2/3$.*

(c) *If we have access to exact gradient information, let $a = b = c = 0$, $s = 1/3$, $n = 0$. Then $\|\nabla \psi_\epsilon(x_R)\| \asymp \frac{\log K}{K^{1/3}}$ with probability at least $2/3$.*

*If Assumption 7 and 8 additionally hold at $x_R$, then the same conclusion holds for $\|\nabla \psi(x_R)\|$.*

Note that since Algorithm 2 only uses $O(1)$ samples per iteration, the overall sample-complexity is upper-bounded by $O(\epsilon^{-5})$, $O(\epsilon^{-4})$, and $O(\epsilon^{-3})$ for fully-stochastic, only upper-level stochastic, and deterministic cases respectively. That is, momentum assistance not only enables single-loop implementation, but also improves the overall sample complexity.

## Appendix D  Auxiliary Lemmas

Throughout the section, we take $\rho \leq c_1/l_{g,1}$ and $\sigma < c_2 l_{g,1}/l_{f,1}$ with sufficiently small universal constants $c_1, c_2 \in (0, 0.01]$. We also assume that Assumptions 2-4 hold by default.

**Theorem D.1 (Danskin's Theorem)** *Let $f(w, \theta)$ be a continuously differentiable and smooth function on $\mathcal{W} \times \Theta$. Let $l^*(w) := \min_{\theta \in \Theta} f(w, \theta)$ and $S(w) := \arg\min_{\theta \in \Theta} f(w, \theta)$, and assume $S(w)$ is compact for all $w$. Then the directional derivative of $l^*(w)$ in direction $v$ with $\|v\| = 1$ is given by:*

$$D_v l^*(w) := \lim_{\delta \to 0} \frac{l^*(w + \delta v) - l^*(w)}{\delta} = \min_{\theta \in S(w)} \langle v, \nabla_w f(w, \theta) \rangle. \tag{21}$$

**Lemma D.2 (Proposition 5 in Shen & Chen (2023))** *For a continuously-differentiable and $L$-smooth function $f(w, \theta)$ in $\mathcal{W} \times \Theta$, consider a minimizer function $l^*(w) = \min_{\theta \in \Theta} f(w, \theta)$ and a solution map $S(w) = \arg\min_{\theta \in \Theta} f(w, \theta)$. If $S(w)$ is $L_S$-Lipschitz continuous at $w$, then $l^*(w)$ is differentiable and $L(1 + L_S)$-smooth at $w$, and $\nabla l^*(w) = \nabla_w f(w, \theta^*)$ for any $\theta^* \in S(w)$.*

**Lemma D.3** *For any $x_1, x_2 \in \mathcal{X}$, and $y_1, y_2 \in \mathcal{Y}$, the following holds:*

$$\|\mathbf{prox}_{\rho g(x_1, \cdot)}(y_1) - \mathbf{prox}_{\rho g(x_2, \cdot)}(y_2)\| \leq O(\rho l_{g,1}) \|x_1 - x_2\| + \|y_1 - y_2\|.$$

*The same property holds with $h_\sigma(x, \cdot)$ instead of $g(x, \cdot)$.*

**Lemma D.4** *For any $x \in \mathcal{X}$, $y \in \mathcal{Y}$ and $\sigma_1, \sigma_2 \in [0, \sigma]$, the following holds:*

$$\|\mathbf{prox}_{\rho h_{\sigma_1}(x, \cdot)}(y) - \mathbf{prox}_{\rho h_{\sigma_2}(x, \cdot)}(y)\| \leq O(\rho l_{f,0}) |\sigma_1 - \sigma_2|.$$

**Lemma D.5** *For the choice of $\rho < 1/(4l_{g,1})$ and $\sigma < c \cdot l_{g,1}/l_{f,1}$ with sufficiently small $c > 0$, $h^*_{\sigma,\rho}(x, y)$ is continuously differentiable and $2\rho^{-1}$-smooth jointly in $(x, y)$.*

## Appendix E  Deferred Proofs in Section 3

### E.1  Special Case: Unique Solution and Invertible Hessian

When the Hessian is invertible at the unique solution, the statement can be made stronger since we can deduce solution-set Lipschitz continuity from the well-understood solution sensitivity analysis in constrained optimization Bonnans & Shapiro (2013). That is, we can provide a strong sufficiency guarantee for the Lipschitz continuity of solution-sets.

**Proposition E.1 (Sufficient Condition for Lipschitz Continuity)** *Suppose $y^* \in T(x, \sigma)$ is the unique lower-level solution at $(x, \sigma)$. Suppose that Assumption 6 holds at $y^*$ with corresponding Lagrangian multipliers $\lambda^*, \nu^*$. Further, suppose that $\nabla^2 \mathcal{L}^*_\mathcal{I}$ is invertible. Then $T(x, \sigma)$ is locally Lipschitz continuous at $(x, \sigma)$.*

Thus, the uniqueness of the solution along with LICQ, strict complementarity, and invertibility of the Hessian is strong enough to guarantee the Lipschitz continuity of solution sets. Therefore, we can conclude that $\frac{\partial^2}{\partial x \partial \sigma} l(x, \sigma)$ exists and that the order of differentiation commutes under Assumption 6.

**Theorem E.2** *Suppose $y^* \in T(x, \sigma)$ is the unique lower-level solution at $(x, \sigma)$. If Assumption 6 holds at $y^*$, and $\nabla^2 \mathcal{L}^*_\mathcal{I}$ is invertible, then $\frac{\partial^2}{\partial x \partial \sigma} l(x, \sigma)$ exists and can be given explicitly by*

$$\frac{\partial^2}{\partial \sigma \partial x} l(x, \sigma) = \frac{\partial^2}{\partial x \partial \sigma} l(x, \sigma)$$
$$= \nabla_x f(x, y^*) - \begin{bmatrix} 0 & \nabla^2_{xy} h_\sigma(x, y^*) \end{bmatrix} (\nabla^2 \mathcal{L}_\mathcal{I}(\lambda^*_\mathcal{I}, \nu^*, y^* | x, \sigma))^{-1} \begin{bmatrix} 0 \\ \nabla_y f(x, y^*) \end{bmatrix}.$$

*If this equality holds at $\sigma = 0^+$, then $\psi(x)$ is differentiable at $x$, and $\lim_{\sigma \to 0} \nabla \psi_\sigma(x) = \nabla \psi(x)$.*

Below, we first provide the proofs of Proposition E.1 and Theorem E.2.

### E.2 Proof of Proposition E.1

The proof is based on the celebrated implicit function theorem. See Appendix C.7 in Evans (2022), for instance. We first show that if $y^*(x, \sigma) \in T(x, \sigma)$ is unique, then there exists $\delta > 0$ and $\delta_y > 0$ such that for all $\|(x', \sigma') - (x, \sigma)\| < \delta$, the solution satisfies $T(x', \sigma') \subset \mathbb{B}(y^*, \delta_y)$ and is singleton where $\mathbb{B}(y^*, \delta_y)$ is an open ball of radius $\delta_y$ centered at $y^*$. When the context is clear, we simply denote $y^*(x, \sigma)$ as $y^*$.

To begin with, we first argue that we can take $\delta$ small enough such that solutions cannot happen outside the neighborhood of $y^*$. Note that unions of all solution sets are contained in $\mathbb{B}(0, R)$ with some finite $R < \infty$ due to Assumption 4. For any $\delta_y > 0$, let $q^* = \min_{y \in (\mathcal{Y} \cup \mathbb{B}(0,R))/\mathbb{B}(y^*, \delta_y)} \sigma f(x, y) + g(x, y)$, and let $M = \max_{y \in \mathbb{B}(0,R)}(\|\sigma \nabla_x f(x, y) + \nabla_x g(x, y)\| + f(x, y))$ (the finite maximum exists since $f, g$ are smooth). Since $T(x, \sigma)$ is singleton, we have $q^* > l(x, \sigma)$. Thus, there exists $0 < \delta_0 \ll (q^* - l(x, \sigma))/M$, such that for all $(x', \sigma') \in \mathbb{B}((x, \sigma), \delta_0)$, we have $T(x', \sigma') \subseteq \mathbb{B}(y, \delta_y)$.

Next, by regularity and strict complementary slackness of $y^*$, there exists a unique $(\lambda^*, \nu^*)$ such that $\lambda_i^* > 0$ for all $i \in \mathcal{I}(y^*)$, $\lambda_i^* = 0$ for all $i \notin \mathcal{I}(y^*)$, and $\nabla \mathcal{L}(\lambda_\mathcal{I}^*, \nu^*, y^* | x, \sigma) = 0$. Since we assumed $\nabla^2 \mathcal{L}(\lambda_\mathcal{I}^*, \nu^*, y^* | x, \sigma)$ being invertible, we can apply implicit function theorem. That is, there exists an sufficiently small $\delta > 0$ such that for all $(x', \sigma')$: $|(x', \sigma') - (x, \sigma)| < \delta$, we can take $\delta_{\lambda, \nu}, \delta_y > 0$ such that there is a unique $(\lambda_\mathcal{I}', \nu', y')$

$$\nabla \mathcal{L}_\mathcal{I}(\lambda_\mathcal{I}', \nu', y' | x', \sigma') = 0,$$

inside the local region $\|(\lambda_\mathcal{I}', \nu') - (\lambda^*, \nu^*)\| < \delta_{\lambda, \nu}$ and $\|y' - y\| < \delta_y$. Thus, we can take $\delta > 0$ sufficiently small such that $\delta_{\lambda, \nu}$ can be sufficiently small to keep $\lambda_\mathcal{I}^*$ non-negative.

Furthermore, in this local region, $T(x', \sigma') \subseteq \mathbb{B}(y, \delta_y)$ for all $(x', \sigma') \in \mathbb{B}((x, \sigma), \delta')$ where $\delta' = \min(\delta_0, \delta)$, which in turn implies that $T(x', \sigma')$ is a singleton and uniquely given by the implicit function theorem. Therefore, $T(x, \sigma)$ is differentiable and thus locally Lipschitz continuous. In addition, $T(x, \sigma)$ is always singleton over $\mathbb{B}((x, \sigma), \delta')$. $\qquad\square$

### E.3 Proof of Theorem E.2

Recall the local region given in the proof of Proposition E.1. We note that the implicit function theorem further says that in this local region, we can define differentiation of $y$ with respect to $x$ and $\sigma$ such that

$$\frac{dy^*(x, \sigma)}{d\sigma} = -\begin{bmatrix} 0 & I \end{bmatrix} \nabla^2 \mathcal{L}_\mathcal{I}(\lambda_\mathcal{I}^*, \nu^*, y^* | x, \sigma)^{-1} \begin{bmatrix} 0 \\ \nabla_y f(x, y^*) \end{bmatrix},$$

$$\nabla_x y^*(x, \sigma) = -\begin{bmatrix} 0 & I \end{bmatrix} \nabla^2 \mathcal{L}_\mathcal{I}(\lambda_\mathcal{I}^*, \nu^*, y^* | x, \sigma)^{-1} \begin{bmatrix} 0 \\ \nabla_{yx}^2 h_\sigma(x, y^*) \end{bmatrix}.$$

As a consequence, $\frac{\partial^2}{\partial \sigma \partial x} l(x, \sigma)$ is given by

$$\frac{\partial^2}{\partial \sigma \partial x} l(x, \sigma) = \frac{\partial}{\partial \sigma}(\sigma \nabla_x f(x, y^*(x, \sigma)) + \nabla_x g(x, y^*(x, \sigma)))$$

$$= \nabla_x f(x, y^*(x, \sigma)) + \sigma \nabla_{xy}^2 f(x, y^*(x, \sigma)) \frac{dy^*(x, \sigma)}{d\sigma} + \nabla_{xy}^2 g(x, y^*(x, \sigma)) \frac{dy^*(x, \sigma)}{d\sigma}$$

$$= \nabla_x f(x, y^*(x, \sigma)) - \begin{bmatrix} 0 & \nabla_{xy}^2 h_\sigma(x, y^*(x, \sigma)) \end{bmatrix} (\nabla^2 \mathcal{L}|_\mathcal{I}^*)^{-1} \begin{bmatrix} 0 \\ \nabla_y f(x, y^*(x, \sigma)) \end{bmatrix}.$$

Similarly, differentiation in swapped order is also given by

$$\frac{\partial^2}{\partial x \partial \sigma} l(x, \sigma) = \frac{\partial}{\partial x} f(x, y^*(x, \sigma))$$

$$= \nabla_x f(x, y^*(x, \sigma)) + \nabla_x y^*(x, \sigma))^\top \nabla_y f(x, y^*(x, \sigma))$$

$$= \nabla_x f(x, y^*(x, \sigma)) - \begin{bmatrix} 0 & \nabla_{xy}^2 h_\sigma(x, y^*(x, \sigma)) \end{bmatrix} (\nabla^2 \mathcal{L}_\mathcal{I}^*)^{-1} \begin{bmatrix} 0 \\ \nabla_y f(x, y^*(x, \sigma)) \end{bmatrix}.$$

Hence $\frac{\partial^2}{\partial x \partial \sigma} l(x, \sigma)$ exists. If this holds at $\sigma = 0^+$, then we have $\lim_{\sigma \to 0} \psi_\sigma(x) = \psi(x)$. $\qquad\square$

### E.4 Proof of Proposition 3.4

We instead prove the general version of Proposition 3.4:

**Theorem E.3** *Suppose that $f(w, \theta)$ in $\mathcal{W} \times \Theta$ is continuously-differentiable and $L$-smooth with $\mathcal{W}, \Theta$ satisfying Assumption 4. Let $l^*(w) = \min_{\theta \in \Theta} f(w, \theta)$. Assume the solution map $S(w) = \arg\min_{\theta \in \Theta} f(w, \theta)$ is locally Lipschitz continuous at $w$. For any $\theta^* \in S(w)$ such that (1) $\theta^*$ satisfies Definition 5 and 6 with Lagrangian multipliers $\lambda^*$, and (2) $\nabla^2 \mathcal{L}|_{\mathcal{I}}^*$ is locally continuous at $(w, \lambda^*, \theta^*)$ jointly in $(w, \lambda, \theta)$. Then the following must hold:*

$$\forall v \in \mathbf{Im}(\nabla^2_{\theta w} f(w, \theta^*)) : \begin{bmatrix} 0 \\ v \end{bmatrix} \in \mathbf{Im}(\nabla^2 \mathcal{L}(\lambda^*, \theta^* | w)).$$

Then Proposition 3.4 follows as a corollary.

*Proof.* We show this by contradiction. For simplicity, we assume that Let $\{g_i\}_{i \in \mathcal{I}(\theta^*)}$ be a set of active constraints of $\Theta$ at $\theta^*$. To simplify the discussion, we assume no equality constraints (there will be no change in the argument). Suppose there exists $v \in \mathbf{Im}(\nabla^2_{\theta w} f(w, \theta^*))$ such that $\|v\| = 1$ and $(0, v)$ is not in the image of Lagrangian Hessian. Let $(0, v) = v_{\mathbf{Ker}} + v_{\mathbf{Im}}$ be the orthogonal decomposition of $v$ into kernel and image of the Hessian of Lagrangian. Note that we can take $v$ such that $\|v_{\mathbf{Ker}}\| > 0$. Let $(dx, d\sigma)$ be such that $\Omega(\delta) \cdot v = \nabla^2_{\theta w} f(w, \theta^*) dw$.

Since $S(w)$ is locally Lipschitz continuous, there exists $\delta > 0$ and $L_T < \infty$ such that for all $\|dw\| < \delta$, there exists $\|d\theta\| < L_T \delta$ such that $\theta^* + d\theta \in S(w + dw)$. We can take $\delta$ small enough such that inactive inequality constraints stay inactive with $d\theta$ change. Thus, when considering $d\lambda$, we do not change coordinates that correspond to inactive constraints.

We claim that there cannot exist $(d\lambda, d\theta)$ with $\|d\theta\| < L_T \delta$ that can satisfy

$$\nabla \mathcal{L}(\lambda^* + d\lambda, \theta^* + d\theta | w + dw) = 0.$$

First, we show that $d\lambda$ cannot be too large. Note that

$$\nabla \mathcal{L}(\lambda^* + d\lambda, \theta^* | w) - \nabla \mathcal{L}(\lambda^*, \theta^* | w) = \sum_{i \in \mathcal{I}(\theta^*)} (d\lambda_i) \nabla g_i(\theta^*)$$
$$= \underbrace{[\nabla g_i(\theta^*), \ i \in \mathcal{I}(\theta^*)]}_{B} [d\lambda_i, \ i \in \mathcal{I}(\theta^*)]$$

Since we assumed that $B$ is full-rank in columns, the minimum (right) singular value $s_{\min}$ of $B$ is strictly positive, *i.e.,* $s_{\min} > 0$. On the other hand, by Lipschitz-continuity of all gradients, perturbations in $w, \theta$ can change gradients of Lagrangian only by order $O(\delta)$:

$$\nabla \mathcal{L}(\lambda^* + d\lambda, \theta^* + d\theta | w + dw) - \nabla \mathcal{L}(\lambda^* + d\lambda, \theta^* | w) \leq \underbrace{L \|dw\|}_{\text{perturbed by } dw} + \underbrace{\sum_{i \in \mathcal{I}(\theta^*)} (\lambda_i^* + d\lambda_i) L \|d\theta\|}_{\text{perturbed by } d\theta}.$$

Thus, since $(dw, d\theta) = O(\delta)$, we have

$$(s_{\min} - O(\delta)) \|d\lambda\| + O(\delta)(1 + \|\lambda^*\|) = 0. \tag{22}$$

By taking $\delta < s_{\min}$ small enough, and due to the existence of Lagrange multipliers $\|\lambda^*\| < \infty$, we have proven that $\|d\lambda\| = O(\delta)$ with sufficiently small $\delta$.

Next, we check that

$$\nabla \mathcal{L}(\lambda^* + d\lambda, \theta^* + d\theta | w + dw, \Theta) - \nabla \mathcal{L}(\lambda^*, \theta^* | w)$$
$$= \Omega(\delta) \cdot (v_{\mathbf{Ker}} + v_{\mathbf{Im}}) + \nabla^2 \mathcal{L}(\lambda^*, y^* | w) \begin{bmatrix} d\lambda \\ d\theta \end{bmatrix} + o(\delta).$$

However, $v_{\mathbf{Ker}}$ is not in the image of $\nabla^2 \mathcal{L}$, and $o(\delta)$ terms cannot eliminate $\Omega(\delta) \cdot v_{\mathbf{Ker}}$ if $\delta \ll \|v_{\mathbf{Ker}}\|$. Thus, $\nabla \mathcal{L}(\lambda^* + d\lambda, \theta^* + d\theta | w + dw, \Theta)$ cannot be 0, which implies there is no feasible optimal solution in $\delta$-ball around $\theta^*$ if we perturb $w$ in direction $dw$. This contradicts $S(w)$ being locally Lipschitz continuous. Thus, (10) is necessary for $S(w)$ to be locally Lipschitz continuous. $\quad\square$

### E.5 Proof of Theorem 3.1

Similarly to the proof of Proposition 3.4, we prove the following general version:

**Theorem E.4** *Suppose that $f(w, \theta)$ in $\mathcal{W} \times \Theta$ is continuously-differentiable and $L$-smooth with $\mathcal{W}, \Theta$ satisfying Assumption 4. Let $l^*(w) = \min_{\theta \in \Theta} f(w, \theta)$. Assume the solution map $S(w) = \arg\min_{\theta \in \Theta} f(w, \theta)$ is locally (uniformly) Lipschitz continuous at all neighborhoods of $w$. For $w \in \mathcal{W}$, if there exists at least one $\theta^* \in S(w)$ such that (1) $\theta^*$ satisfies Definition 5 and 6 with Lagrangian multipliers $\lambda^*$, and (2) $\nabla^2 f, \nabla^2 \mathcal{L}_{\mathcal{I}}^*$ is locally continuous at $(w, \lambda_{\mathcal{I}}^*, \theta^*)$ jointly in $(w, \lambda_{\mathcal{I}}, \theta)$, then $\nabla^2 l^*(w)$ exists and is given by*

$$\nabla^2 l^*(w) = \nabla^2_{ww} f(w, \theta^*) - \begin{bmatrix} 0 & \nabla^2_{w\theta} f(w, \theta^*) \end{bmatrix} (\nabla^2 \mathcal{L}_{\mathcal{I}}(\lambda_{\mathcal{I}}^*, \theta^* | w))^\dagger \begin{bmatrix} 0 \\ \nabla^2_{\theta w} f(w, \theta^*) \end{bmatrix}. \quad (23)$$

Theorem 3.1 follows as a corollary by only taking $\frac{\partial^2}{\partial x \partial \sigma}$ and $\frac{\partial^2}{\partial \sigma \partial x}$ parts with $w = (\sigma, x)$.

*Proof.* Let $\theta_t^* \in S(w + tv)$ be the closest solution to $\theta^* \in S(w)$, and let $\lambda_t^*$ be the corresponding Lagrangian multiplier. Let $\mathcal{I} := \mathcal{I}(\theta^*)$ be a set of active constraints of $\Theta$ at $\theta^*$. As in the proof of Theorem E.3, to simplify the discussion, we assume there is no equality constraints (including equality constraints needs only a straightforward modification). We first show that the active constraints $\mathcal{I}$ does not change due to the perturbation $tv$ in $w$ when the solution set is Lipschitz continuous. To see this, note that all inactive inequality constraints remain strictly negative $g_i(\theta) < 0$ for all $i \neq \mathcal{I}(\theta^*)$. For active constraints, due to Definition 6, we have $\lambda_i^* > 0$ for all $g_i(\theta^*) = 0$ with $i \in \mathcal{I}$. By the solution-set continuity given as assumption, we have $\|\theta_t^* - \theta^*\| = O(t)$. Thus, by the same argument as deriving (22), we have $\|\lambda_t^* - \lambda^*\| = O(t)$ as well for sufficiently small $t$. Thus, active constraints remain the same with perturbation of amount $O(t)$ as long as $t \ll \min_{i \in \mathcal{I}} \lambda_i^*$.

Now by the Lipschitzness of the solution map $S(\theta)$ and Lemma D.2, we have

$$\nabla l^*(w) = \nabla_w f(w, \theta), \qquad \forall \theta \in S(w).$$

To begin with, for any unit vector $v$ and arbitrarily small $t > 0$, we consider

$$\frac{\nabla l^*(w + tv) - \nabla l^*(w)}{t},$$

which approximates $\nabla^2 l^*(w)v$. Furthermore, due to Lemma D.2 and the local continuity of $\nabla^2 f$, it holds that

$$\frac{\nabla l^*(w + tv) - \nabla l^*(w)}{t} = \frac{\nabla_w f(w + tv, \theta_t^*) - \nabla_w f(w, \theta^*)}{t}$$

$$= \frac{t \nabla^2_{ww} f(w, \theta^*) v + \nabla^2_{w\theta} f(w, \theta^*)(\theta_t^* - \theta^*)}{t} + o(1).$$

If $\nabla^2 \mathcal{L}_{\mathcal{I}}^* := \nabla^2 \mathcal{L}_{\mathcal{I}}((\lambda^*)_{\mathcal{I}}, \theta^* | w, \Theta)$ is invertible, then by the implicit function theorem,

$$\begin{bmatrix} (\lambda_t^*)_{\mathcal{I}} - (\lambda^*)_{|\mathcal{I}} \\ \theta_t^* - \theta^* \end{bmatrix} = (\nabla^2 \mathcal{L}_{\mathcal{I}}^*)^{-1} \begin{bmatrix} 0 \\ \nabla^2_{\theta w} f(w, \theta^*)(tv) \end{bmatrix} + o(t).$$

In general, let the eigen-decomposition $\nabla^2 \mathcal{L}_{\mathcal{I}}^* = Q \Sigma Q^\top$ and let $r$ be the rank of $\nabla^2 \mathcal{L}_{\mathcal{I}}^*$. Without loss of generality, assume that the first $r$ columns of $Q$ correspond to non-zero eigenvalues. Let $\mu_{\min} > 0$ be the smallest *absolute* value of *non-zero* eigenvalue, and let $U := Q_r$ be the first $r$ columns of $Q$, and let $U_\perp$ be the orthogonal complement of $U$, *i.e.*, $U_\perp$ is the kernel basis of $\nabla^2 \mathcal{L}_{\mathcal{I}}^*$. We fix $U$ henceforth.

Our goal is to show that

$$U^\top \begin{bmatrix} (\lambda_t^*)_{\mathcal{I}} - (\lambda^*)_{\mathcal{I}} \\ \theta_t^* - \theta^* \end{bmatrix} = -t U^\top \nabla^2 (\mathcal{L}_{\mathcal{I}}^*)^\dagger \begin{bmatrix} 0 \\ \nabla^2_{\theta w} f(w, \theta^*) \end{bmatrix} v + o(t). \quad (24)$$

If this holds, then we can plug this into the original differentiation formula, yielding

$$\frac{\nabla l^*(w + tv) - \nabla l^*(w)}{t} = \frac{t \cdot \nabla^2_{ww} f(w, \theta^*) v + \nabla^2_{w\theta} f(w, \theta^*)(\theta_t^* - \theta^*)}{t} + o(1)$$

$$
\begin{aligned}
&= \frac{t \cdot \nabla^2_{ww} f(w,\theta^*)v + \begin{bmatrix} 0 & \nabla^2_{w\theta} f(w,\theta^*) \end{bmatrix} \begin{bmatrix} (\lambda^*_t - \lambda^*)_{\mathcal{I}} \\ \theta^*_t - \theta^* \end{bmatrix}}{t} + o(1) \\
&= \nabla^2_{ww} f(w,\theta^*)v - \left( \begin{bmatrix} 0 & \nabla^2_{w\theta} f(w,\theta^*) \end{bmatrix} U \right) (U^\top \nabla^2 \mathcal{L}^*_{\mathcal{I}} U)^{-1} \left( U^\top \begin{bmatrix} 0 \\ \nabla^2_{\theta w} f(w,\theta^*) \end{bmatrix} \right) v + o(1).
\end{aligned}
$$

Since $\mathbf{Im}(\begin{bmatrix} 0 & \nabla^2_{w\theta} f(w,\theta^*) \end{bmatrix}^\top) \subseteq \mathbf{span}(U)$, sending $t \to 0$, the limit is given by

$$
\nabla^2 l^*(w)v = \nabla^2_{ww} f(w,\theta^*)v - \begin{bmatrix} 0 & \nabla^2_{w\theta} f(w,\theta^*) \end{bmatrix} (\nabla^2 \mathcal{L}^*_{\mathcal{I}})^\dagger \begin{bmatrix} 0 \\ \nabla^2_{\theta w} f(w,\theta^*) \end{bmatrix} v.
$$

The above holds for any unit vector $v$, and we conclude (23). Note that this holds for any $\theta^* \in S(w)$ where $\mathcal{L}(\lambda^*, \theta^*|w, \Theta)$ is locally Hessian-Lipschitz (jointly in $w$, $\lambda$ and $\theta$), concluding the proof.

We are left with showing (24). For simplicity, let $y = \begin{bmatrix} \lambda_{\mathcal{I}} \\ \theta \end{bmatrix}$, and we simply denote $\mathcal{L}(w,y) := \mathcal{L}|_{\mathcal{I}}(\lambda_{\mathcal{I}}, \theta|w, \Theta)$. Consider $\mathcal{L}_U(w,z) := \mathcal{L}(w, Uz + y_0)$ where $y_0$ is a projected point of $y^*$ onto the kernel of $\nabla^2 \mathcal{L}|^*_{\mathcal{I}}$. Note that since kernel and image are orthogonal complements of each other, $y^* = Uz^* + y_0$ where $z^* = U^\top y^*$. We list a few properties of $\mathcal{L}_U(w,z)$:

$$
\begin{aligned}
\nabla_z \mathcal{L}_U(w,z) &= U^\top \nabla_\theta \mathcal{L}(w, Uz + y_0), \\
\nabla^2_{zz} \mathcal{L}_U(w,z) &= U^\top \nabla^2_{yy} \mathcal{L}(w, Uz + y_0) U, \\
\nabla^2_{wz} \mathcal{L}_U(w,z) &= \nabla^2_{wy} \mathcal{L}(w, Uz + y_0) U,
\end{aligned}
$$

and $\nabla^2 \mathcal{L}_U$ is locally uniformly Lipschitz continuous at $(w, z^*)$ jointly in $(w, z)$.

A crucial observation is that $z^*$ is a critical point of $\mathcal{L}_U(w,z)$, i.e., $\nabla_z \mathcal{L}_U(w, z^*) = U^\top \nabla_y \mathcal{L}(w, y^*) = 0$, and at $(w, z^*)$,

$$
\nabla^2_{zz} \mathcal{L}_U(w, z^*) = U^\top \nabla^2_{\theta\theta} \mathcal{L}(w, Uz^* + \theta_0) U = U^\top \nabla^2_{\theta\theta} \mathcal{L}(w, \theta^*) U,
$$

and $\min_{u:\|u\|=1} \|\nabla^2_{zz} \mathcal{L}_U(w, z^*)u\| \geq \mu_{\min}$. Tracking the movement from $z^*$ to $z^*_t$ with respect to $(tv)$ perturbations in $w$, by implicit function theorem, we have

$$
z^*_t - z^* = -t(\nabla^2_{zz} \mathcal{L}_U(w, z^*))^{-1}(\nabla^2_{zw} \mathcal{L}_U(w, z^*))v + o(t).
$$

where $z^*_t$ is the only $O(t)$-neighborhood of $z^*$ that satisfies $\nabla_z \mathcal{L}_U(w + tv, z^*_t) = 0$. Note that for any $z$ in the neighborhood of $z^*_t$,

$$
\begin{aligned}
\|\nabla_z \mathcal{L}_U(w + tv, z) - \nabla_z \mathcal{L}_U(w + tv, z^*_t)\| &= \|\nabla^2_{zz} \mathcal{L}_U(w + tv, z^*_t)(z - z^*_t)\| + O(\|z - z^*_t\|^2) \\
&\geq (\mu_{\min} - O(t) - O(\|z^* - z^*_t\|))\|z - z^*_t\| - O(\|z - z^*_t\|^2).
\end{aligned}
$$

Now let $y^t_0$ be the projection of $y^*_t$ onto the kernel of $\nabla^2 \mathcal{L}(w, \theta^*)$. Since $y^*_t = (\lambda^*_t, \theta^*_t) \in S(w + tv)$ is a global solution for $w + tv$ (without active constraints changed thanks to $\theta^*$ satisfying Definition 6), we have

$$
\begin{aligned}
0 = \|\nabla_y \mathcal{L}(w + tv, y^*_t)\| &\geq \frac{1}{\sqrt{2}} \|U^\top \nabla_y \mathcal{L}(w + tv, y^*_t)\| + \frac{1}{\sqrt{2}} \|U^\top_\perp \nabla_y \mathcal{L}(w + tv, y^*_t)\| \\
&= \frac{1}{\sqrt{2}} \underbrace{\|U^\top \nabla^2_{yy} \mathcal{L}(w + tv, Uz^*_t + y_0)(U(z^*_t - U^\top y^*_t) + (y_0 - y^t_0))\|}_{(i)} + \frac{1}{\sqrt{2}} \underbrace{\|U^\top_\perp \nabla_y \mathcal{L}(w + tv, y^*_t)\|}_{(ii)} + o(t),
\end{aligned}
$$

where we used $U^\top \nabla_y \mathcal{L}(w + tv, Uz^*_t + y_0) = \nabla_z \mathcal{L}(w + tv, y^*_t) = 0$, continuity of $\nabla^2 \mathcal{L}$, and $\|y^*_t - y^*\| = O(t)$ in the last equality. To bound $(i)$, we observe that

$$
\begin{aligned}
(i) &\geq \|U^\top \nabla^2_{yy} \mathcal{L}(w + tv, Uz^*_t + y_0)U(z^*_t - U^\top y^*_t)\| - \|U^\top \nabla^2_{yy} \mathcal{L}(w + tv, Uz^*_t + y_0)(y_0 - y^t_0)\| \\
&= \|\nabla^2_{zz} \mathcal{L}_U(w + tv, z^*_t)(z^*_t - U^\top y^*_t)\| - \|U^\top(\nabla^2_{yy} \mathcal{L}(w + tv, Uz^*_t + y_0) - \nabla^2_{yy} \mathcal{L}(w, y^*))(y^t_0 - y_0)\| \\
&\geq ((\mu_{\min} - O(t) - O(\|z^*_t - z^*\|))O(\|z^*_t - U^\top y^*_t\|) - o(t)) - o(t) \\
&= O(\mu_{\min})\|z^*_t - U^\top \theta^*_t\| - o(t).
\end{aligned}
$$

where we used $\|y_0^t - y_0\| \leq \|y_t^* - y^*\| = O(t)$, and assuming $t \ll \mu_{\min}$. On the other hand,

$$
\begin{aligned}
(ii) &= \|U_\perp^\top (\nabla_y \mathcal{L}(w + tv, y_t^*) - \nabla_y \mathcal{L}(w, y^*))\| \\
&\leq \|U_\perp^\top \left(t\nabla_{yw}^2 \mathcal{L}(w, y^*)v + \nabla_{yy}^2 \mathcal{L}(w, y^*)(y_t^* - y^*)\right)\| + o(t) = o(t),
\end{aligned}
$$

where the first equality follows from the optimality condition of $\theta^*$, and the last equality is due to necessity condition (Proposition 3.4) for the Lipscthiz-continuity of solution maps. Therefore, we conclude that

$$
0 = \|\nabla_y \mathcal{L}(w + tv, y_t^*)\| \geq O(\mu_{\min})\|z_t^* - U^\top y_t^*\| - o(t),
$$

which can only be true if $\|z_t^* - U^\top y_t^*\| = o(t)$. This means

$$
\begin{aligned}
U^\top (y_t^* - y^*) &= (U^\top y_t^* - z_t^*) + (z_t^* - z^*) \\
&= -t(\nabla_{zz}^2 \mathcal{L}_U(w, z^*)^{-1} \nabla_{zw}^2 \mathcal{L}_U(w, z^*))v + o(t),
\end{aligned}
$$

and thus we get

$$
U^\top (y_t^* - y^*) = -t \left((U^\top \nabla_{yy}^2 \mathcal{L}(w, y^*)U)^{-1} (U^\top \nabla_{yw}^2 \mathcal{L}(w, y^*))\right)v + o(t). \tag{25}
$$

Note that the constraint does not depend on $w$, and thus

$$
\nabla_{yw}^2 \mathcal{L}(w, y^*) = \begin{bmatrix} 0 \\ \nabla_{yw}^2 f(w, y^*) \end{bmatrix}.
$$

On the other hand, the necessity condition given in Proposition E.3 implies

$$
\mathbf{Im}(\nabla_{yw}^2 \mathcal{L}(w, y^*)) \subseteq \mathbf{Im}(\nabla_{yy}^2 \mathcal{L}(w, y^*)) = \mathbf{span}(U).
$$

From the above inclusion and (25), we conclude (24).

$\square$

### E.6 Proof of Theorem 3.7

*Proof.* For simplicity, $y_\sigma^* \in T(x, \sigma)$, let $z_p^*$ be a projected point of $y_\sigma^*$ onto $S(x) := T(x, 0)$. To bound $|\psi_\sigma(x) - \psi(x)|$, we first see that

$$
\begin{aligned}
\psi_\sigma(x) &= \min_{y \in \mathcal{Y}} \left(f(x, y) + g(x, y)/\sigma\right) - \min_{z \in \mathcal{Y}} g(x, z)/\sigma \\
&\leq \min_{y \in S(x)} \left(f(x, y) + g(x, y)/\sigma\right) - \min_{z \in \mathcal{Y}} g(x, z)/\sigma = \min_{z \in S(x)} f(x, z) = \psi(x).
\end{aligned}
$$

We first show that $g(x, y_\sigma^*) - g(x, z_p^*) \leq \delta$. To see this, note that

$$
\sigma f(x, y_\sigma^*) + g(x, y_\sigma^*) \leq \sigma f(x, z_p^*) + g(x, z_p^*),
$$

and thus $g(x, y_\sigma^*) - g(x, z_p^*) \leq \sigma(f(x, z_p^*) - f(x, y_\sigma^*)) \leq 2\sigma C_f$. As long as $\sigma \leq \frac{\delta}{2C_f}$, we have $g(x, y_\sigma^*) - g(x, z_p^*) \leq \delta$. Then, since we have Assumption 1, we get

$$
\psi_\sigma(x) = f(x, y_\sigma^*) + \frac{g(x, y_\sigma^*) - g(x, z_p^*)}{\sigma} \geq f(x, y_\sigma^*) + \frac{\mu\|y_\sigma^* - z_p^*\|^2}{2\sigma}.
$$

We can further observe that

$$
\begin{aligned}
f(x, y_\sigma^*) + \mu_g \frac{\|y_\sigma^* - z_p^*\|^2}{2\sigma} &\geq f(x, z_p^*) + \mu_g \frac{\|y_\sigma^* - z_p^*\|^2}{2\sigma} - l_{f,0}\|y_\sigma^* - z_p^*\| \\
&\geq f(x, z_p^*) - \frac{l_{f,0}^2}{2\mu}\sigma \geq \psi(x) - \frac{l_{f,0}^2}{2\mu}\sigma.
\end{aligned}
$$

Thus, we conclude that

$$
0 \leq \psi(x) - \psi_\sigma(x) \leq \frac{l_{f,0}^2}{2\mu}\sigma.
$$

**Gradient Convergence.** As long as the active constraint set does not change, we can only consider $\lambda_{\mathcal{I}}^*$. By $(l_{f,0}/\mu)$-Lipschitz continuity of solution sets, for all $\sigma_1, \sigma_2 \in [0, \sigma]$, we can find $y^*(\sigma_1) \in T(x, \sigma_1), y^*(\sigma_2) \in T(x, \sigma_2)$ such that

$$\|y^*(\sigma_1) - y^*(\sigma_2)\| = O(l_{f,0}/\mu) \cdot |\sigma_1 - \sigma_2|.$$

On the other hand, we check that

$$\nabla \mathcal{L}(\lambda^*(\sigma_2), \nu^*(\sigma_2), y^*(\sigma_1)|x, \sigma_1) - \nabla \mathcal{L}(\lambda^*(\sigma_1), \nu^*(\sigma_1), y^*(\sigma_1)|x, \sigma_1)$$

$$= \sum_{i \in \mathcal{I}} (\lambda^*(\sigma_2) - \lambda^*(\sigma_1)) \nabla g_i(y^*(\sigma_1)) + \sum_{i \in [m_2]} (\nu^*(\sigma_2) - \nu^*(\sigma_1)) \nabla h_i(y^*(\sigma_1))$$

$$= \nabla^2 \mathcal{L}(\lambda^*(\sigma_1), \nu^*(\sigma_1), y^*(\sigma_1)|x, \sigma_1) \begin{bmatrix} \lambda^*(\sigma_2) - \lambda^*(\sigma_1) \\ \nu^*(\sigma_2) - \nu^*(\sigma_1) \\ 0 \end{bmatrix}.$$

At the same time, we also know that

$$\nabla \mathcal{L}(\lambda^*(\sigma_2), \nu^*(\sigma_2), y^*(\sigma_2)|x, \sigma) - \nabla \mathcal{L}(\lambda^*(\sigma2), \nu^*(\sigma_2), y^*(\sigma_1)|x, \sigma_1)$$

$$\leq l_{f,0} |\sigma_2 - \sigma_1| + O(l_{g,1})(\|\lambda^*(\sigma_2)\| + \|\nu^*(\sigma_2)\|) \|y^*(\sigma_2) - y^*(\sigma_1)\|.$$

Since the two must sum up to 0, we have

$$\nabla^2 \mathcal{L}(\lambda^*, \nu^*, y^*|x, \sigma_1) \begin{bmatrix} \lambda^*(\sigma_2) - \lambda^*(\sigma_1) \\ \nu^*(\sigma_2) - \nu^*(\sigma_1) \\ 0 \end{bmatrix} = O(l_{g,1} l_{f,0}/\mu)|\sigma_2 - \sigma_1|.$$

Thus, with Assumption 8, we have

$$\|\lambda_{\mathcal{I}}^*(\sigma_2) - \lambda_{\mathcal{I}}^*(\sigma_1)\|, \|\nu_{\mathcal{I}}^*(\sigma_2) - \nu_{\mathcal{I}}^*(\sigma_1)\| = O(l_{f,0} l_{g,1}/(\mu s_{\min}))|\sigma_2 - \sigma_1|.$$

Thus, we can conclude that

$$\|\nabla^2 \mathcal{L}_{\mathcal{I}}^*(\sigma_2)) - \nabla^2 \mathcal{L}_{\mathcal{I}}^*(\sigma_1)\| \lesssim \left( \frac{l_{f,0} l_{g,1}^2}{\mu s_{\min}} + \frac{l_{h,2} l_{f,0}}{\mu} \right) |\sigma_2 - \sigma_1|.$$

where $(\nabla^2 \mathcal{L}_{\mathcal{I}}^*(\sigma))$ is a short-hand for $\nabla^2 \mathcal{L}(\lambda_{\mathcal{I}}^*(\sigma), \nu^*(\sigma), y^*(\sigma)|x, \sigma, \mathcal{Y})$.

To check whether $\nabla \psi_\sigma(x)$ well-approximates $\nabla \psi(x) = \frac{\partial^2}{\partial x \partial \sigma} l(x, \sigma)|_{\sigma=0^+}$, we first check that for any $\sigma_1, \sigma_2 \in [0, \sigma]$,

$$\left\| \frac{\partial^2}{\partial x \partial \sigma} l(x, \sigma_2) - \frac{\partial^2}{\partial x \partial \sigma} l(x, \sigma_1) \right\| \leq \|\nabla_x f(x, y^*(\sigma_2)) - \nabla_x f(x, y^*(\sigma_1))\|$$

$$+ \underbrace{\left\| \nabla_{xy}^2 h_{\sigma_2}(x, y^*(\sigma_2)) - \nabla_{xy}^2 h_{\sigma_1}(x, y^*(\sigma_1)) \right\| \cdot \left\| \nabla^2 \mathcal{L}_{\mathcal{I}}^*(\sigma_1)^\dagger \begin{bmatrix} 0 \\ \nabla_y f(x, y^*(\sigma_1)) \end{bmatrix} \right\|}_{(i)}$$

$$+ \underbrace{\left\| \begin{bmatrix} 0 & \nabla_{xy}^2 h_{\sigma_2}(x, y^*(\sigma_2)) \end{bmatrix} (\nabla^2 \mathcal{L}_{\mathcal{I}}^*(\sigma_2))^\dagger - (\nabla^2 \mathcal{L}_{\mathcal{I}}^*(\sigma_1)^\dagger) \begin{bmatrix} 0 \\ \nabla_y f(x, y^*(\sigma_1)) \end{bmatrix} \right\|}_{(ii)}$$

$$+ \underbrace{\left\| \begin{bmatrix} 0 & \nabla_{xy}^2 h_{\sigma_2}(x, y^*(\sigma_2)) \end{bmatrix} \nabla^2 \mathcal{L}_{\mathcal{I}}^*(\sigma_2))^\dagger \right\| \|\nabla_y f(x, y^*(\sigma_2)) - \nabla_y f(x, y^*(\sigma_1))\|}_{(iii)}.$$

Here, we use the explicit formula of $\frac{\partial^2}{\partial x \partial \sigma} l(x, \sigma)$ given in Theorem 3.1. To bound $(i)$, note the meaning of the latter term:

$$\begin{bmatrix} d\lambda/d\sigma \\ dy/d\sigma \end{bmatrix} = \nabla^2 \mathcal{L}_{\mathcal{I}}^*(\sigma_1)^\dagger \begin{bmatrix} 0 \\ \nabla_y f(x, y^*(\sigma_1)) \end{bmatrix},$$

where $\begin{bmatrix} d\lambda/d\sigma \\ dy/d\sigma \end{bmatrix}$ is the movement of $y^*(\sigma_1)$ to the nearest solution by perturbing $\sigma$ projected to the image of $\nabla^2 \mathcal{L}_{\mathcal{I}}^*(\sigma_1)$. By Lemma 3.6, $\|dy/d\sigma\|$ must not exceed $O(l_{f,0}/\mu)$. Consequently,

$$[\nabla g_i(y^*(\sigma)), \forall i \in \mathcal{I} \quad | \nabla h_i(y^*(\sigma)), \forall i \in [m_2]] \frac{d\lambda}{d\sigma} + \nabla_{yy}^2 \mathcal{L}_{\mathcal{I}}^*(\sigma_1) \frac{dy}{d\sigma} = \nabla_y f(x, y^*(\sigma_1)),$$

which enforces that $\|d\lambda/d\sigma\| \leq \frac{l_{g,1}l_{f,0}}{\mu s_{\min}}$ by Assumption 8. Thus,

$$(i) \lesssim \frac{l_{h,2}l_{f,0}}{\mu} \frac{l_{g,1}l_{f,0}}{\mu s_{\min}} |\sigma_1 - \sigma_2| = \frac{l_{h,2}l_{g,1}l_{f,0}^2}{\mu^2 s_{\min}} |\sigma_1 - \sigma_2|.$$

Similarly, we can show that

$$(iii) \lesssim \frac{l_{f,1}l_{g,1}l_{f,0}}{\mu^2 s_{\min}} |\sigma_1 - \sigma_2|.$$

For $(ii)$, note that

$$\left\| \begin{bmatrix} 0 & \nabla_{xy}^2 h_{\sigma_2}(x, y^*(\sigma_2)) \end{bmatrix} (\nabla^2 \mathcal{L}_{\mathcal{I}}^*(\sigma_2))^\dagger - (\nabla^2 \mathcal{L}_{\mathcal{I}}^*(\sigma_1)^\dagger) \begin{bmatrix} 0 \\ \nabla_y f(x, y^*(\sigma_1)) \end{bmatrix} \right\|$$

$$= \left\| \begin{bmatrix} 0 & \nabla_{xy}^2 h_{\sigma_2}(x, y^*(\sigma_2)) \end{bmatrix} \nabla^2 \mathcal{L}_{\mathcal{I}}^*(\sigma_2)^\dagger \left( \nabla^2 \mathcal{L}_{\mathcal{I}}^*(\sigma_2) - \nabla^2 \mathcal{L}_{\mathcal{I}}^*(\sigma_1) \right) \nabla^2 \mathcal{L}_{\mathcal{I}}^*(\sigma_1)^\dagger \begin{bmatrix} 0 \\ \nabla_y f(x, y^*(\sigma_1)) \end{bmatrix} \right\|$$

$$\leq \frac{l_{g,1}^2 l_{f,0}^2}{\mu^2 s_{\min}^2} \|\nabla^2 \mathcal{L}_{\mathcal{I}}^*(\sigma_2) - \nabla^2 \mathcal{L}_{\mathcal{I}}^*(\sigma_1)\|$$

$$\lesssim \frac{l_{g,1}^2 l_{f,0}^2}{\mu^2 s_{\min}^2} \left( \frac{l_{f,0}l_{g,1}^2}{\mu s_{\min}} + \frac{l_{h,2}l_{f,0}}{\mu} \right) |\sigma_2 - \sigma_1|,$$

where the first equality comes from the fact that

$$\mathbf{Im}\left( \begin{bmatrix} 0 \\ \nabla_{yx}^2 h_{\sigma_2}(x, y^*(\sigma_2)) \end{bmatrix} \right) \subseteq \mathbf{Im}\nabla^2 \mathcal{L}_{\mathcal{I}}^*(\sigma_2),$$

and similarly,

$$\begin{bmatrix} 0 \\ \nabla_y f(x, y^*(\sigma_1)) \end{bmatrix} \in \mathbf{Im}\nabla^2 \mathcal{L}_{\mathcal{I}}^*(\sigma_1),$$

by Proposition 3.4. Therefore, $\frac{\partial^2}{\partial x \partial \sigma} l(x, \sigma)$ is Lipschitz-continuous in $\sigma$, and by Mean-Value Theorem, we can conclude that

$$\|\nabla \psi_\sigma(x) - \nabla \psi(x)\| \leq O(\sigma/\mu^3) \cdot \left( \frac{l_{g,1}^4 l_{f,0}^3}{s_{\min}^3} + \frac{l_{h,2}l_{g,1}^2 l_{f,0}^3}{s_{\min}^2} \right),$$

counting the dominating term. $\qquad\square$

### E.7 $\epsilon$-STATIONARY POINT AND $\epsilon$-KKT SOLUTION

To simplify the argument, we assume that $\mathcal{X} = \mathbb{R}^{d_x}$. Then, define an $\epsilon$-KKT condition of ($\mathbf{P}_{\mathrm{con}}$) as:

$$\|\nabla_x f(x, y) + \lambda_x(\nabla_x g(x, y) - \nabla g^*(x))\| \leq \epsilon,$$
$$\|\nabla_y f(x, y) + \lambda_x \nabla_y g(x, y) + \sum_{i \in [m_1]} \lambda_i^* \nabla g_i(y) + \sum_{i \in [m_2]} \nu_i^* \nabla h_i(y)\| \leq \epsilon,$$
$$g(x, y) - g^*(x) \leq \epsilon^2.$$

for some Lagrangian multipliers $\lambda_x \geq 0$ and $\lambda^* \geq 0, \nu^*$ with some $(x, y) \in \mathcal{X} \times \mathcal{Y}$.

**Theorem E.5** *Suppose Assumptions 1-3 hold, $\mathcal{X} = \mathbb{R}^{d_x}$. Then $\nabla \psi_\sigma(x)$ is well-defined with $\sigma \leq \sigma_0$. If $x$ is an $\epsilon$-stationary point of $\psi_\sigma(x)$ with $\sigma \leq 1$, that is,*

$$\|\nabla \psi_\sigma(x)\| \leq \epsilon,$$

*then $x$ is an $O(\epsilon + \sigma)$-KKT solution of ($\mathbf{P}_{\mathrm{con}}$).*

*Proof.* This comes almost immediately from Lemma D.2. Let $y$ and $z$ as minimizers:

$$y \in \arg\min_{w \in \mathcal{Y}} \sigma f(x, w) + g(x, w),$$

$$z \in \arg\min_{w \in \mathcal{Y}} g(x, w).$$

Then, by the optimality condition of $y$, the $\epsilon$-optimality condition with respect to $\nabla_y$ is automatically satisfied with $\lambda_x = 1/\sigma$. Furthermore, since $T(x, \sigma)$ is Lipshictz-continuous due to Assumption 1 with $L_T = l_{f,0}/\mu$, we have $\nabla g^*(x) = \nabla_x g(x, z)$ and $\nabla h_\sigma^*(x) = \nabla_x \sigma f(x, y) + \nabla_x g(x, y)$. Finally, by the optimality condition, we know that $y$ is the optimal solution of a proximal operation $\mathbf{prox}_{\rho h_\sigma(x,\cdot)}(y)$:

$$\sigma f(x, y) + g(x, y) \leq \sigma f(x, z) + g(x, z) + \frac{1}{2\rho}\|z - y\|^2.$$

Using $|f(x, y) - f(x, z)| \leq l_{f,0}\|y - z\|$ and $\|y - z\| \leq \sigma L_T$, we have

$$g(x, y) - g(x, z) = g(x, y) - g^*(x) \leq \sigma^2 l_{f,0} L_T + \frac{\sigma^2 L_T^2}{2\rho} = O(\sigma^2),$$

as claimed. $\qquad\square$

## APPENDIX F    ANALYSIS FOR ALGORITHM 1

For simplicity, let $w_{y,k}^* = \mathbf{prox}_{\rho h_{\sigma_k}(x_k,\cdot)}(y_k)$ and $w_{z,k}^* = \mathbf{prox}_{\rho g(x_k,\cdot)}(z_k)$.

### F.1    DESCENT LEMMA FOR $w_{y,k}, w_{z,k}$

We first analyze $\|w_{y,k} - \mathbf{prox}_{\rho h_{\sigma_k}(x_k,\cdot)}(y_k)\|^2$. We start by observing that

$$\|w_{y,k+1} - w_{y,k+1}^*\|^2 = \|w_{y,k+1} - w_{y,k}^*\|^2 + \|w_{y,k+1}^* - w_{y,k}^*\|^2 - 2\langle w_{y,k+1} - w_{y,k}^*, w_{y,k+1}^* - w_{y,k}^* \rangle$$

$$\leq \left(1 + \frac{\lambda_k}{4}\right)\underbrace{\|w_{y,k+1} - w_{y,k}^*\|^2}_{(i)} + \left(1 + \frac{4}{\lambda_k}\right)\underbrace{\|w_{y,k+1}^* - w_{y,k}^*\|^2}_{(ii)}, \qquad (26)$$

where we used $\langle a, b \rangle \leq c\|a\|^2 + \frac{1}{4c}\|b\|^2$, and $\lambda_k = T_k \gamma_k/(4\rho)$ as defined. They are bounded in two following lemmas.

**Lemma F.1** *At every $k^{th}$ iteration, the following holds:*

$$\mathbb{E}[\|w_{y,k+1} - w_{y,k}^*\|^2|\mathcal{F}_k] \leq \left(1 - \frac{\gamma_k}{4\rho}\right)^{T_k}\mathbb{E}[\|w_{y,k} - w_{y,k}^*\|^2|\mathcal{F}_k] + 2\left(T_k \gamma_k^2\right)(\sigma_k^2 \cdot \sigma_f^2 + \sigma_g^2).$$

$$(27)$$

*Similarly, we also have that*

$$\mathbb{E}[\|w_{z,k+1} - w_{z,k}^*\|^2|\mathcal{F}_k] \leq \left(1 - \frac{\gamma_k}{4\rho}\right)^{T_k}\mathbb{E}[\|w_{z,k} - w_{z,k}^*\|^2|\mathcal{F}_k] + 2(T_k \gamma_k^2)(\sigma_k^2 \cdot \sigma_f^2 + \sigma_g^2). \quad (28)$$

*Proof.* We use the linear convergence of projected gradient steps. To simplify the notation, let

$$\widetilde{G}_t = \nabla_y(\sigma_k f(x_k, u_t; \zeta_{wy}^{k,t}) + g(x_k, u_t; \xi_{wy}^{k,t})) + \rho^{-1}(u_t - y_k),$$

and $G_t = \mathbb{E}[\widetilde{G}_t]$. Also let $G^* = \nabla h_{\sigma_k}(x, w_{y,k}^*) + \rho^{-1}(w_{y,k}^* - y_k)$. We first check that

$$\|u_{t+1} - w_{y,k}^*\|^2 = \left\|\Pi_{\mathcal{Y}}\left\{u_t - \gamma_k \widetilde{G}_t\right\} - \Pi_{\mathcal{Y}}\left\{w_{y,k}^* - \gamma_k G^*\right\}\right\|^2$$

$$\leq \left\|u_t - \gamma_k \widetilde{G}_t - (w_{y,k}^* - \gamma_k G^*)\right\|^2$$

$$= \left\| u_t - w_{y,k}^* \right\|^2 + \gamma_k^2 \left\| \widetilde{G}_t - G^* \right\|^2 - 2\gamma_k \langle u_t - w_{y,k}^*, \widetilde{G}_t - G^* \rangle.$$

Taking expectation conditioned on $\mathcal{F}_{k,t}$ yields:

$$\mathbb{E}[\|u_{t+1} - w_{y,k}^*\|^2 | \mathcal{F}_{k,t}] \leq \mathbb{E}[\|u_t - w_{y,k}^*\|^2 | \mathcal{F}_{k,t}] + \gamma_k^2 \mathbb{E}[\|\widetilde{G}_t - G^*\|^2 | \mathcal{F}_{k,t}]$$
$$- 2\gamma_k \langle u_t - w_{y,k}^*, G_t - G^* \rangle.$$

Note that

$$\mathbb{E}[\|\widetilde{G}_t - G^*\|^2 | \mathcal{F}_k] \leq 2\|G_t - G^*\|^2 + 2\mathbb{E}[\|\widetilde{G}_t - G_t\|^2 | \mathcal{F}_{k,t}].$$

By co-coercivity of strongly convex function, since the inner minimization is $(1/(3\rho))$-strongly convex and $(1/\rho)$-smooth, we have

$$\|G_t - G^*\|^2 \leq (1/\rho) \cdot \langle u_t - w_{y,k}^*, G_t - G^* \rangle,$$

$$\frac{1}{3\rho} \cdot \|u_t - w_{y,k}^*\|^2 \leq \langle u_t - w_{y,k}^*, G_t - G^* \rangle.$$

Given $\gamma_k \ll \rho$, we have

$$\mathbb{E}[\|u_{t+1} - w_{y,k}^*\|^2 | \mathcal{F}_{k,t}] \leq \left(1 - \frac{\gamma_k}{4\rho}\right) \mathbb{E}[\|u_t - w_{y,k}^*\|^2 | \mathcal{F}_{k,t}] + 2\gamma_k^2 (\sigma_k^2 \cdot \sigma_f^2 + \sigma_g^2).$$

Applying this for $T_k$ steps, we get the lemma. □

**Lemma F.2** *At every $k^{th}$ iteration, the following holds:*

$$\mathbb{E}[\|w_{y,k+1} - w_{y,k+1}^*\|^2 | \mathcal{F}_k] \leq \left(1 + \frac{\lambda_k}{4}\right) \mathbb{E}[\|w_{y,k+1} - w_{y,k}^*\|^2 | \mathcal{F}_k] + O\left(\frac{\rho^2 l_{f,0}^2}{\lambda_k}\right) |\sigma_k - \sigma_{k+1}|^2$$

$$+ O\left(\frac{\rho l_{g,1}}{\lambda_k}\right) \|x_{k+1} - x_k\|^2 + \frac{8}{\lambda_k} \|y_{k+1} - y_k\|^2. \tag{29}$$

*Similarly, we have*

$$\mathbb{E}[\|w_{z,k+1} - w_{z,k+1}^*\|^2 | \mathcal{F}_k] \leq \left(1 + \frac{\lambda_k}{4}\right) \mathbb{E}[\|w_{z,k+1} - w_{z,k}^*\|^2 | \mathcal{F}_k] + O\left(\frac{\rho^2 l_{f,0}^2}{\lambda_k}\right) |\sigma_k - \sigma_{k+1}|^2$$

$$+ O\left(\frac{\rho l_{g,1}}{\lambda_k}\right) \|x_{k+1} - x_k\|^2 + \frac{8}{\lambda_k} \|z_{k+1} - z_k\|^2. \tag{30}$$

*Proof.* By Lemmas D.3 and D.4, we have

$$\|w_{y,k+1}^* - w_{y,k}^*\| \leq O(\rho l_{g,1}) \|x_{k+1} - x_k\| + \|y_{k+1} - y_k\| + O(\rho l_{f,0}) |\sigma_k - \sigma_{k+1}|.$$

Take square and conditional expectation, and plug this to the bound for $(i), (ii)$ in (26), we get the lemma. □

## F.2 DESCENT LEMMA FOR $\Phi_{\sigma,\rho}$

**Proposition F.3** *At every $k^{th}$ iteration, we have*

$$\sigma_k \left( \Phi_{\sigma_{k+1},\rho}(x_{k+1}, y_{k+1}, z_{k+1}) - \Phi_{\sigma_k,\rho}(x_k, y_k, z_k) \right)$$

$$\leq C_1 \rho^{-1} \left\{ \|y_k - y_{k+1}\|^2 + \frac{l_{f,0}^2}{\mu^2} |\sigma_k - \sigma_{k+1}|^2 + \mathbf{dist}^2(z_k, T(x_k, 0)) + \mathbf{dist}^2(y_k, T(x_k, \sigma_k)) \right\}$$

$$+ \left( \frac{C_1 l_{g,1}^2}{\rho \mu^2} - \frac{1}{4\alpha_k} \right) \|x_k - x_{k+1}\|^2 + \left( \frac{C_1}{\rho} + O(\rho^{-2})\alpha_k \right) \left( \|z_k - z_{k+1}\|^2 + \mathbf{dist}^2(z_k, T(x_k, 0)) \right)$$

$$- \frac{\beta_k}{4\rho} (\|y_k - w_{y,k}^*\|^2 + \|y_k - w_{y,k+1}\|^2) - \frac{\beta_k}{\rho} (\|z_k - w_{z,k}^*\|^2 + \|z_k - w_{z,k+1}\|^2)$$

$$+ O\left(l_{g,1}^2 \alpha_k + \rho^{-1} \beta_k\right) \left( \|w_{y,k}^* - w_{y,k+1}\|^2 + \|w_{z,k}^* - w_{z,k+1}\|^2 \right) + \alpha_k \|\widetilde{G} - G\|^2 + \sigma_{k+1} C_1 C_f \tag{31}$$

*where $C_1 = O\left(\frac{\sigma_k - \sigma_{k+1}}{\sigma_{k+1}}\right)$.*

*Proof.* To start with, note that

$$\Phi_{\sigma_{k+1},\rho}(x_{k+1}, y_{k+1}, z_{k+1}) - \Phi_{\sigma_k,\rho}(x_k, y_k, z_k) = \underbrace{\Phi_{\sigma_k,\rho}(x_{k+1}, y_{k+1}, z_{k+1}) - \Phi_{\sigma_k,\rho}(x_k, y_k, z_k)}_{(i)}$$

$$+ \underbrace{\Phi_{\sigma_{k+1},\rho}(x_{k+1}, y_{k+1}, z_{k+1}) - \Phi_{\sigma_k,\rho}(x_{k+1}, y_{k+1}, z_{k+1})}_{(ii)}.$$

Note that by Lemma 3.6, we have $\mathbf{dist}(T(x_1, \sigma_1), T(x_2, \sigma_2)) \leq \frac{l_{g,1}}{\mu}\|x_1 - x_2\| + \frac{l_{f,0}}{\mu}|\sigma_1 - \sigma_2|$ for all $x_1, x_2 \in \mathcal{X}, \sigma_1, \sigma_2 \in [0, \delta/C_f]$. Applying this to (32) in the subsequent subsection, we obtain that

$$(ii) \leq O\left(\frac{\sigma_k - \sigma_{k+1}}{\sigma_k \sigma_{k+1}}\right) \rho^{-1} \left\{ \|y_k - y_{k+1}\|^2 + \|z_k - z_{k+1}\|^2 + \frac{l_{g,1}^2}{\mu^2}\|x_k - x_{k+1}\|^2 + \frac{l_{f,0}^2}{\mu^2}|\sigma_k - \sigma_{k+1}|^2 \right.$$

$$\left. \mathbf{dist}^2(z_k, T(x_k, 0)) + \mathbf{dist}^2(y_k, T(x_k, \sigma_k)) \right\} + O\left(\frac{\sigma_k - \sigma_{k+1}}{\sigma_k}\right) C_f.$$

Combining this with the estimation of $(i)$ given in (36), we conclude. $\qquad\square$

### F.2.1 BOUNDING $(ii)$

For $(ii)$, we realize that for any $x, y, z$ with $\sigma_{k+1} < \sigma_k$,

$$\Phi_{\sigma_{k+1},\rho}(x, y, z) - \Phi_{\sigma_k,\rho}(x, y, z) = \frac{h_{\sigma_{k+1},\rho}^*(x, y) - g^*(x, z)}{\sigma_{k+1}} - \frac{h_{\sigma_k,\rho}^*(x, y) - g^*(x, z)}{\sigma_k}$$

$$+ \left(\frac{C}{\sigma_{k+1}} - \frac{C}{\sigma_k}\right)\left(g_\rho^*(x, z) - g^*(x)\right)$$

$$\leq \underbrace{\frac{h_{\sigma_{k+1},\rho}^*(x, y) - g_\rho^*(x, y)}{\sigma_{k+1}} - \frac{h_{\sigma_k,\rho}^*(x, y) - g_\rho^*(x, y)}{\sigma_k}}_{(iii)}$$

$$+ \underbrace{\left(\frac{\sigma_k - \sigma_{k+1}}{\sigma_k \sigma_{k+1}}\right)\left(g_\rho^*(x, y) - g^*(x)\right)}_{(iv)}$$

$$+ \underbrace{\left(\frac{C(\sigma_k - \sigma_{k+1})}{\sigma_{k+1}\sigma_k}\right)\left(g_\rho^*(x, z) - g^*(x)\right)}_{(v)}.$$

To bound $(iii)$, for any $\sigma_1 > \sigma_2$, note that

$$h_{\sigma_1,\rho}^*(x, y) \leq \sigma_1 f(x, w_2^*) - g(x, w_2^*) + \frac{\|w_2^* - y\|^2}{2\rho} = h_{\sigma_2,\rho}^*(x, y) + (\sigma_1 - \sigma_2)f(x, w_2^*),$$

where $w_2^* = \arg\min_{w \in \mathcal{Y}} \sigma_2 f(x, w) + g(x, w) + \frac{\|w - y\|^2}{2\rho}$. Thus,

$$(iii) \leq \left(\frac{1}{\sigma_{k+1}} - \frac{1}{\sigma_k}\right)\left(h_{\sigma_{k+1},\rho}^*(x, y) - h_{0,\rho}^*(x, y)\right) - \frac{1}{\sigma_k}(h_{\sigma_k,\rho}^*(x, y) - h_{\sigma_{k+1},\rho}^*(x, y))$$

$$\leq 2\frac{\sigma_k - \sigma_{k+1}}{\sigma_k} \cdot \max_{w \in \mathcal{Y}}|f(x, w)| \leq \frac{\sigma_k - \sigma_{k+1}}{\sigma_k} \cdot O(C_f).$$

In order to bound $(iv)$, note that for any $y_\sigma^* \in T(x, \sigma)$,

$$g_\rho^*(x, y) - g^*(x) = (g_\rho^*(x, y) - h_\sigma^*(x)) + (h_\sigma^*(x) - g^*(x))$$

$$\leq (h_{\sigma,\rho}^*(x, y) - h_{\sigma,\rho}^*(x, y_\sigma^*)) + O(\sigma C_f)$$

$$\leq \underbrace{\langle\nabla_y h_{\sigma,\rho}^*(x, y_\sigma^*)}_{=0}, y - y_\sigma^*\rangle + O(\rho^{-1})\|y - y_\sigma^*\|^2 + O(\sigma C_f).$$

where $\nabla_y h_{\sigma,\rho}^*(x, y_\sigma^*) = \rho^{-1}(y_\sigma^* - \mathbf{prox}_{\rho h_\sigma(x,\cdot)}(y_\sigma^*)) = 0$ since $y_\sigma^*$ is a fixed point of $\mathbf{prox}_{\rho h_\sigma(x,\cdot)}$ operation. Taking $y_\sigma^*$ the closest element to $y$, we get

$$(iv) \leq \frac{\sigma_k - \sigma_{k+1}}{\sigma_k \sigma_{k+1}} \left( \rho^{-1}\mathbf{dist}^2(y_{k+1}, T(x_{k+1}, \sigma_{k+1})) + O(\sigma_{k+1}C_f) \right).$$

Similarly, we can also show that

$$(v) \leq \frac{C(\sigma_k - \sigma_{k+1})}{\sigma_k \sigma_{k+1}} \cdot \rho^{-1}\mathbf{dist}^2(z_{k+1}, T(x_{k+1}, 0)).$$

Thus, we can conclude that

$$(ii) \leq O\left( \frac{\sigma_k - \sigma_{k+1}}{\sigma_k \sigma_{k+1}} \right) \left( \sigma_{k+1}C_f + \rho^{-1}\mathbf{dist}^2(y_{k+1}, T(x_{k+1}, \sigma_{k+1})) + \rho^{-1}\mathbf{dist}^2(z_{k+1}, T(x_{k+1}, 0)) \right)$$

$$\leq O\left( \frac{\sigma_k - \sigma_{k+1}}{\sigma_k \sigma_{k+1}} \right) \rho^{-1} \left( \|z_{k+1} - z_k\|^2 + \mathbf{dist}^2(z_k, T(x_k, 0)) + \mathbf{dist}^2(T(x_{k+1}, 0), T(x_k, 0)) \right)$$

$$+ O\left( \frac{\sigma_k - \sigma_{k+1}}{\sigma_k \sigma_{k+1}} \right) \rho^{-1} \left( \|y_{k+1} - y_k\|^2 + \mathbf{dist}^2(y_k, T(x_k, \sigma_k)) + \mathbf{dist}^2(T(x_{k+1}, \sigma_{k+1}), T(x_k, \sigma_k)) \right)$$

$$+ O\left( \frac{\sigma_k - \sigma_{k+1}}{\sigma_k} \right) C_f. \tag{32}$$

### F.2.2 Bounding $(i)$

Henceforth, to simplify the notation, we simply denote $\sigma = \sigma_k$.

$$(i) = \frac{1}{\sigma} \underbrace{\left( h_{\sigma,\rho}^*(x_{k+1}, y_{k+1}) - g_\rho^*(x_{k+1}, z_{k+1}) - (h_{\sigma,\rho}^*(x_k, y_k) - g_\rho^*(x_k, z_k)) \right)}_{(a) = \sigma \cdot (\psi_{\sigma,\rho}(x_{k+1}, y_{k+1}, z_{k+1}) - \psi_{\sigma,\rho}(x_k, y_k, z_k))}$$

$$+ \frac{C}{\sigma} \underbrace{\left( (g_\rho^*(x_{k+1}, z_{k+1}) - g^*(x_{k+1})) - (g_\rho^*(x_k, z_k) - g^*(x_k)) \right)}_{(b)}.$$

**Bounding** $(a)$. It is easy to check using Lemma D.2 that

$$\nabla_y h_{\sigma,\rho}^*(x_k, y_k) = \rho^{-1}(y_k - w_{y,k}^*),$$
$$\nabla_z g_\rho^*(x_k, z_k) = \rho^{-1}(z_k - w_{z,k}^*),$$

Note that $y_{k+1} - y_k = -\beta_k(y_k - w_{y,k+1})$, and thus,

$$\langle \nabla_y h_{\sigma,\rho}^*(x_k, y_k), y_{k+1} - y_k \rangle = \frac{-\beta_k}{\rho} \langle y_k - w_{y,k}^*, y_k - w_{y,k+1} \rangle$$

$$= \frac{-\beta_k}{2\rho} \left( \|y_k - w_{y,k}^*\|^2 + \|y_k - w_{y,k+1}\|^2 - \|w_{y,k}^* - w_{y,k+1}\|^2 \right).$$

Similarly, $z_{k+1} - z_k = -\beta_k(z_k - w_{z,k+1})$, and thus

$$\langle \nabla_z g_\rho^*(x_k, z_k), z_{k+1} - z_k \rangle = \frac{-\beta_k}{\rho} \langle z_k - w_{z,k}^*, z_k - w_{z,k+1} \rangle$$

$$= \frac{-\beta_k}{2\rho} \left( \|z_k - w_{z,k}^*\|^2 + \|z_k - w_{z,k+1}\|^2 - \|w_{z,k}^* - w_{z,k+1}\|^2 \right).$$

Using smoothness of $h_{\sigma,\rho}^*$ and $g_\rho^*$, and noting that

$$\nabla_x(h_{\sigma,\rho}^*(x_k, y_k) - g_\rho^*(x_k, z_k)) = \nabla_x\psi_{\sigma,\rho}(x_k, y_k, z_k) = \nabla_x(h_\sigma(x_k, w_{y,k}^*) - g(x_k, w_{z,k}^*)),$$

we get (caution on the sign of $g_\rho^*(x_k, z_k)$ terms):

$$(a) \leq \langle \sigma \cdot \nabla_x\psi_{\sigma,\rho}(x_k, y_k, z_k), x_{k+1} - x_k \rangle + \frac{O(1)}{\rho}\|x_{k+1} - x_k\|^2$$

$$- \frac{\beta_k}{2\rho} \left( \|y_k - w_{y,k}^*\|^2 + \frac{1}{2}\|y_k - w_{y,k+1}\|^2 \right) + \frac{\beta_k}{2\rho}\|w_{y,k}^* - w_{y,k+1}\|^2$$

$$+ \frac{\beta_k}{2\rho} \left( \|z_k - w_{z,k}^*\|^2 + 2\|z_k - w_{z,k+1}\|^2 \right) - \frac{\beta_k}{2\rho}\|w_{z,k}^* - w_{z,k+1}\|^2. \tag{33}$$

where we assume $\beta_k \ll 1$. For terms regarding $x$, let

$$\widetilde{G} := \frac{1}{M_k} \sum_{m=1}^{M_k} \nabla_x \left( \sigma_k f(x_k, w_{y,k+1}; \zeta_x^{k,m}) + g(x_k, w_{y,k+1}; \xi_{xy}^{k,m}) - g(x_k, w_{z,k+1}; \xi_{xz}^{k,m}) \right),$$

and $G = \mathbb{E}[\widetilde{G}]$. By projection lemma, we have

$$\langle (x_k - \alpha_k \widetilde{G}) - x_{k+1}, x - x_{k+1} \rangle \le 0, \qquad \forall x \in \mathcal{X},$$

and therefore $\langle \widetilde{G}, x_{k+1} - x \rangle \le -\frac{1}{\alpha_k}\langle x_k - x_{k+1}, x - x_{k+1} \rangle$ for all $x \in \mathcal{X}$. Plugging $x = x_k$ here, we have

$$\langle G^*, x_{k+1} - x_k \rangle \le -\frac{1}{\alpha_k}\|x_k - x_{k+1}\|^2 + \langle G^* - \widetilde{G}, x_{k+1} - x_k \rangle$$

$$\le -\frac{1}{2\alpha_k}\|x_k - x_{k+1}\|^2 + \alpha_k \left( \|G^* - G\|^2 + \|\widetilde{G} - G\|^2 \right).$$

Note that

$$\|G^* - G\| \le l_{g,1}(\|w_{y,k}^* - w_{y,k+1}\| + \|w_{z,k}^* - w_{z,k+1}\|).$$

In conclusion, omitting expectations on both sides, we have

$$(a) \le -\frac{1}{2\alpha_k}\|x_{k+1} - x_k\|^2 - \frac{\beta_k}{4\rho}(\|y_k - w_{y,k}^*\|^2 + \|y_k - w_{y,k+1}\|^2)$$

$$+ \frac{O(1)}{\rho}\|x_{k+1} - x_k\|^2 + \frac{\beta_k}{\rho}(\|z_k - w_{z,k}^*\|^2 + \|z_k - w_{z,k+1}\|^2)$$

$$+ \left( O(l_{g,1}^2)\alpha_k + \frac{\beta_k}{2\rho} \right)(\|w_{y,k}^* - w_{y,k+1}\|^2 + \|w_{z,k}^* - w_{z,k+1}\|^2) + \alpha_k\|\widetilde{G} - G\|^2. \tag{34}$$

**Bounding (b).** We realize that in (34), coefficients of proximal error terms on $z$, *i.e.*, $\|z_k - w_{z,k}^*\|^2$ are positive, unlike terms regarding $y_k$. We show that these terms will be canceled out with $(b)$ when Assumption 1 holds. Using Lemma 3.6,

$$(b) = (g_\rho^*(x_{k+1}, z_{k+1}) - g^*(x_{k+1})) - (g_\rho^*(x_k, z_{k+1}) - g^*(x_k)) + (g_\rho^*(x_k, z_{k+1}) - g_\rho^*(x_k, z_k))$$

$$\le \langle \nabla_x g_\rho^*(x_k, z_{k+1}) - \nabla_x g^*(x_k), x_{k+1} - x_k \rangle + O\left( \frac{l_{g,1}}{\mu} \right)\|x_{k+1} - x_k\|^2$$

$$+ \langle \nabla_z g_\rho^*(x_k, z_k), z_{k+1} - z_k \rangle + O(\rho^{-1})\|z_{k+1} - z_k\|^2. \tag{35}$$

Taking conditional expectation on both sides and using $z_{k+1} - z_k = -\beta_k(z_k - w_{z,k+1})$ and $\nabla_z g_\rho^*(x_k, z_k) = \rho^{-1}(z_k - w_{z,k}^*)$,

$$\mathbb{E}[(b)|\mathcal{F}_k'] \le -\langle \nabla_x g_\rho^*(x_k, z_{k+1}) - \nabla_x g^*(x_k), x_{k+1} - x_k \rangle + O\left( \frac{l_{g,1}}{\mu} \right)\mathbb{E}[\|x_{k+1} - x_k\|^2|\mathcal{F}_k']$$

$$- \beta_k\langle \nabla_z g_\rho^*(x_k, z_k), z_k - w_{z,k+1} \rangle + O(\beta_k^2\rho^{-1})\|z_k - w_{z,k+1}\|^2$$

$$\le \left( O(\rho^{-2})\alpha_k C \cdot \mathbf{dist}^2(z_{k+1}, T(x_k, 0)) + \frac{\|x_k - x_{k+1}\|^2}{16C\alpha_k} \right) + O\left( \frac{l_{g,1}}{\mu} \right)\mathbb{E}[\|x_{k+1} - x_k\|^2|\mathcal{F}_k']$$

$$- \frac{\beta_k}{2\rho} \left( \|z_k - w_{z,k}^*\|^2 + \|z_k - w_{z,k+1}\|^2 - \|w_{z,k}^* - w_{z,k+1}\|^2 \right) + O\left( \frac{\beta_k^2}{\rho} \right)\|z_k - w_{z,k+1}\|^2.$$

**Combining (a) and (b).** We take $C \ge 4$. Given that $\alpha_k^{-1} \ll \max(\rho^{-1}, l_{g,1}/\mu)$, we can conclude that

$$\sigma_k \cdot (i) \le -\frac{1}{4\alpha_k}\|x_{k+1} - x_k\|^2 - \frac{\beta_k}{4\rho}(\|y_k - w_{y,k}^*\|^2 + \|y_k - w_{y,k+1}\|^2) - \frac{\beta_k}{\rho}(\|z_k - w_{z,k}^*\|^2 + \|z_k - w_{z,k+1}\|^2)$$

$$+ O\left( l_{g,1}^2\alpha_k + \rho^{-1}\beta_k \right)(\|w_{y,k}^* - w_{y,k+1}\|^2 + \|w_{z,k}^* - w_{z,k+1}\|^2)$$

$$+ O(\rho^{-2})\alpha_k \left( \mathbf{dist}^2(z_k, T(x_k, 0)) + \|z_k - z_{k+1}\|^2 \right) + \alpha_k\|\widetilde{G} - G\|^2. \tag{36}$$

### F.3 PROOF OF THEOREM C.1

Note that

$$\|x_{k+1} - \hat{x}_k\|^2 = \|\Pi_{\mathcal{X}}\left\{x_k - \alpha_k \widetilde{G}\right\} - \Pi_{\mathcal{X}}\left\{x_k - \alpha_k G^*\right\}\|^2 \leq \alpha_k^2 \|\widetilde{G} - G^*\|^2$$
$$\leq O(l_{g,1}^2)\alpha_k^2(\|w_{y,k}^* - w_{y,k+1}\|^2 + \|w_{z,k}^* - w_{z,k+1}\|^2) + 2\alpha_k^2 \mathbb{E}[\|\widetilde{G} - G\|^2],$$

and also note that

$$\|x_k - x_{k+1}\|^2 \geq \frac{1}{2}\|x_k - \hat{x}_k\|^2 - 2\|\hat{x}_k - x_{k+1}\|^2,$$
$$\mathbb{E}[\|\widetilde{G} - G\|^2] \leq \frac{1}{M_k}(\sigma_k^2 \sigma_f^2 + \sigma_g^2).$$

The following lemma is also useful:

**Lemma F.4** *Under Assumption 1, for all $x \in \mathcal{X}, y \in \mathcal{Y}$ and $\sigma \in [0, \delta/C_f]$, we have*

$$\mathbf{dist}(y, T(x, \sigma)) \leq \left(\frac{1}{\mu} + \frac{D_{\mathcal{Y}}}{\delta}\right)\rho^{-1}\left\|y - \mathbf{prox}_{\rho h_\sigma(x, \cdot)}(y)\right\|.$$

*Proof.* This can be shown with a simple algebra:

$$\mathbf{dist}(y, T(x, \sigma)) \leq \frac{1}{\mu} \cdot \rho^{-1}\|y - \mathbf{prox}_{\rho h_\sigma(x, \cdot)}(y)\| \cdot \mathbb{1}\left\{\rho^{-1}\|y - \mathbf{prox}_{\rho h_\sigma(x, \cdot)}(y)\| \leq \delta\right\}$$
$$+ D_{\mathcal{Y}} \cdot \mathbb{1}\left\{\rho^{-1}\|y - \mathbf{prox}_{\rho h_\sigma(x, \cdot)}(y)\| > \delta\right\},$$

and noting that $\mathbb{1}\left\{\rho^{-1}\|y - \mathbf{prox}_{\rho h_\sigma(x, \cdot)}(y)\| > \delta\right\} < \frac{1}{\delta}\left(\rho^{-1}\|y - \mathbf{prox}_{\rho h_\sigma(x, \cdot)}(y)\|\right).$ $\square$

We now combine results in Proposition F.3, Lemma F.2 and Lemma 3.6, we have (omitting expectations):

$$\mathbb{V}_{k+1} - \mathbb{V}_k \leq -\frac{1}{16\sigma_k\alpha_k}\|x_k - \hat{x}_k\|^2 - \frac{1}{8\sigma_k\alpha_k}\|x_k - x_{k+1}\|^2 - \frac{\beta_k}{4\sigma_k\rho}\left(\|y_k - w_{y,k}^*\|^2 + \|z_k - w_{z,k}^*\|^2\right)$$
$$+ O\left(\frac{\sigma_k - \sigma_{k+1}}{\sigma_k\sigma_{k+1}}\right)\rho^{-1}(\mathbf{dist}^2(y_k, T(x_k, \sigma_k)) + \mathbf{dist}^2(z_k, T(x_k, 0)))$$
$$+ O\left(\frac{\alpha_k}{\rho^2\sigma_k}\right)\mathbf{dist}^2(z_k, T(x_k, 0)) + O\left(\frac{\sigma_k - \sigma_{k+1}}{\sigma_k}\right)C_f$$
$$+ O\left(\frac{\sigma_k - \sigma_{k+1}}{\sigma_k\sigma_{k+1}}\right)\rho^{-1}\left(\|y_k - y_{k+1}\|^2 + \|z_k - z_{k+1}\|^2 + \frac{l_{g,1}^2}{\mu^2}\|x_k - x_{k+1}\|^2 + \frac{l_{f,0}^2}{\mu^2}|\sigma_k - \sigma_{k+1}|^2\right)$$
$$+ \frac{O(1 + l_{g,1}/\mu + C_w\rho l_{g,1})}{\sigma_k\rho}\|x_{k+1} - x_k\|^2 + \frac{O(C_w\rho l_{f,0}^2)}{\sigma_k}|\sigma_k - \sigma_{k+1}|^2$$
$$- \frac{1}{\sigma_k\rho}\left(\frac{1}{4\beta_k} - 16C_w - \rho^{-1}\alpha_k\right)\left(\|y_k - y_{k+1}\|^2 + \|z_k - z_{k+1}\|^2\right)$$
$$+ \frac{C_w\lambda_k}{\sigma_k\rho}\left(1 + \frac{\sigma_k - \sigma_{k+1}}{\sigma_{k+1}} + \frac{\lambda_k}{4} + \frac{O(l_{g,1}^2)\rho\alpha_k + 2\beta_k}{C_w\lambda_k}\right)\left(\|w_{y,k}^* - w_{y,k+1}\|^2 + \|w_{z,k}^* - w_{z,k+1}\|^2\right)$$
$$- \frac{C_w\lambda_k}{\sigma_k\rho}\left(\|w_{y,k}^* - w_{y,k}\|^2 + \|w_{z,k}^* - w_{z,k}\|^2\right) + \frac{2\alpha_k}{\sigma_k}\|\widetilde{G} - G\|^2.$$

Note that $w_{y,k}^* = \mathbf{prox}_{\rho h_{\sigma_k}(x_k, \cdot)}(y_k)$ and $w_{z,k}^* = \mathbf{prox}_{\rho g(x_k, \cdot)}(z_k)$. Using Lemma F.4 and rearranging the terms in the above inequality, we obtain that

$$\mathbb{V}_{k+1} - \mathbb{V}_k \leq -\frac{1}{16\sigma_k\alpha_k}\|x_k - \hat{x}_k\|^2 + \left(\frac{O(1 + l_{g,1}/\mu + C_w\rho l_{g,1})}{\sigma_k\rho} - \frac{1}{8\sigma_k\alpha_k} + \frac{d_k l_{g,1}^2}{\sigma_k\rho\mu^2}\right)\|x_k - x_{k+1}\|^2$$

$$+\left(-\frac{\beta_k}{4\sigma_k\rho}+\frac{d_kC_\delta}{\rho\sigma_k}\right)\|y_k-w_{y,k}^*\|^2+\left(\frac{l_{f,0}^2}{\mu^2}+\frac{O(C_w\rho l_{f,0}^2)}{\sigma_k}\right)|\sigma_k-\sigma_{k+1}|^2$$

$$+\left(-\frac{\beta_k}{4\sigma_k\rho}+\frac{d_kC_\delta}{\rho\sigma_k}+C_\delta O\left(\frac{\alpha_k}{\rho^2\sigma_k}\right)\right)\|z_k-w_{z,k}^*\|^2+d_kC_f$$

$$+\left(\frac{d_k}{\sigma_k\rho}-\frac{1}{\sigma_k\rho}\left(\frac{1}{4\beta_k}-16C_w-\rho^{-1}\alpha_k\right)\right)\left(\|y_k-y_{k+1}\|^2+\|z_k-z_{k+1}\|^2\right)$$

$$+\frac{C_w\lambda_k}{\sigma_k\rho}\left(1+\frac{\sigma_k-\sigma_{k+1}}{\sigma_{k+1}}+\frac{\lambda_k}{4}+\frac{O(l_{g,1}^2)\rho\alpha_k+2\beta_k}{C_w\lambda_k}\right)\left(\|w_{y,k}^*-w_{y,k+1}\|^2+\|w_{z,k}^*-w_{z,k+1}\|^2\right)$$

$$-\frac{C_w\lambda_k}{\sigma_k\rho}\left(\|w_{y,k}^*-w_{y,k}\|^2+\|w_{z,k}^*-w_{z,k}\|^2\right)+\frac{2\alpha_k}{\sigma_k}\|\widetilde{G}-G\|^2.$$

where $d_k=O\left(\frac{\sigma_k-\sigma_{k+1}}{\sigma_{k+1}}\right)$, and $C_\delta=\left(\frac{1}{\mu}+\frac{D_\mathcal{Y}}{\delta}\right)^2\rho^{-2}$.

We state several step-size conditions to keep target quantities to be bounded via telescope sum.

1. To keep the $\|x_k-x_{k+1}\|^2$ term negative, we need $\alpha_k\ll\rho(1+l_{g,1}/\mu+C_w\rho l_{g,1})^{-1}$.

2. To keep $\|y_k-w_{y,k}^*\|^2$ term negative, along with Lemma F.4, we require

$$\beta_k\gg\left(\frac{\sigma_k-\sigma_{k+1}}{\sigma_{k+1}}\right)\rho^{-2}\left(\mu^{-2}+D_\mathcal{Y}^2/\delta^2\right).$$

3. To keep $\|z_k-w_{z,k}^*\|^2$ term negative, we additionally require

$$\beta_k\gg\rho^{-3}\left(\mu^{-2}+D_\mathcal{Y}^2/\delta^2\right)\alpha_k.$$

4. To keep terms on $\|y_k-y_{k+1}\|^2$ and $\|z_k-z_{k+1}\|^2$ negative, we first require

$$\beta_k\ll C_w,\rho\alpha_k^{-1},$$

and then

$$\frac{1}{\beta_k}\gg\frac{\sigma_k-\sigma_{k+1}}{\sigma_{k+1}},$$

which trivially holds as $\beta_k=o(1)$ and $(\sigma_k-\sigma_{k+1}/\sigma_k)=O(1/k)$.

Once the above are satisfied, we get

$$\mathbb{V}_{k+1}-\mathbb{V}_k\leq-\frac{\alpha_k}{16\sigma_k}\|\Delta_k^x\|^2-\frac{\beta_k}{16\sigma_k\rho}\left(\|y_k-w_{y,k}^*\|^2+\|z_k-w_{z,k}^*\|^2\right)$$

$$-\frac{1}{16\sigma_k\beta_k\rho}(\|y_k-y_{k+1}\|^2+\|z_k-z_{k+1}\|^2)$$

$$+O\left(\frac{1+l_{g,1}/\mu+C_w\rho l_{g,1}}{\sigma_k\rho}\right)\frac{(\alpha_k^2+\alpha_k\rho)}{M_k}\cdot(\sigma_k^2\sigma_f^2+\sigma_g^2)+\frac{(\sigma_k-\sigma_{k+1})}{\sigma_k}\cdot O(C_f)$$

$$+\frac{C_w\lambda_k}{\sigma_k\rho}\left(1+\frac{\sigma_k-\sigma_{k+1}}{\sigma_{k+1}}+\frac{\lambda_k}{4}+\frac{O(l_{g,1}^2)\rho\alpha_k+2\beta_k}{C_w\lambda_k}\right)\left(\|w_{y,k}^*-w_{y,k+1}\|^2+\|w_{z,k}^*-w_{z,k+1}\|^2\right)$$

$$-\frac{C_w\lambda_k}{\sigma_k\rho}\left(\|w_{y,k}^*-w_{y,k}\|^2+\|w_{z,k}^*-w_{z,k}\|^2\right)+o(1/k),\tag{37}$$

where $o(1/k)$-term collectively represents the terms asymptotically smaller than $\frac{\sigma_k-\sigma_{k+1}}{\sigma_k}=O(1/k)$ since we use polynomially decaying penalty parameters $\{\sigma_k\}$. Now we can apply Lemma F.1, and plug $\lambda_k=\frac{T_k\gamma_k}{4\rho}$, and using the step-size condition:

$$\frac{\sigma_k-\sigma_{k+1}}{\sigma_{k+1}}\ll\lambda_k,\ \max\left(\rho l_{g,1}^2\alpha_k,\beta_k\right)\ll C_w\lambda_k^2,$$

we get

$$
\mathbb{V}_{k+1} - \mathbb{V}_k \leq -\frac{\alpha_k}{16\sigma_k}\|\Delta_k^x\|^2 - \frac{\beta_k}{16\sigma_k\rho}\left(\|y_k - w_{y,k}^*\|^2 + \|z_k - w_{z,k}^*\|^2\right)
$$
$$
+ O\left(\frac{1 + l_{g,1}/\mu + C_w\rho l_{g,1}}{\rho}\right)\frac{\alpha_k}{\sigma_k M_k}(\sigma_k^2\sigma_f^2 + \sigma_g^2) + O\left(\frac{C_w}{\rho^2}\right)\frac{T_k^2\gamma_k^3}{\sigma_k}(\sigma_k^2\sigma_f^2 + \sigma_g^2).
$$
$$(38)$$

Arranging terms and sum over $k = 0$ to $K - 1$, we have

$$
\mathbb{E}\left[\sum_{k=0}^{K-1}\frac{\alpha_k}{16\sigma_k}\|\Delta_k^x\|^2 + \frac{\rho\beta_k}{16\sigma_k}(\|\Delta_k^y\|^2 + \|\Delta_k^z\|^2)\right]
$$
$$
\leq (\mathbb{V}_0 - \mathbb{V}_K) + O(C_f)\cdot\sum_{k=0}^{K-1}\left(\frac{\sigma_k - \sigma_{k+1}}{\sigma_{k+1}}\right)
$$
$$
+ \frac{O(l_{g,1}/\mu + C_w)}{\rho}\left(\sum_{k=0}^{K-1}\sigma_k^{-1}\left(\frac{\alpha_k}{M_k} + \rho^{-1}T_k^2\gamma_k^3\right)(\sigma_k^2\sigma_f^2 + \sigma_g^2)\right).
$$

### F.4 Proof of Corollary C.2

The remaining part is to show that $\mathbb{V}_K$ is lower-bounded by $O(1)$, and to check the rates. To see this, recall our definition in (14), and note that

$$
\mathbb{V}_K \geq \frac{h_{\sigma,\rho}^*(x,y) - g^*(x)}{\sigma},
$$

as long as $C \geq 1$. Then,

$$
h_{\sigma,\rho}^*(x,y) \geq \sigma f(x,w_{y,k}^*) + g(x,w_{y,k}^*) \geq \sigma f(x,w_{y,k}^*) + g^*(x),
$$

and therefore $\mathbb{V}_K \geq f(x,w_{y,k}^*) > -C_f$ by Assumption 3 on the lower bounded value of $f$.

Now, since $\sigma_k = k^{-s}$ for some $s > 0$, we know that

$$
\frac{\sigma_k - \sigma_{k+1}}{\sigma_{k+1}} = O(1/k),
$$

and thus,

$$
\mathbb{E}\left[\sum_{k=0}^{K-1}\frac{\alpha_k}{16\sigma_k}\|\Delta_k^x\|^2 + \frac{\rho\beta_k}{16\sigma_k}(\|\Delta_k^y\|^2 + \|\Delta_k^z\|^2)\right]
$$
$$
\leq O(\log K) + O\left(\sum_{k=0}^{K-1}\sigma_k^{-1}\left(\frac{\alpha_k}{M_k} + \rho^{-1}T_k^2\gamma_k^3\right)(\sigma_k^2\sigma_f^2 + \sigma_g^2)\right).
$$

Plugging the step-size rates, the right-hand side is bounded by $O(\log K)$, and thus we get the corollary.

## Appendix G   Analysis for Algorithm 2

In addition to descent lemmas for $w_{y,k}$ and $w_{z,k}$, we also need descent lemmas for noise variances in momentum-assisted gradient estimators. We define the outer-variable gradient estimators as the following:

$$
\widetilde{f}_x^k := \nabla_x f(x_k, w_{y,k+1}; \zeta_x^k) + (1 - \eta_k)\left(\widetilde{f}_x^{k-1} - \nabla_x f(x_{k-1}, w_{y,k}; \zeta_x^k)\right),
$$
$$
\widetilde{g}_{xy}^k := \nabla_x g(x_k, w_{y,k+1}; \xi_{xy}^k) + (1 - \eta_k)\left(\widetilde{g}_{xy}^{k-1} - \nabla_x g(x_{k-1}, w_{y,k}; \xi_{xy}^k)\right),
$$
$$
\widetilde{g}_{xz}^k := \nabla_x g(x_k, w_{y,k+1}; \xi_{xz}^k) + (1 - \eta_k)\left(\widetilde{g}_{xz}^{k-1} - \nabla_x g(x_{k-1}, w_{z,k}; \xi_{xz}^k)\right),
$$

### G.1 DESCENT LEMMA FOR NOISE-VARIANCES

We first show that noise-variances for $e_{wy}^k$ and $e_{wz}^k$ decay.

**Lemma G.1** *At every $k^{th}$ iteration, the following holds:*

$$\mathbb{E}[\|e_{wz}^{k+1}\|^2] \leq (1 - \eta_{k+1})^2 \mathbb{E}[\|e_{wz}^k\|^2] + 2\eta_{k+1}^2 \sigma_g^2$$
$$+ O(l_{g,1}^2) \left( \|x_{k+1} - x_k\|^2 + \|w_{z,k+1} - w_{z,k}\|^2 \right),$$

*and similarly,*

$$\mathbb{E}[\|e_{wy}^{k+1}\|^2] \leq (1 - \eta_{k+1})^2 \mathbb{E}[\|e_{wy}^k\|^2] + 4\eta_{k+1}^2 (\sigma_k^2 \sigma_f^2 + \sigma_g^2) + 2(\sigma_k - \sigma_{k+1})^2 \sigma_f^2$$
$$+ O(l_{g,1}^2) \left( \|x_{k+1} - x_k\|^2 + \|w_{y,k+1} - w_{y,k}\|^2 \right).$$

*Proof.* We start with

$$\mathbb{E}\left[\|e_{wz}^{k+1}\|^2\right] = \mathbb{E}\left[\|\widetilde{g}_{wz}^{k+1} - G_{wz}^{k+1}\|^2\right]$$
$$= \mathbb{E}\left[\|(\nabla_y g(x_{k+1}, w_{z,k+1}; \xi_{wz}^{k+1}) - G_{wz}^{k+1}) + (1 - \eta_{k+1})(\widetilde{g}_{wz}^k - \nabla_y g(x_k, w_{z,k}; \xi_{wz}^{k+1}))\|^2\right]$$
$$= (1 - \eta_{k+1})^2 \mathbb{E}[\|e_{wz}^k\|^2]$$
$$+ \mathbb{E}\left[\|(\nabla_y g(x_{k+1}, w_{z,k+1}; \xi_{wz}^{k+1}) - G_{wz}^{k+1}) + (1 - \eta_{k+1})(G_{wz}^k - \nabla_y g(x_k, w_{z,k}; \xi_{wz}^{k+1}))\|^2\right],$$

where the inequality holds since gradient-oracles are unbiased:

$$\mathbb{E}[\langle e_{wz}^k, \nabla_y g(x_{k+1}, w_{z,k+1}; \xi_{wz}^{k+1}) - G_{wz}^{k+1}\rangle | \mathcal{F}_{k+1}] = 0,$$
$$\mathbb{E}[\langle e_{wz}^k, \nabla_y g(x_k, w_{z,k}; \xi_{wz}^{k+1}) - G_{wz}^k\rangle | \mathcal{F}_{k+1}] = 0.$$

The remaining part is to bound

$$\mathbb{E}\left[\|(\nabla_y g(x_{k+1}, w_{z,k+1}; \xi_{wz}^{k+1}) - G_{wz}^{k+1}) + (1 - \eta_{k+1})(G_{wz}^k - \nabla_y g(x_k, w_{z,k}; \xi_{wz}^{k+1}))\|^2\right]$$
$$\leq 2\eta_{k+1}^2 \mathbb{E}[\|\nabla_y g(x_{k+1}, w_{z,k+1}; \xi_{wz}^{k+1}) - G_{wz}^{k+1}\|^2]$$
$$+ 2(1 - \eta_{k+1})^2 \mathbb{E}\left[\|(\nabla_y g(x_{k+1}, w_{z,k+1}; \xi_{wz}^{k+1}) - \nabla_y g(x_k, w_{z,k}; \xi_{wz}^{k+1})) + (G_{wz}^k - G_{wz}^{k+1})\|^2\right]$$
$$\leq 2\eta_{k+1}^2 \sigma_g^2 + O(l_{g,1}^2) \left( \|x_{k+1} - x_k\|^2 + \|w_{z,k+1} - w_{z,k}\|^2 \right).$$

Similarly, we can show the similar result for $e_{wy}^k$. Let $\bar{G}_{wy}^k = \sigma_{k+1} \nabla_y f(x_k, w_{y,k}) + \nabla_y g(x_k, w_{y,k})$, and we have that

$$\mathbb{E}\left[\|e_{wy}^{k+1}\|^2\right] = \mathbb{E}\left[\|\sigma_{k+1}\widetilde{f}_{wy}^{k+1} + \widetilde{g}_{wy}^{k+1} - G_{wy}^{k+1}\|^2\right]$$
$$= (1 - \eta_{k+1})^2 \mathbb{E}[\|e_{wy}^k\|^2] + 2(\sigma_k - \sigma_{k+1})^2 \mathbb{E}[\|\nabla_y f(x_k, w_{y,k}; \xi_{wy}^{k+1}) - \nabla_y f(x_k, w_{y,k})\|^2]$$
$$+ 2\mathbb{E}\left[\|(\nabla_y h_{\sigma_{k+1}}(x_{k+1}, w_{y,k+1}; \xi_{wy}^{k+1}) - \nabla_y h_{\sigma_{k+1}}(x_k, w_{y,k}; \xi_{wy}^{k+1})) + (1 - \eta_{k+1})(G_{wy}^{k+1} - \bar{G}_{wy}^k)\|^2\right]$$
$$\leq (1 - \eta_{k+1})^2 \mathbb{E}[\|e_{wy}^k\|^2] + 2(\sigma_k - \sigma_{k+1})^2 \sigma_f^2 + 4\eta_{k+1}^2 (\sigma_k^2 \sigma_f^2 + \sigma_g^2)$$
$$+ O(l_{g,1}^2) \left( \|x_{k+1} - x_k\|^2 + \|w_{y,k+1} - w_{y,k}\|^2 \right),$$

where we used Assumption 11 to bound

$$\mathbb{E}[\|(\nabla_y h_{\sigma_{k+1}}(x_{k+1}, w_{y,k+1}; \xi_{wy}^{k+1}) - \nabla_y h_{\sigma_{k+1}}(x_k, w_{y,k}; \xi_{wy}^{k+1}))\|^2]$$
$$\leq O(l_{g,1}^2)(\|x_{k+1} - x_k\|^2 + \|w_{y,k+1} - w_{y,k}\|^2).$$

$\square$

We can state a similar descent lemma for $e_x^k$:

**Lemma G.2** *At every $k^{th}$ iteration, the following holds:*

$$\mathbb{E}[\|e_x^{k+1}\|^2] \leq (1 - \eta_{k+1})^2 \mathbb{E}[\|e_x^k\|^2] + O(\eta_{k+1}^2)(\sigma_k^2 \sigma_f^2 + \sigma_g^2) + O(\sigma_{k+1} - \sigma_k)^2 \sigma_f^2 + O(l_{g,1}^2)\|x_{k+1} - x_k\|^2$$
$$+ O(l_{g,1}^2) \left( \|w_{y,k+2} - w_{y,k+1}\|^2 + \|w_{z,k+2} - w_{z,k+1}\|^2 \right).$$

The proof follows exactly the same procedure for bounding $e_{wy}^{k+1}$, and thus we omit the proof.

## G.2 Descent Lemma for $w_{y,k}$, $w_{z,k}$

The strategy is again to start with (26):

$$\|w_{y,k+1} - w^*_{y,k+1}\|^2 = \|w_{y,k+1} - w^*_{y,k}\|^2 + \|w^*_{y,k+1} - w^*_{y,k}\|^2 - 2\langle w_{y,k+1} - w^*_{y,k}, w^*_{y,k+1} - w^*_{y,k}\rangle$$

$$\leq \left(1 + \frac{\lambda_k}{4}\right) \underbrace{\|w_{y,k+1} - w^*_{y,k}\|^2}_{(i)} + \left(1 + \frac{4}{\lambda_k}\right) \underbrace{\|w^*_{y,k+1} - w^*_{y,k}\|^2}_{(ii)},$$

For bounding $(ii)$, we can recall Lemma F.2. For bounding $(i)$, we can slightly modify Lemma F.1.

**Lemma G.3** *At every $k^{th}$ iteration, the following holds:*

$$\mathbb{E}[\|w_{y,k+1} - w^*_{y,k}\|^2|\mathcal{F}_k] \leq \left(1 - \frac{\gamma_k}{4\rho}\right) \mathbb{E}[\|w_{y,k} - w^*_{y,k}\|^2|\mathcal{F}_k] + O(\gamma_k\rho)\mathbb{E}[\|e^k_{wy}\|^2|\mathcal{F}_k]. \quad (39)$$

*Similarly, we also have that*

$$\mathbb{E}[\|w_{z,k+1} - w^*_{z,k}\|^2|\mathcal{F}_k] \leq \left(1 - \frac{\gamma_k}{4\rho}\right) \mathbb{E}[\|w_{z,k} - w^*_{z,k}\|^2|\mathcal{F}_k] + O(\gamma_k\rho)\mathbb{E}[\|e^k_{wz}\|^2|\mathcal{F}_k], \quad (40)$$

*Proof.* We use the linear convergence of projected gradient steps. To simplify the notation, let $\widetilde{G} = \sigma_k\widetilde{f}^k_{wy} + \widetilde{g}^k_{wy} + \rho^{-1}(w_{y,k} - y_k)$ and $G = \nabla_y h_{\sigma_k}(x_k, w_{y,k}) + \rho^{-1}(w_{y,k} - y_k)$. Also let $G^* = \nabla h_{\sigma_k}(x, w^*_{y,k}) + \rho^{-1}(w^*_{y,k} - y_k)$. We first check that

$$\|w_{y,k+1} - w^*_{y,k}\|^2 = \left\|\Pi_{\mathcal{Y}}\left\{w_{y,k} - \gamma_k\widetilde{G}\right\} - \Pi_{\mathcal{Y}}\left\{w^*_{y,k} - \gamma_kG^*\right\}\right\|^2$$

$$\leq \left\|w_{y,k} - \gamma_k\widetilde{G} - (w^*_{y,k} - \gamma_kG^*)\right\|^2$$

$$= \left\|w_{y,k} - w^*_{y,k}\right\|^2 + \gamma_k^2\left\|\widetilde{G} - G^*\right\|^2 - 2\gamma_k\langle w_{y,k} - w^*_{y,k}, \widetilde{G} - G^*\rangle.$$

Taking expectation conditioned on $\mathcal{F}_k$ yields:

$$\mathbb{E}[\|w_{y,k+1} - w^*_{y,k}\|^2|\mathcal{F}_k] \leq \mathbb{E}[\|w_{y,k} - w^*_{y,k}\|^2|\mathcal{F}_k] + \gamma_k^2\mathbb{E}[\|\widetilde{G} - G^*\|^2|\mathcal{F}_k]$$

$$- 2\gamma_k\langle w_{y,k} - w^*_{y,k}, G - G^*\rangle - 2\gamma_k\mathbb{E}[\langle w_{y,k} - w^*_{y,k}, G - \widetilde{G}\rangle|\mathcal{F}_k].$$

Note that we have

$$\mathbb{E}[\|\widetilde{G} - G^*\|^2|\mathcal{F}_k] \leq 2\|G - G^*\|^2 + 2\mathbb{E}[\|\widetilde{G} - G\|^2|\mathcal{F}_k],$$

$$\gamma_k\mathbb{E}[|\langle w_{y,k} - w^*_{y,k}, G - \widetilde{G}\rangle||\mathcal{F}_k] \leq \frac{\gamma_k}{8\rho}\|w_{y,k} - w^*_{y,k}\|^2 + (2\gamma_k\rho)\mathbb{E}[\|\widetilde{G} - G\|^2|\mathcal{F}_k],$$

Now again using the co-coercivity of strongly convex function, since the inner minimization is $(1/(3\rho))$-strongly convex and $(1/\rho)$-smooth, we have

$$\|G - G^*\|^2 \leq (1/\rho) \cdot \langle w_{y,k} - w^*_{y,k}, G - G^*\rangle,$$

$$\frac{1}{3\rho} \cdot \|w_{y,k} - w^*_{y,k}\|^2 \leq \langle w_{y,k} - w^*_{y,k}, G - G^*\rangle.$$

Given $\gamma_k \ll \rho$, and noting that $\widetilde{G} - G = e^k_{wy}$, we have

$$\mathbb{E}[\|w_{y,k+1} - w^*_{y,k}\|^2] \leq \left(1 - \frac{\gamma_k}{4\rho}\right) \mathbb{E}[\|w_{y,k} - w^*_{y,k}\|^2] + O(\gamma_k\rho)\mathbb{E}[\|e^k_{wy}\|^2].$$

Similar arguments can show the bound on $\|w_{z,k+1} - w^*_{z,k}\|$. $\qquad\square$

In addition to the contraction of $w_{y,k}$ toward the proximal operators, we will also use the following on the bounds on expected movements:

**Lemma G.4** *At every $k^{th}$ iteration, the following holds:*

$$\frac{1}{2\gamma_k}\mathbb{E}[\|w_{y,k+1} - w_{y,k}\|^2] \leq \mathbb{E}[h_{\sigma_k}(x_k, y_k, w_{y,k}) - h_{\sigma_k}(x_k, y_k, w_{y,k+1}) + 4\gamma_k\|e^k_{wy}\|^2]. \quad (41)$$

*Similarly for $w_{z,k}$, we have*

$$\frac{1}{2\gamma_k}\mathbb{E}[\|w_{z,k+1} - w_{z,k}\|^2] \leq \mathbb{E}[g(x_k, z_k, w_{z,k}) - g(x_k, z_k, w_{z,k+1}) + 4\gamma_k\|e^k_{wz}\|^2]. \quad (42)$$

*Proof.* By $2\rho^{-1}$-smoothness of $h_{\sigma_k}(x, y, w)$, we have

$$h_{\sigma_k}(x_k, y_k, w_{y,k+1}) \le h_{\sigma_k}(x_k, y_k, w_{y,k}) + \langle \nabla_w h_{\sigma_k}(x_k, y_k, w_{y,k}), w_{y,k+1} - w_{y,k} \rangle + \frac{1}{\rho}\|w_{y,k+1} - w_{y,k}\|^2.$$

Let $\bar{w}_k = w_{y,k} - \gamma_k(\nabla_w h_{\sigma_k}(x_k, y_k, w_{y,k}) + e_{wy}^k)$, and thus $\nabla_w h_{\sigma_k}(x_k, y_k, w_{y,k}) = \frac{1}{\gamma_k}(w_{y,k} - \bar{w}_k) - e_{wy}^k$. Plugging this back, we have

$$
\begin{aligned}
h_{\sigma_k}(x_k, y_k, w_{y,k+1}) &\le h_{\sigma_k}(x_k, y_k, w_{y,k}) + \gamma_k^{-1}\langle w_{y,k} - \bar{w}_k, w_{y,k+1} - w_{y,k} \rangle \\
&\quad - \langle e_{wy}^k, w_{y,k+1} - w_{y,k} \rangle + \frac{1}{\rho}\|w_{y,k+1} - w_{y,k}\|^2 \\
&\le h_{\sigma_k}(x_k, y_k, w_{y,k}) - \gamma_k^{-1}\|w_{y,k} - w_{y,k+1}\|^2 + \gamma_k^{-1}\langle w_{y,k+1} - \bar{w}_k, w_{y,k+1} - w_{y,k} \rangle \\
&\quad + 4\gamma_k\|e_{wy}^k\|^2 + \frac{1}{16\gamma_k}\|w_{y,k+1} - w_{y,k}\|^2 + \frac{1}{\rho}\|w_{y,k+1} - w_{y,k}\|^2.
\end{aligned}
$$

By projection lemma, $\langle w_{y,k+1} - \bar{w}_k, w_{y,k+1} - w_{y,k} \rangle \le 0$, and since $\gamma_k \ll \rho$, we have

$$h_{\sigma_k}(x_k, y_k, w_{y,k+1}) \le h_{\sigma_k}(x_k, y_k, w_{y,k}) - \frac{1}{2\gamma_k}\|w_{y,k+1} - w_{y,k}\|^2 + 4\gamma_k\|e_{wy}^k\|^2.$$

Arranging this, we get (41). (42) can be obtained similarly, and hence we omit the details. $\qquad\square$

### G.3 DESCENT LEMMA FOR $\Phi_{\sigma,\rho}$

For simplicity, let $G = G_x^k = \nabla_x(\sigma_k f(x_k, w_{y,k+1}) + g(x_k, w_{y,k+1}) - g(x_k, w_{z,k+1}))$ and $\widetilde{G} = e_x^k + G_x^k$. This part follows exactly the same as Appendix F.2, yielding the similar result to (36):

$$
\begin{aligned}
\sigma_k \cdot (i) &\le -\frac{1}{4\alpha_k}\|x_{k+1} - x_k\|^2 - \frac{\beta_k}{4\rho}(\|y_k - w_{y,k}^*\|^2 + \|y_k - w_{y,k+1}\|^2) - \frac{\beta_k}{\rho}(\|z_k - w_{z,k}^*\|^2 + \|z_k - w_{z,k+1}\|^2) \\
&\quad + O\left(l_{g,1}^2 \alpha_k + \rho^{-1}\beta_k\right)\left(\|w_{y,k}^* - w_{y,k+1}\|^2 + \|w_{z,k}^* - w_{z,k+1}\|^2\right) \\
&\quad + O(\rho^{-2})\alpha_k\left(\mathbf{dist}^2(z_k, T(x_k, 0)) + \|z_k - z_{k+1}\|^2\right) + \alpha_k\|\widetilde{G} - G\|^2.
\end{aligned}
$$

### G.4 DESCENT IN POTENTIALS

Note again that

$$
\begin{aligned}
\mathbb{E}[\|x_{k+1} - \hat{x}_k\|^2] &= \mathbb{E}\left[\left\|\Pi_{\mathcal{X}}\left\{x_k - \alpha_k\widetilde{G}\right\} - \Pi_{\mathcal{X}}\left\{x_k - \alpha_k G^*\right\}\right\|^2\right] \le \alpha_k^2\|\widetilde{G} - G^*\|^2 \\
&\le O(l_{g,1}^2)\alpha_k^2(\|w_{y,k}^* - w_{y,k+1}\|^2 + \|w_{z,k}^* - w_{z,k+1}\|^2) + 2\alpha_k^2\mathbb{E}[\|\widetilde{G} - G\|^2],
\end{aligned}
$$

and also note that

$$
\begin{aligned}
\|x_k - x_{k+1}\|^2 &\ge \frac{1}{2}\|x_k - \hat{x}_k\|^2 - 2\|\hat{x}_k - x_{k+1}\|^2, \\
\mathbb{E}[\|\widetilde{G} - G\|^2] &= \mathbb{E}[\|e_x^k\|^2].
\end{aligned}
$$

Similarly to the proof of Appendix F.3, using Lemma F.2, (32), (36), and Lemma F.4, and using the step-size conditions, we obtain a similar inequality to (37), with extra terms on noise-variances:

$$
\begin{aligned}
\mathbb{V}_{k+1} - \mathbb{V}_k &\le -\frac{\alpha_k}{16\sigma_k}\|\Delta_k^x\|^2 - \frac{\beta_k}{16\sigma_k\rho}\left(\|y_k - w_{y,k}^*\|^2 + \|z_k - w_{z,k}^*\|^2\right) \\
&\quad - \frac{1}{16\sigma_k\beta_k\rho}(\|y_k - y_{k+1}\|^2 + \|z_k - z_{k+1}\|^2) + \frac{(\sigma_k - \sigma_{k+1})}{\sigma_k} \cdot O(C_f) \\
&\quad + \frac{C_w}{\sigma_k\rho}\left(1 + \frac{\sigma_k - \sigma_{k+1}}{\sigma_{k+1}} + \frac{\lambda_k}{4} + \frac{O(l_{g,1}^2)\rho\alpha_k + 2\beta_k}{C_w}\right)\left(\|w_{y,k}^* - w_{y,k+1}\|^2 + \|w_{z,k}^* - w_{z,k+1}\|^2\right) \\
&\quad - \frac{C_w}{\sigma_k\rho}\left(\|w_{y,k}^* - w_{y,k}\|^2 + \|w_{z,k}^* - w_{z,k}\|^2\right) - \frac{1}{16\sigma_k\alpha_k}\|x_k - x_{k+1}\|^2 + o(1/k)
\end{aligned}
$$

$$+ (C_\eta \rho^2) \underbrace{\left( \frac{1}{\sigma_{k+1}\gamma_k} \|e_x^k\|^2 - \frac{1}{\sigma_k \gamma_{k-1}} \|e_x^{k-1}\|^2 \right) + O\left( \frac{1 + l_{g,1}/\mu + C_w \rho l_{g,1}}{\sigma_k} \right)(\alpha_k + \rho^{-1}\alpha_k^2) \cdot \|e_x^k\|^2}_{(i)}$$

$$+ (C_\eta \rho^2) \underbrace{\left( \frac{1}{\sigma_{k+1}\gamma_k} \|e_{wy}^{k+1}\|^2 - \frac{1}{\sigma_k \gamma_{k-1}} \|e_{wy}^k\|^2 \right)}_{(ii)}$$

$$+ (C_\eta \rho^2) \underbrace{\left( \frac{1}{\sigma_{k+1}\gamma_k} \|e_{wz}^{k+1}\|^2 - \frac{1}{\sigma_k \gamma_{k-1}} \|e_{wz}^k\|^2 \right)}_{(iii)}.$$

In order to bound $e_x^k$ term, given that

$$\eta_{k+1} \gg \left( \frac{\sigma_k \gamma_{k-1}}{\sigma_{k+1}\gamma_k} - 1 + \frac{l_{g,1}/\mu + C_w}{C_\eta \rho^2} \alpha_k \gamma_{k-1} \right),$$

using Lemma G.2, we have

$$(i) \le -\frac{C_\eta \rho^2 \eta_k}{\sigma_k \gamma_{k-1}} \|e_x^{k-1}\|^2 + C_\eta \rho^2 \cdot \frac{O(\eta_k^2)(\sigma_{k-1}^2 \sigma_f^2 + \sigma_g^2) + O(l_{g,1}^2)\|x_k - x_{k-1}\|^2}{\sigma_k \gamma_{k-1}}$$

$$+ C_\eta \rho^2 \cdot \frac{O(l_{g,1}^2)(\|w_{y,k+1} - w_{y,k}\|^2 + \|w_{z,k+1} - w_{z,k}\|^2)}{\sigma_k \gamma_{k-1}}.$$

Similarly, by Lemma G.1,

$$(ii) \le -\frac{C_\eta \rho^2 \eta_{k+1}}{\sigma_k \gamma_{k-1}} \|e_{wy}^k\|^2 + C_\eta \rho^2 \cdot \frac{O(\eta_{k+1}^2)(\sigma_k^2 \sigma_f^2 + \sigma_g^2) + O(l_{g,1}^2)\|x_{k+1} - x_k\|^2}{\sigma_k \gamma_{k-1}}$$

$$+ C_\eta \rho^2 \cdot \frac{O(l_{g,1}^2)\|w_{y,k+1} - w_{y,k}\|^2}{\sigma_k \gamma_{k-1}},$$

$$(iii) \le -\frac{C_\eta \rho^2 \eta_{k+1}}{\sigma_k \gamma_{k-1}} \|e_{wz}^k\|^2 + C_\eta \rho^2 \cdot \frac{O(\eta_{k+1}^2)\sigma_g^2 + O(l_{g,1}^2)\|x_{k+1} - x_k\|^2}{\sigma_k \gamma_{k-1}}$$

$$+ C_\eta \rho^2 \cdot \frac{O(l_{g,1}^2)\|w_{z,k+1} - w_{z,k}\|^2}{\sigma_k \gamma_{k-1}}.$$

Now setting $C_\eta > 0$ and $\rho \ll 1/l_{g,1}$ properly, with $\alpha_k \ll \gamma_k$, we can keep $\|x_{k+1} - x_k\|^2$ terms negative, and have

$$\mathbb{V}_{k+1} - \mathbb{V}_k \le -\frac{\alpha_k}{16\sigma_k} \|\Delta_k^x\|^2 - \frac{\beta_k}{16\sigma_k \rho} \left( \|y_k - w_{y,k}^*\|^2 + \|z_k - w_{z,k}^*\|^2 \right)$$

$$- \frac{1}{16\sigma_k \beta_k \rho}(\|y_k - y_{k+1}\|^2 + \|z_k - z_{k+1}\|^2) + \frac{(\sigma_k - \sigma_{k+1})}{\sigma_k} \cdot O(C_f)$$

$$+ \frac{C_w}{\sigma_k \rho} \left( 1 + \frac{\sigma_k - \sigma_{k+1}}{\sigma_{k+1}} + \frac{\lambda_k}{4} + \frac{O(l_{g,1}^2)\rho\alpha_k + 2\beta_k}{C_w} \right) \left( \|w_{y,k}^* - w_{y,k+1}\|^2 + \|w_{z,k}^* - w_{z,k+1}\|^2 \right)$$

$$- \frac{C_w}{\sigma_k \rho} \left( \|w_{y,k}^* - w_{y,k}\|^2 + \|w_{z,k}^* - w_{z,k}\|^2 \right) - \frac{1}{32\sigma_k \alpha_k} \|x_k - x_{k+1}\|^2 + \frac{O(\rho^2 l_{g,1}^2)}{\sigma_k \gamma_{k-1}} \|x_k - x_{k-1}\|^2$$

$$- \frac{C_\eta \rho^2 \eta_{k+1}}{\sigma_k \gamma_{k-1}}(\|e_{wy}^k\|^2 + \|e_{wz}^k\|^2) + C_\eta O(\rho^2 l_{g,1}^2) \frac{\|w_{y,k+1} - w_{y,k}\|^2 + \|w_{z,k+1} - w_{z,k}\|^2}{\sigma_k \gamma_{k-1}}$$

$$+ C_\eta \rho^2 \frac{O(\eta_{k+1}^2)}{\sigma_k \gamma_k}(\sigma_k^2 \sigma_f^2 + \sigma_g^2) + o(1/k).$$

Given that

$$\eta_{k+1} \gg O(l_{g,1}^2)\gamma_k^2,$$

using Lemma G.4, we can manipulate $\|w_{y,k+1} - w_{y,k}\|$ and $\|w_{z,k+1} - w_{z,k}\|$ terms to be bounded by

$$
\begin{aligned}
\mathbb{V}_{k+1} - \mathbb{V}_k \leq &-\frac{\alpha_k}{16\sigma_k}\|\Delta_k^x\|^2 - \frac{\beta_k}{16\sigma_k\rho}\left(\|y_k - w_{y,k}^*\|^2 + \|z_k - w_{z,k}^*\|^2\right) \\
&- \frac{1}{16\sigma_k\beta_k\rho}(\|y_k - y_{k+1}\|^2 + \|z_k - z_{k+1}\|^2) + \frac{(\sigma_k - \sigma_{k+1})}{\sigma_k}\cdot O(C_f) \\
&+ \frac{C_w}{\sigma_k\rho}\left(1 + \frac{\sigma_k - \sigma_{k+1}}{\sigma_{k+1}} + \frac{\lambda_k}{4} + \frac{O(l_{g,1}^2)\rho\alpha_k + 2\beta_k}{C_w}\right)\left(\|w_{y,k}^* - w_{y,k+1}\|^2 + \|w_{z,k}^* - w_{z,k+1}\|^2\right) \\
&- \frac{C_w}{\sigma_k\rho}\left(\|w_{y,k}^* - w_{y,k}\|^2 + \|w_{z,k}^* - w_{z,k}\|^2\right) - \frac{1}{32\sigma_k\alpha_k}\|x_k - x_{k+1}\|^2 + \frac{O(\rho^2 l_{g,1}^2)}{\sigma_k\gamma_{k-1}}\|x_k - x_{k-1}\|^2 \\
&- \frac{C_\eta\rho^2\eta_{k+1}}{2\sigma_k\gamma_{k-1}}(\|e_{wy}^k\|^2 + \|e_{wz}^k\|^2) + C_\eta\rho^2\frac{O(\eta_{k+1}^2)}{\sigma_k\gamma_k}(\sigma_k^2\sigma_f^2 + \sigma_g^2) + o(1/k) \\
&+ \frac{C_\eta O(\rho^2 l_{g,1}^2)}{\sigma_k}\left(\underbrace{h_{\sigma_k}(x_k, y_k, w_{y,k}) - h_{\sigma_k}(x_k, y_k, w_{y,k+1})}_{(iv)} + \underbrace{g(x_k, z_k, w_{z,k}) - g(x_k, z_k, w_{z,k+1})}_{(v)}\right).
\end{aligned}
\tag{43}
$$

To proceed, we note that

$$
\begin{aligned}
(iv) &= (h_{\sigma_k}(x_k, y_k, w_{y,k}) - h_{\sigma_k,\rho}^*(x_k, y_k)) - (h_{\sigma_k}(x_k, y_k, w_{y,k+1}) - h_{\sigma_k,\rho}^*(x_k, y_k)) \\
&= (h_{\sigma_k}(x_k, y_k, w_{y,k}) - h_{\sigma_k,\rho}^*(x_k, y_k)) - (h_{\sigma_k}(x_{k+1}, y_{k+1}, w_{y,k+1}) - h_{\sigma_k,\rho}^*(x_{k+1}, y_{k+1})) \\
&\quad + \underbrace{h_{\sigma_k}(x_{k+1}, y_{k+1}, w_{y,k+1}) - h_{\sigma_k,\rho}^*(x_{k+1}, y_{k+1})) - (h_{\sigma_k}(x_k, y_k, w_{y,k+1}) - h_{\sigma_k,\rho}^*(x_k, y_k))}_{(a)},
\end{aligned}
$$

and the term $(a)$ is bounded as

$$
\begin{aligned}
(a) &\leq \langle\nabla_x(h_{\sigma_k}(x_k, y_k, w_{y,k+1}) - h_{\sigma_k,\rho}^*(x_k, y_k)), x_{k+1} - x_k\rangle + l_{g,1}\|x_{k+1} - x_k\|^2 \\
&\quad + \langle\nabla_y(h_{\sigma_k}(x_k, y_k, w_{y,k+1}) - h_{\sigma_k,\rho}^*(x_k, y_k)), y_{k+1} - y_k\rangle + \rho^{-1}\|y_{k+1} - y_k\|^2 \\
&= \langle\nabla_x(h_{\sigma_k}(x_k, w_{y,k+1}) - h_{\sigma_k}(x_k, w_{y,k}^*)), x_{k+1} - x_k\rangle + l_{g,1}\|x_{k+1} - x_k\|^2 \\
&\quad + \rho^{-1}\langle(y_k - w_{y,k+1}) - (y_k - w_{y,k}^*), y_{k+1} - y_k\rangle + \rho^{-1}\|y_{k+1} - y_k\|^2 \\
&\leq 32l_{g,1}^2\alpha_k\|w_{y,k+1} - w_{y,k}^*\|^2 + \frac{1}{128\alpha_k}\|x_{k+1} - x_k\|^2 + \frac{16\beta_k}{\rho}\|w_{y,k+1} - w_{y,k}^*\|^2 + \frac{1}{64\beta_k\rho}\|y_{k+1} - y_k\|^2.
\end{aligned}
$$

Thus, we can conclude that

$$
\begin{aligned}
(iv) &\leq (h_{\sigma_k}(x_k, y_k, w_{y,k}) - h_{\sigma_k,\rho}^*(x_k, y_k)) - (h_{\sigma_k}(x_{k+1}, y_{k+1}, w_{y,k+1}) - h_{\sigma_k,\rho}^*(x_{k+1}, y_{k+1})) \\
&\quad + 32l_{g,1}^2\alpha_k\|w_{y,k+1} - w_{y,k}^*\|^2 + \frac{1}{128\alpha_k}\|x_{k+1} - x_k\|^2 \\
&\quad + \frac{16\beta_k}{\rho}\|w_{y,k+1} - w_{y,k}^*\|^2 + \frac{1}{64\beta_k\rho}\|y_{k+1} - y_k\|^2.
\end{aligned}
$$

Similarly, we have

$$
\begin{aligned}
(v) &\leq (g(x_k, z_k, w_{z,k}) - g_\rho^*(x_k, z_k)) - (g(x_{k+1}, z_{k+1}, w_{z,k+1}) - g_\rho^*(x_{k+1}, z_{k+1})) \\
&\quad + 32l_{g,1}^2\alpha_k\|w_{z,k+1} - w_{z,k}^*\|^2 + \frac{1}{128\alpha_k}\|x_{k+1} - x_k\|^2 \\
&\quad + \frac{16\beta_k}{\rho}\|w_{z,k+1} - w_{z,k}^*\|^2 + \frac{1}{64\beta_k\rho}\|z_{k+1} - z_k\|^2.
\end{aligned}
$$

Now plugging this back, we have reduced (43) to

$$
\begin{aligned}
&\mathbb{V}_{k+1} - \mathbb{V}_k \\
&\leq -\frac{\alpha_k}{16\sigma_k}\|\Delta_k^x\|^2 - \frac{\beta_k}{16\sigma_k\rho}\left(\|y_k - w_{y,k}^*\|^2 + \|z_k - w_{z,k}^*\|^2\right)
\end{aligned}
$$

$$- \frac{1}{32\sigma_k\beta_k\rho}(\|y_k - y_{k+1}\|^2 + \|z_k - z_{k+1}\|^2) + \frac{(\sigma_k - \sigma_{k+1})}{\sigma_k} \cdot O(C_f)$$

$$+ \frac{C_w}{\sigma_k\rho}\left(1 + \frac{\sigma_k - \sigma_{k+1}}{\sigma_{k+1}} + \frac{\lambda_k}{4} + \frac{O(l_{g,1}^2\rho)\alpha_k + O(1)\beta_k}{C_w}\right)(\|w_{y,k}^* - w_{y,k+1}\|^2 + \|w_{z,k}^* - w_{z,k+1}\|^2)$$

$$- \frac{C_w}{\sigma_k\rho}\left(\|w_{y,k}^* - w_{y,k}\|^2 + \|w_{z,k}^* - w_{z,k}\|^2\right) - \frac{1}{64\sigma_k\alpha_k}\|x_k - x_{k+1}\|^2 + \frac{O(\rho^2 l_{g,1}^2)}{\sigma_k\gamma_{k-1}}\|x_k - x_{k-1}\|^2$$

$$- \frac{C_\eta\rho^2\eta_{k+1}}{2\sigma_k\gamma_{k-1}}(\|e_{wy}^k\|^2 + \|e_{wz}^k\|^2) + C_\eta\rho^2\frac{O(\eta_{k+1}^2)}{\sigma_k\gamma_k}(\sigma_k^2\sigma_f^2 + \sigma_g^2) + o(1/k)$$

$$+ C_\eta O(\rho^2 l_{g,1}^2)\left(\frac{h_{\sigma_k}(x_k, y_k, w_{y,k}) - h_{\sigma_k,\rho}^*(x_k, y_k)}{\sigma_k} - \frac{h_{\sigma_k}(x_{k+1}, y_{k+1}, w_{y,k+1}) - h_{\sigma_k,\rho}^*(x_{k+1}, y_{k+1})}{\sigma_k}\right)$$

$$+ C_\eta O(\rho^2 l_{g,1}^2)\left(\frac{g(x_k, z_k, w_{z,k}) - g_\rho^*(x_k, z_k)}{\sigma_k} - \frac{g(x_{k+1}, z_{k+1}, w_{z,k+1}) - g_\rho^*(x_{k+1}, z_{k+1})}{\sigma_k}\right).$$

$$(44)$$

Now applying Lemma G.3, along with

$$\lambda_k = \frac{\gamma_k}{4\rho} \gg \frac{O(l_{g,1}^2\rho)\alpha_k + O(\beta_k)}{C_w},$$

and

$$\eta_{k+1} \gg \frac{C_w}{C_\eta}\frac{\lambda_k\gamma_k}{\rho} \gg \frac{C_w\gamma_k^2}{4C_\eta\rho^2},$$

we can further bound (44) by

$$\mathbb{V}_{k+1} - \mathbb{V}_k$$

$$\leq -\frac{\alpha_k}{16\sigma_k}\|\Delta_k^x\|^2 - \frac{\beta_k}{16\sigma_k\rho}\left(\|y_k - w_{y,k}^*\|^2 + \|z_k - w_{z,k}^*\|^2\right)$$

$$- \frac{1}{32\sigma_k\beta_k\rho}(\|y_k - y_{k+1}\|^2 + \|z_k - z_{k+1}\|^2) - \frac{1}{64\sigma_k\alpha_k}\|x_k - x_{k+1}\|^2 + \frac{O(\rho^2 l_{g,1}^2)}{\sigma_k\gamma_{k-1}}\|x_k - x_{k-1}\|^2$$

$$+ \frac{(\sigma_k - \sigma_{k+1})}{\sigma_k} \cdot O(C_f) + C_\eta\rho^2\frac{O(\eta_{k+1}^2)}{\sigma_k\gamma_k}(\sigma_k^2\sigma_f^2 + \sigma_g^2) + o(1/k)$$

$$- \frac{C_w\lambda_k}{16\sigma_k\rho}(\|w_{y,k}^* - w_{y,k}\|^2 + \|w_{z,k}^* - w_{z,k}\|^2)$$

$$+ C_\eta O(\rho^2 l_{g,1}^2)\left(\frac{h_{\sigma_k}(x_k, y_k, w_{y,k}) - h_{\sigma_k,\rho}^*(x_k, y_k)}{\sigma_k} - \frac{h_{\sigma_k}(x_{k+1}, y_{k+1}, w_{y,k+1}) - h_{\sigma_k,\rho}^*(x_{k+1}, y_{k+1})}{\sigma_k}\right)$$

$$+ C_\eta O(\rho^2 l_{g,1}^2)\left(\frac{g(x_k, z_k, w_{z,k}) - g_\rho^*(x_k, z_k)}{\sigma_k} - \frac{g(x_{k+1}, z_{k+1}, w_{z,k+1}) - g_\rho^*(x_{k+1}, z_{k+1})}{\sigma_k}\right).$$

$$(45)$$

### G.5 Proof of Theorem C.4

Summing the bound (45) for $k = 0$ to $K - 1$, we can cancel out $\|x_k - x_{k+1}\|^2$ terms given that

$$O(\rho^2 l_{g,1}^2)\alpha_k \ll \frac{\sigma_k}{\sigma_{k+1}}\gamma_k.$$

Leaving only relevant terms in the final bound, we get

$$\mathbb{V}_K - \mathbb{V}_0$$

$$\leq \sum_{k=0}^{K-1}\left(-\frac{\alpha_k}{16\sigma_k}\|\Delta_k^x\|^2 - \frac{\beta_k}{16\sigma_k\rho}\left(\|y_k - w_{y,k}^*\|^2 + \|z_k - w_{z,k}^*\|^2\right)\right)$$

$$
+ \sum_{k=0}^{K-1} \left( \frac{(\sigma_k - \sigma_{k+1})}{\sigma_k} \cdot O(C_f) + C_\eta \rho^2 \frac{O(\eta_{k+1}^2)}{\sigma_k \gamma_k} (\sigma_k^2 \sigma_f^2 + \sigma_g^2) \right)
$$

$$
+ \sum_{k=0}^{K-1} \left( -\frac{C_w \lambda_k}{16 \sigma_k \rho} (\|w_{y,k}^* - w_{y,k}\|^2 + \|w_{z,k}^* - w_{z,k}\|^2) \right)
$$

$$
+ \sum_{k=0}^{K-2} \left( \left( \frac{1}{\sigma_{k+1}} - \frac{1}{\sigma_k} \right) \left( h_{\sigma_k}(x_k, y_k, w_{y,k}) - h_{\sigma_k,\rho}^*(x_k, y_k) + g(x_k, z_k, w_{z,k}) - g_\rho^*(x_k, z_k) \right) \right)
$$

$$
+ C_\eta O(\rho^2 l_{g,1}^2) \left( \frac{h_{\sigma_0}(x_0, y_0, w_{y,0}) - h_{\sigma_0,\rho}^*(x_0, y_0)}{\sigma_0} - \frac{h_{\sigma_K}(x_K, y_K, w_{y,K}) - h_{\sigma_K,\rho}^*(x_K, y_K)}{\sigma_{K-1}} \right)
$$

$$
+ C_\eta O(\rho^2 l_{g,1}^2) \left( \frac{g(x_0, z_0, w_{z,0}) - g_\rho^*(x_0, z_0)}{\sigma_0} - \frac{g(x_K, z_K, w_{z,K}) - g_\rho^*(x_K, z_K)}{\sigma_{K-1}} \right),
$$

where the last two lines come from the telescoping sum. Note that

$$
h_{\sigma_K}(x_K, y_K, w_{y,K}) - h_{\sigma_K,\rho}^*(x_K, y_K) \geq 0,
$$
$$
g(x_K, z_K, w_{z,K}) - g_\rho^*(x_K, z_K) \geq 0,
$$

by definition, and furthermore,

$$
h_{\sigma_k}(x_k, y_k, w_{y,k}) - h_{\sigma_k,\rho}^*(x_k, y_k) \leq \langle \nabla_w h_{\sigma_k}(x_k, y_k, w_{y,k}^*), w_{y,k} - w_{y,k}^* \rangle,
$$
$$
g(x_k, z_k, w_{z,k}) - g_\rho^*(x_k, z_k) \leq \langle \nabla_w g(x_k, z_k, w_{z,k}^*), w_{z,k} - w_{z,k}^* \rangle.
$$

Here we rely on Assumption 12 (along with $\mathcal{Y}$ being compact) to bound them:

$$
h_{\sigma_k}(x_k, y_k, w_{y,k}) - h_{\sigma_k,\rho}^*(x_k, y_k) \leq M_w \|w_{y,k} - w_{y,k}^*\|,
$$
$$
g(x_k, z_k, w_{z,k}) - g_\rho^*(x_k, z_k) \leq M_w \|w_{z,k} - w_{z,k}^*\|.
$$

Plugging this back, and using $-ax^2 + bx \leq \frac{b^2}{4a}$, we have

$$
\mathbb{V}_K - \mathbb{V}_0 \leq \sum_{k=0}^{K-1} \left( -\frac{\alpha_k}{16 \sigma_k} \|\Delta_k^x\|^2 - \frac{\beta_k}{16 \sigma_k \rho} \left( \|y_k - w_{y,k}^*\|^2 + \|z_k - w_{z,k}^*\|^2 \right) \right)
$$

$$
+ \sum_{k=0}^{K-1} \left( \frac{(\sigma_k - \sigma_{k+1})}{\sigma_k} \cdot O(C_f) + C_\eta \rho^2 \frac{O(\eta_{k+1}^2)}{\sigma_k \gamma_k} (\sigma_k^2 \sigma_f^2 + \sigma_g^2) \right)
$$

$$
+ \sum_{k=0}^{K-1} \left( \frac{O(M_w^2) \rho^2}{C_w \sigma_k \gamma_k} \left( \frac{\sigma_k - \sigma_{k+1}}{\sigma_{k+1}} \right)^2 \right)
$$

$$
+ C_\eta O(\rho^2 l_{g,1}^2) \left( \frac{h_{\sigma_0}(x_0, y_0, w_{y,0}) - h_{\sigma_0,\rho}^*(x_0, y_0)}{\sigma_0} + \frac{g(x_0, z_0, w_{z,0}) - g_\rho^*(x_0, z_0)}{\sigma_0} \right).
$$

Note that $w_{y,0} = y_0, w_{z,0} = z_0$, and thus $h_{\sigma_0}(x_0, y_0, w_{y,0}) = h_{\sigma_0}(x_0, y_0), g(x_0, z_0, w_{z,0}) = g(x_0, z_0)$. Arranging the terms, we have the theorem.

### G.6    PROOF OF COROLLARY C.5

The proof is almost identical to the proof of Corollary C.2 in Appendix F.4, and thus we omit the proof.

## APPENDIX H    PROOFS OF AUXILIARY LEMMAS

### H.1    PROOF OF LEMMA 3.6

The Lemma essentially follows the proof of Proposition 4 in Shen & Chen (2023). It suffices to show the *local* Lipschitz-continuity of solution-sets. For every $x_1, x_2 \in \mathcal{X}$ such that $\|x_1 - x_2\| \leq c_x \mu \delta / l_{g,1}$

and $|\sigma_1 - \sigma_2| \leq c_s \mu \delta / l_{f,0}$ with sufficiently small constants $c_x, c_s > 0$, let $y_1 \in T(x_1, \sigma_1)$ and $\bar{y} = \mathbf{prox}_{\rho h_{\sigma_2}(x_2, \cdot)}(y_1)$. Note that $y_1 = \mathbf{prox}_{\rho h_{\sigma_1}(x_1, \cdot)}(y_1)$. By Lemma D.3 and Lemma D.4,

$$
\begin{aligned}
\|y_1 - \bar{y}\| &= \|\mathbf{prox}_{\rho h_{\sigma_1}(x_1, \cdot)}(y_1) - \mathbf{prox}_{\rho h_{\sigma_2}(x_2, \cdot)}(y_1)\| \\
&\leq O(\rho l_{g,1})\|x_1 - x_2\| + O(\rho l_{f,0})|\sigma_1 - \sigma_2| \leq \rho \delta.
\end{aligned}
$$

Thus, $\rho^{-1}\|y_1 - \mathbf{prox}_{\rho h_{\sigma_2}(x_2, \cdot)}(y_1)\| \leq \delta$, and applying Assumption 1,

$$
\mathbf{dist}(y_1, T(x_2, \sigma_2)) \leq \mu^{-1} \cdot \left( O(l_{g,1})\|x_1 - x_2\| + O(l_{f,0})|\sigma_1 - \sigma_2| \right),
$$

proving the (local) $O(l_{g,1}/\mu)$-Lipschitz continuity in $x$ and $O(l_{f,0}/\mu)$-Lipscthiz continuity in $\sigma$ of $T(x, \sigma)$.

### H.2 Proof of Lemma B.1

*Proof.* By Danskin's theorem (Theorem D.1), if $\nabla h_\sigma^*(x)$ and $\nabla g^*(x)$ exist, then they are given by

$$
\nabla h_\sigma^*(x) = \nabla_x h_\sigma(x, w_\sigma^*), \ \nabla g^*(x) = \nabla_x g(x, w^*),
$$

for any $w_\sigma^* \in T(x, \sigma), w^* \in T(x, 0)$. Let $w_y^* = \mathbf{prox}_{\rho h_\sigma(x^*, \cdot)}(y^*), w_z^* = \mathbf{prox}_{\rho g(x^*, \cdot)}(z^*)$. By Assumption 1, there exists $w_\sigma^* \in T(x, \sigma)$ that satisfies

$$
\sigma \epsilon \geq \rho^{-1}\|y^* - w_y^*\| \geq \mu\|y^* - w_\sigma^*\|,
$$

and thus $\|y^* - w_\sigma^*\| \leq \frac{\sigma \epsilon}{\mu}$. Similarly, $\|z^* - w^*\| \leq \frac{\sigma \epsilon}{\mu}$. Therefore, we have

$$
\begin{aligned}
\|\nabla_x \psi_\sigma(x^*, y^*, z^*) - \nabla \psi_\sigma(x^*)\| &\leq \frac{\|\nabla_x h_\sigma(x^*, y^*) - \nabla_x h_\sigma(x, w_\sigma^*)\| + \|\nabla_x g(x^*, z^*) - \nabla_x g(x^*, w^*)\|}{\sigma} \\
&\leq \frac{l_{g,1}}{\sigma} \left( \|y^* - w_\sigma^*\| + \|z^* - w^*\| \right) \leq \frac{2l_{g,1}\epsilon}{\mu}.
\end{aligned}
$$

To bound the projection error, note that

$$
\frac{1}{\rho}\|\Pi_{\mathcal{X}}\{x^* - \rho \nabla \psi_\sigma(x^*)\} - \Pi_{\mathcal{X}}\{x^* - \rho \nabla_x \psi_\sigma(x^*, y^*, z^*)\}\| \leq \|\nabla_x \psi_\sigma(x^*, y^*, z^*) - \nabla \psi_\sigma(x^*)\| \leq \frac{2l_{g,1}}{\mu}\epsilon,
$$

by non-expansiveness of projection operators. Thus,

$$
\frac{1}{\rho}\|x^* - \Pi_{\mathcal{X}}\{x^* - \rho \nabla \psi_\sigma(x^*)\}\| \leq (1 + 2l_{g,1}/\mu) \cdot \epsilon,
$$

concluding that $x^*$ is an $O(\epsilon)$-stationary point of $\nabla \psi_\sigma(x)$.

The second part comes from the mean-value theorem: by the assumption, there exists $\sigma' \in [0, \sigma]$ such that

$$
\nabla \psi_\sigma(x) = \frac{\partial_x l(x, \sigma) - \partial_x l(x, 0)}{\sigma} = \frac{\partial^2}{\partial \sigma \partial x} l(x, \sigma') = \frac{\partial^2}{\partial x \partial \sigma} l(x, \sigma').
$$

We also assumed $L_\sigma$-Lipschitz continuity of second cross-partial derivatives, and thus

$$
\left\| \frac{\partial^2}{\partial x \partial \sigma} l(x, \sigma') - \frac{\partial^2}{\partial x \partial \sigma} l(x, 0) \right\| \leq L_\sigma \sigma' \leq L_\sigma \sigma,
$$

which implies $\|\nabla \psi_\sigma(x) - \nabla \psi(x)\| \leq L_\sigma \sigma$. Using a similar projection non-expansiveness argument, we can show that

$$
\frac{1}{\rho}\|x - \Pi_{\mathcal{X}}\{x - \rho \nabla \psi(x)\}\| \leq (1 + 2l_{g,1}/\mu) \cdot \epsilon + L_\sigma \sigma,
$$

concluding the proof. $\square$

### H.3   PROOF OF LEMMA D.3

*Proof.*   Note that

$$\|\mathbf{prox}_{\rho g(x_1,\cdot)}(y_1) - \mathbf{prox}_{\rho g(x_2,\cdot)}(y_2)\| \leq \|\mathbf{prox}_{\rho g(x_1,\cdot)}(y_1) - \mathbf{prox}_{\rho g(x_2,\cdot)}(y_1)\|$$
$$+ \|\mathbf{prox}_{\rho g(x_2,\cdot)}(y_1) - \mathbf{prox}_{\rho g(x_2,\cdot)}(y_2)\|.$$

Due to non-expansiveness of proximal operators, the second term is less than $\|y_1 - y_2\|$. For bounding the first term, define

$$w_1^* = \mathbf{prox}_{\rho g(x_1,\cdot)}(y_1) = \arg\min_{y\in\mathcal{Y}} g(x_1, w) + \frac{1}{2\rho}\|w - y_1\|^2,$$

$$w_2^* = \mathbf{prox}_{\rho g(x_2,\cdot)}(y_1) = \arg\min_{y\in\mathcal{Y}} g(x_2, w) + \frac{1}{2\rho}\|w - y_1\|^2.$$

Due to the optimality condition, for any $\beta = \rho/4$, we can check that

$$w_1^* = \Pi_{\mathcal{Y}}\left\{w_1^* - \beta(\nabla_y g(x_1, w_1^*) + \rho^{-1}(w_1^* - y_1))\right\}.$$

On the other hand, define $w' = \Pi_{\mathcal{Y}}\left\{w_1^* - \beta(\nabla_y g(x_2, w_1^*) + \rho^{-1}(w_1^* - y_1))\right\}$. This is one projected gradient-descent step (PGD) for finding $w_2^*$ started from $w_1^*$. By the linear convergence of PGD for strongly convex functions over convex domain Bubeck et al. (2015), since the inner function is $1/(3\rho)$-strongly convex and $1/\rho$-smooth by choosing proper $\rho$, we have

$$\|w' - w_2^*\| \leq \frac{9}{10}\|w_1^* - w_2^*\|.$$

Thus,

$$\|w_1^* - w_2^*\|$$
$$\leq \|w_1^* - w'\| + \|w' - w_2^*\|$$
$$\leq \|\Pi_{\mathcal{Y}}\left\{w_1^* - \beta(\nabla_y g(x_1, w_1^*) + \rho^{-1}(w_1^* - y_1))\right\} - \Pi_{\mathcal{Y}}\left\{w_1^* - \beta(\nabla_y g(x_2, w_1^*) + \rho^{-1}(w_1^* - y_1))\right\}\|$$
$$+ \frac{9}{10}\|w_1^* - w_2^*\|$$
$$\leq \beta\|\nabla_y g(x_1, w_1^*) - \nabla_y g(x_2, w_1^*)\| + \frac{9}{10}\|w_1^* - w_2^*\|$$
$$= \beta O(l_{g,1})\|x_1 - x_2\| + \frac{9}{10}\|w_1^* - w_2^*\|.$$

where in the third inequality we used non-expansive property of projection onto convex sets. Arranging the term and plug $\beta = \rho/4$, we get

$$\|w_1^* - w_2^*\| \leq O\left(\rho l_{g,1}\right)\|x_1 - x_2\|.$$

$\square$

### H.4   PROOF OF LEMMA D.4

*Proof.*   Again, let $w_1^* = \mathbf{prox}_{\rho h_{\sigma_1}(x,\cdot)}(y)$ and $w_2^* = \mathbf{prox}_{\rho h_{\sigma_2}(x,\cdot)}(y)$. Using similar arguments to the proof in Appendix H.3, we have

$$\|w_1^* - w_2^*\| \leq \beta\|\nabla_y h_{\sigma_1}(x, w_1^*) - \nabla_y h_{\sigma_2}(x, w_1^*)\| + \frac{9}{10}\|w_1^* - w_2^*\|$$

$$\leq \beta\|\sigma_1 \nabla_y f(x, w_1^*) - \sigma_2 \nabla_y f(x, w_1^*)\| + \frac{9}{10}\|w_1^* - w_2^*\|.$$

where $\beta = \rho/4$. Arranging terms and using $\|\nabla_y f(x, w_1^*)\| \leq l_{f,0}$, we get the lemma.   $\square$

### H.5   PROOF OF LEMMA D.5

*Proof.*   By Lemma D.3, we know that the solution of $\min_{w\in\mathcal{Y}} h_\sigma(x, w) + \frac{1}{2\rho}\|w - y\|^2$ is $O(\rho l_{g,1})$ Lipschitz-continuous in $x$ and 1 Lipschitz-continuous in $y$. Therefore by Lemma D.2, since the inner minimization problem is $\rho^{-1}$-smooth, $h_{\sigma,\rho}^*(x, y)$ is $2\rho^{-1}$-smooth.   $\square$

