# OpenReview forum: "On Penalty Methods for Nonconvex Bilevel Optimization and First-Order Stochastic Approximation"
_ICLR.cc/2024/Conference — ICLR 2024 spotlight_

### Official Review · Reviewer_vCGE · 2023-10-19

**Soundness:** 3 good
**Presentation:** 2 fair
**Contribution:** 3 good
**Rating:** 6
**Confidence:** 3

**Summary:**

This paper studies a class of bilevel problems with a class of nonconvex lower-level problems. It establishes a strong connection between the landscape of the penalty reformulation and the original problem. This gives an interesting explanation about why the penalty method works even with nonconvex lower-level functions. Based on this, a single-loop stochastic first-order algorithm is proposed.

**Strengths:**

1. I found Thm. 3.2 very interesting. It converts the differentiability problem of $\nabla \psi(x)$ into the problem of whether the partial derivative of $l(x,\sigma)$ is commutable. It solves the open problem that was not solved by Chen et al. (2023b), Shen & Chen (2023), and  Arbel & Mairal (2022) in an elegant way.
2. The algorithm works for the stochastic case, which is not analyzed in Chen et al. (2023b), Shen & Chen (2023), and  Arbel & Mairal (2022).

**Weaknesses:**

1. (minor) The readability of the proof is not very good, with some typos. An incomplete list：
*  Page 24: "perturbations in $x,y,\sigma$ can change gradients of Lagrangian only by order $O(\delta)$ ". It seems that from the context $x,y,\sigma$ should be $\theta,w$.
*  Page 24: After "Next, we check that...". It seems that $y^*$ in the second line should be $\theta^*$.
*  Page 26: After "We list a few properties of...". It seems $\theta_0$ in the second line should be $y_0$.
*  Page 26: After "where we use ..." and before "continuity of ". It seems that $z_t^*$ should be $y_t^*$.

2. (minor) Assumption 6, 7(1), 8 may not be easily verified in practice. But given their natural appearance in the proof, i think this point does not detract from the main contribution of this paper.

**Questions:**

1. Prop 3.1. How to get Prop. 3.1 from Thm. E.3? How does the additional ${\rm Span}(\,...\,, \nabla_y f(x,y^*))$ appear?
2. Thm. 3.2. Can we derive the explicit form of $\nabla \psi(x)$? Or we can only define it implicitly as the limit of $\nabla \psi_{\sigma}(x)$ when  ${\sigma \rightarrow 0^+}$?
3.  Appendix E.6. Proof of Thm. 3.6. After "Then, since we have Assumption 1, we get", why we have $2 (g(x,y_{\sigma}^*) - g(x,z_p^*)) \ge \mu \Vert y_{\sigma}^* - z_p^* \Vert^2$. It seems like the QG condition in Karimi et al. (2016), but it seems like this reference only shows that it can be implied by the PL condition in the unconstrained case. But how we can get this in the constrained case?
4.  Proof of Thm. E.3 in Appendix. Why we always have $\Omega(\delta) v = \nabla^2_{\theta w} f(w,\theta^\ast) {\rm d}w$? What if $\nabla_{\theta w}^2 f(w,\theta^\ast)$ is zero, then the RHS is also zero?
5.  Appendix B.2. Can the authors explain more about the necessity of using the Moreau envelope? What is the benefit of using the Moreau smoothing? What will happen if it is not used? Sorry, I don't fully understand the technical challenge 1. Why it fails if one simply estimate the value of $(\lambda^*,\nu^*,y^*)$ for a given $x$ and plug into the expression of $\nabla \psi_{\sigma}(x)$ in Thm. 3.2?

**Details Of Ethics Concerns:**

n/a.

---

> ### Author Response · Authors · 2023-11-20
> **Author Response**
>
> We thank the reviewer for the positive feedback and comments. We answer the questions raised by the reviewer below.
>
> *Q1. Prop 3.1. How to get Prop. 3.1 from Thm. E.3?:* We can plug the Theorem E.3 with $w=(x,\sigma)$ and $\theta = y$, and $func(w,\theta) := h_{\sigma}(x,y) = \sigma f(x,y) + g(x,y)$. The image of $\nabla_{w\theta}^2 func$ is the span of columns, which is $span(\nabla_{xy}^2 h_{\sigma}(x,y), \nabla_y f(x,y))$.
>
> *Q2. Can we derive the explicit form of $\nabla \psi$:* Our result is also the explicit form of $\nabla \psi$, but yes, you can also directly derive the formula $\nabla \psi$ following the same steps in our proof in Appendix E.5. In that case, we would need the Lipschitz continuity of solution sets in $x$ at $T(x,0^+)$.
>
>
> *Q3. Appendix E.6. Proof of Thm. 3.6:* Here we used Assumption 1, (proximal-EB): by definition of the proximal operation, $$g(x,y) \ge g(x,prox(y)) + \frac{1}{\rho} \|y - prox(y)\|^2 \ge g(x,z) + \frac{1}{\rho} \|y - prox(y)\|^2.$$
> Then use Assumption 1 to lower bound $\frac{1}{\rho} \|y - prox(y)\|^2$. We will add this step in our proof.
>
> *Q4. Proof of Thm. E.3 in Appendix:* We took $dw$ in the range space of $\nabla^2 f$, so that $\nabla^2 f dw$ is always nonzero. If $\nabla^2 f = 0$, then the span of the image is $\{\emptyset\}$, so the proposition is trivial. We will add a comment that we do not consider the case when $\nabla^2 f = 0$.
>
> *Q5. Appendix B.2. Why Moreau envelope?* Thank you for the question, this is a great point. Yes, we can always solve the inner problems with sufficiently good accuracy first, before updating the outer variable. Though with stochastic noises and constraints, it is not entirely obvious to obtain (near) optimal multipliers along with solutions -- essentially, this is another constrained optimization in stochastic settings, which is a complicated subproblem by itself. Using Moreau Envelop is a convenient technique to side-step such issues, and further it becomes easy to design the algorithm in a single-loop. Nevertheless, as the reviewer questioned, we also think that it is an interesting question whether we can directly work with multipliers without Projection steps and/or Moreau Envelop smoothing.
>
>
> Thank you for pointing out other typos in the proof. We will fix them in our revision.

---

### Official Review · Reviewer_dzvn · 2023-10-31

**Soundness:** 3 good
**Presentation:** 3 good
**Contribution:** 3 good
**Rating:** 6
**Confidence:** 3

**Summary:**

The paper analyzes bi-level stochastic optimization problems where the objectives on both levels are smooth but not necessarily convex.
The paper proves a connection between solutions of a penalized reformulation of the problem to solutions of the original objective. Based on this, the authors provide a first-order algorithm with a non-asymptotic guarantee.

**Strengths:**

Though I am not an expert on bilevel programming, the contribution seems solid and of significance.
The authors clearly situate the results with respect to previous work on the topic.
The paper is fairly well-written, and pretty easy to follow with most relevant details reminded and cited adequately.

**Weaknesses:**

The main weakness in my opinion is the sheer amount of assumptions made throughout the paper. While some are standard and minor, some seem to highly hint towards the desired results - especially Assumption 5-8. As a non-expert on the topic, I find it hard to assess whether the assumptions are indeed (approximately) minimal for the results to hold, and how standard they are in this field. They are somewhat unmotivated in my opinion, and possibly ad-hoc.
The authors should attempt at explaining why these assumptions are required, and moreover that the results do not simply follow from them in a relatively-trivial manner.

**Questions:**

- Assumption 1: Was this assumption considered in previous work?
Moreover, the quantifiers are unclear: "satisfies (4) for all $y$ that satisfies... with some positive $\mu,\delta,\sigma_0$". Do you mean that  there exist $\mu,\delta,\sigma_0$ so that $y$ satisfies... *implies* (4)? This should be properly revised.
- Assumption 3 vs. 4: It is assumed that $f,g\to\infty$ as $\|y\|\to\infty$, but then that $\|y\|\leq O(1)$. Why aren't these in contradiction to one another?
- Assumptions 5-8: As previously mentioned, can the authors elaborate on these assumptions? They are not well-enough motivated. Statements such as "Assumption 6 helps ensure that the active set does not change when ... is perturbed slightly" seems to suggest the desired consequence is almost assumed in the first place. Also, in particular, is Assumption 7.1 standard? It seems strong, and I am not aware of such an assumption in previous works (I may be wrong, though would like the authors to clarify).
- High-level remark: I am not sure whether the authors decision do defer the entire algorithmic aspect of the paper to the appendix is preferable. Of course I acknowledge the strict page quota, and this may be inevitable in a lengthy work such as this, but it seems unsatisfactory that the only result mentioned in the main text is Theorem 3.6, which remains rather un-motivated without the algorithmic contribution. This becomes most clear in the conclusion, when the authors mention that their algorithm is simple, general, useful etc., though the reader did not encounter it at all in the main text. For the author's consideration.

minor comments:
- 1st paragraph after (P): "continuously differentiable *and* smooth" - are these synonymous?
- "since this issue is fundamental" - this phrasing is unclear, I suggest explaining what this means (possibly informally).
- Throughout the paper only \citet is used, while \citep is more adequate whenever the work is not part of the sentence.
- Theorem 1.1 (Informal): the phrase "at least one sufficiently regular solution path exists" is unclear at this point. What is even a solution path (not previously mentioned or defined)?
- mean squared smoothness condition mentioned without definition or ref. For example, the authors can add "(as formally defined in Section ...)".
- typo: to appendix = to the appendix.
- Assumption 2 involves a norm, which only later on (in the notation paragraph) is explained to be the operator norm. This should be clarified beforehand.
- The normal cone is mentioned without definition, I suggest adding a brief reminder (even in a footnote, for example).

---

> ### Author Response · Authors · 2023-11-20
> **Author Response**
>
> We appreciate the positive evaluation of our work, and thoughtful comments from the reviewer. For the major concerns on our Assumptions and organization, please see our general responses. Below, we answer to other major questions raised by the reviewer.
>
>
> *Assumption 1:* The most well-studied assumption is the strong-convexity of $g$, which implies Assumption 1. EB is essentially PL in the constrained settings, and considered only in a few recent work (Shen and Chen, 2023). Thank you for the suggestion, we will revise the part accordingly.
>
> *Assumption 3-4:* Assumption 3 is about the functions, and Assumption 4 is about the domain of $y$, and they do not contradict each other.
>
> *Theorem 1.1 (Informal):* We will clarify that the detailed meaning of our phrases is explained over Section 3, and the formal version is presented in Theorem 3.6.
>
> *continuously differentiable and smooth:* A function can be continuously differentiable but not smooth; an example is $f(x) = sign(x) \cdot x^2$.
>
> We thank for other comments on clarification and typos. We will revise them accordingly.

---

> > ### Comment · Reviewer_dzvn · 2023-11-23
> > **Rebuttal response**
> >
> > I thank the authors for the detailed response. I maintain my score.

---

### Official Review · Reviewer_ocU5 · 2023-11-05

**Soundness:** 3 good
**Presentation:** 3 good
**Contribution:** 3 good
**Rating:** 6
**Confidence:** 3

**Summary:**

This work studies a penalty method for constrained bilevel optimization, where the objective functions are possibly nonconvex in both levels with closed convex constraint sets. While the penalty technique is not new, its landscape relationship with the original problem was not known. This paper, for the first time, characterizes the conditions under which the value and derivatives of the penalty function is $O(\sigma)$ close to those of the hyperobjective function of the original problem, making the penalty method more useful theoretically. This paper then suggests efficient (stochastic) first-order methods for finding an $\epsilon$-stationary solution, by optimizing the penalty formulation with $\sigma = O(\epsilon)$. In particular, under the small-error proximal error-bound (EB) condition, that is closely related to the PL condition, the proposed algorithm finds an $\epsilon$-stationary solution in total $O(\epsilon^{-7})$ oracle accesses, which is better than the existing method for a relatively easier PL optimization.

**Strengths:**

1. This paper provides a theoretical justification of using the penalty method for bilevel optimization, which is not trivial as seen from Examples 1 and 2. In particular, this paper found minimal(?) conditions, Assumptions 1-4 and 7-8, to show that the value and derivatives of the penalty function are $O(\sigma)$ close to those of the hyperobjective function of the original problem, which is certainly not trivial.

2. Under the proximal EB condition, this paper presents a (stochastic) first-order method for efficiently finding an $O(\epsilon)$-stationary solution of the penalty (saddle-point) problem, which is also an $O(\epsilon)$-stationary solution of the original problem by the previous argument. This result is interesting as its rate improves upon that of the existing method for a similar PL saddle-point problem.

3. This paper provides the first explicit formula of the gradient of the hyperobjective function of the bilevel optimization with multiple solutions for the lower-level problem (under Assumptions 5-6).

**Weaknesses:**

1. Although the authors claim that they use nearly minimal conditions, they are not few, not simple, and are not guaranteed to satisfy in practice. (Providing an example satisfying those minimal conditions could be useful.)

2. This paper becomes difficult to read towards the end, probably due to the page limit.

**Questions:**

1. Title: What do you mean by "first-order stochastic approximation"?
2. Page 2: The third paragraph states that there are results under the uniform PL condition, which seemed to imply that there is a method for the PL condition, but the next paragraph claims that there is no algorithm finding the stationary point under the PL condition. Could you clarify this?
3. Page 2: In the fourth paragraph, how about making the term "landscape" more specific? It was not clear to me here, although it becomes clearer later.
4. Page 3: How about adding "small-error" in front of the "proximal EB"?
5. Page 3: It is implied that Assumption 1 guarantees the local Lipschitz continuity of solution sets (Assumption 5), but it seems that it is not explicitly stated (with proof) anywhere.
6. Page 3: PL and proximal EB conditions are almost identical, and you only assume the proximal EB within the neighborhood of solutions. Then, how were you able to get an improved rate over existing rate for the PL condition? Most of the explanations of the algorithm are deferred to the appendix, and it was not clear for what reason that the proposed algorithm is more efficient even with weaker condition. Could you comment on this?
7. Page 7: How does Assumption 6 help the active set to not change much?
8. Page 7: I was not able to follow Proposition 3.1 and its corresponding explanation. Could you help me better understand the context?
9. Page 9: Can Theorem 3.6 be proven with Assumption 5 rather than Assumption 1?

---

> ### Author Response · Authors · 2023-11-20
> **Author Response**
>
> We thank the reviewer for the positive feedback and comments. For the major concern on our Assumptions and organization, please see our general responses. We answer the questions raised by the reviewer below.
>
> *"first-order stochastic approximation" in title:* We design a stochastic algorithm for solving Bilevel optimization with only first-order derivatives of objective functions, hence first order stochastic approximation.
>
> *The third paragraph states that there are results under the uniform PL condition ... Could you clarify this?:* Yes, there are previous works that study the PL condition for the Bilevel optimization, but all existing works use some relaxed notion of ($\epsilon$-approximate) stationarity rather than the most straight-forward stationarity notion of $\|\nabla \psi(x)\| \le \epsilon$. For example, we can alternatively use the $\epsilon$-stationarity of penalty formulation, or a $\epsilon$-KKT point for $P_{con}$ as we discussed in Remark 3.7. Our contribution is to establish strong bond between such alternative stationarity measures and the original stationarity measure.
>
> *It is implied that Assumption 1 guarantees ... not explicitly stated (with proof) anywhere:* This implication is stated in our Lemma 3.5 and Lemma D.2.
>
> *PL and proximal EB conditions are almost identical ... the proposed algorithm is more efficient even with weaker condition. Could you comment on this?:* Thank you for your question. To our best knowledge, we are not aware the specific rates discussed in stochastic settings with PL condition. In the deterministic setting, (Chen et al., 2023) gave a $O(\epsilon^{-3})$ guarantee with PL condition for the penalty method, so it matches to what we provide. Our result should be viewed as an approach generalizing to a broader class of instances, rather than improving the compleixty.
>
>
> *Proposition 3.1 Clarification:* Sure, it migth be easier if we think about Proposition 3.1 in this way: in constrained optimization (with viewing $x$ as outer parameters), the solution sensitivity (the change of a solution responding to the change in $x$) is given by $-(\nabla_{yy}^2 \mathcal{L})^{-1} (\nabla_{xy}^2 \mathcal{L})$. So, with slight abuse, assuming $0/0=0$, for the solution sensivity to be bounded by a finite value, the image of $(\nabla_{xy}^2 \mathcal{L})$ must be in the range of $(\nabla_{yy}^2 \mathcal{L})$. Proposition 3.1 is the formal version of this argument.
>
>
>
> *Can Theorem 3.6 be proven with Assumption 5 rather than Assumption 1?:* Good point. The answer is yes, you can show the same result (with replacing $\mu$ with something else), as long as we assume that Assumption 5 holds globally for all $x$ and $\sigma$ (and yes, Assumption 1 is slightly stronger than that).
>
> We thank for other comments on clarification and suggestions. We will revise the manuscript accordingly.

---

### Official Review · Reviewer_fS1u · 2023-11-06

**Soundness:** 3 good
**Presentation:** 3 good
**Contribution:** 3 good
**Rating:** 5
**Confidence:** 3

**Summary:**

The paper discusses the extension of first order methods from bilevel optimisation problems with convex objective to bilevel optimisation problems with non convex objectives

**Strengths:**

The abstract is neat and the paper is dense and well written. The problem is clearly stated and there is a need for a solution which the authors provide.

**Weaknesses:**

I would be more straightforward and from the very beginning of the paper emphasize the narrative. My main concern has to do with the fact that the paper might be too dense for a conference. You either have to make it simpler/lighter or you should submit it to a journal. There does not seem to be enough space for you to give the complete details, or if you prefer, we miss some intuitive explanation (i.e there are too many details and the main thread is not always clear). Before providing formal statements, you should be able to explain in simple terms what the main ideas are (e.g. for Proposition 3.1.).  At this point, it appears as if you had a very detailed result and quickly tried to wrap it up as a conference paper.  You took some of the details and put some others in the appendices, but the general organization (the decision of retaining some results instead of others, such as Proposition 3.1) is unclear.. I would be in favor of reducing the amount of mathematical details (see my detailed comments below) and increasing intuitive explanations (at least in the main body of the paper).  You could do a much better job at focusing on your main results. An example of this is section 3 where the reader has to wait the very end of the section to discover the connection to the earlier sections (see my comments below).

**Questions:**

A couple of typos:

- Section 3.3. first line: “via finite differentation” —> “via finite differentiation”
- Section 3.3. “we can apply mean-value theorem” —> “we can apply the mean value theorem”
- If you check, on both page 3 (below Assumption 1) and page 8, you use the sentence “the crux of assumption 1 is the guaranteed (Lipschitz) continuity of solution sets”
- Page 8: “With a more standard assumption on the uniqueness of the lower-level solution and the invertability ” —> “invertibility” ?



Page 1/2
- You first introduce formuation (P) then you introduce formulation P_con which to me is exactly equivalent to P, isn’t it ? If it is, I would not relabel it but simply emphasize the equivalence between P and this formulation. There are already many formulations. If some of those are exactly equivalent (like I suspect it is the case for P and Pcon, I would not introduce additional notations)
-  Finally you introduce a relaxation of the constraint in P_pen. intuitively, we understand that for sigma small enough, this relaxation should recover P_con and this is what you formalize by introducing appropriate conditions. So far, this is clear. What I find less clear is the sentence “Although finding the solution of (Ppen) is a more familiar task than solving (P) ”. For small \sigma, from Theorem 1.1., there must be some conservation of difficulty (at least for some \sigma). It would be helpful to have some more intuition on that (one or two sentences). I.e. it seems miraculous to end up with a formulation that suddenly becomes so much easier so solve.

Page 2
- In the formulation P_{pen}, I don’t really understand how you can know the value of min_z  g(x, z) while you don’t have any explicit solution for min_y g(x, y). This should be commented

Page 3
- The result of Theorem 1 is clearly interesting but it could be expected that a smaller value of sigma would enforce more weight on the penalty and hence ultimately recover the solution to (1)
- You should define the notion of epsilon-stationnary point. In particular What is the meaning of epsilon in regard to sigma. When you introduce the notion of epsilon-solution you seem to replace the sigma in psi_sigma by the epsillon, but then the two appear simultaneously when discussing the oracle complexity
- You use the terms “complexity” and “oracle complexity” before clearly defining them. Are you talking about the number of iterations/steps? If yes, this should appear somewhere. The notion of “oracle complexity” is also used in a number of different settings. What are the oracles you are referring to ? What function classes are you considering. This is not self contained (at least on page 3). I can see you clarify this in the statement of Theorem 1.2. but it should appear earlier. In fact I’m wondering if you should not put Theorem 1.2 before Theorem 1.1. Also, is your oracle only given by the noisy/stochastic gradient or do you also assume access to the value of the function? The information that appear in section 3.4. should appear way earlier.
- What do you mean by the sentence “Assumption 1 holds only in a neighborhood of the lower-level solution ” ? do you mean that Theorem 1.1. only requires Assumption 1 to hold in a neighborhood of y^* ? Then you should clarify this in the statement of Theorem 1.

Page 4
- You mention the mean-squared smoothness condition before defining it. Either include it in the main part of the paper or remove the second part of the Theorem. The sentence “fully single-loop manner” is also unclear so I would be in favor of removing the second part of the Theorem (at least from the main body of the paper)

Page 5
- Why not use V for the value function and Y for the solution set ?
- In section 3, we don’t really understand what you are doing
- I think you can spare some space by removing the definition of Lipschitz continuity from the main body of the paper (this is a relatively well known concept and it would free some space for some other (perhaps more crucial) explanations — see my other comments)

Page 6
- I would be in favor of a couple of additional explanatory sentences in Examples 1 and 2. E.g. just clarify the definition of S(x), i.e. S = {-1,1} except at 0 where it is the whole [-1,1] interval, or recall the definitions of S and psi
- The whole introduction in section 3 is confusing. It is not really clear what you are trying to do do until the last sentences. I.e. we understand you want to show examples in which \nabla \psi(x) \neq \lim_{\sigma\rightarrow 0} \nabla \psi_{\sigma}(x). But it is not completely clear why, especially given that you seem to already have introduced in assumption 1, a condition that ensures that \nabla \psi_{\sigma} can be made arbitrarily close to \nabla \psi.
- Under example 2, when you mention the continuity of the solution set. Do you mean continuity with respect to x ?
- Before introducing examples 1 and 2, I would add a sentence insisting on the fact that in both examples, the non existence of $\nabla \psi$ comes from the discontinuity in $\psi$. Also I known you don’t have much space but it would help to have a plot of the solution sets for both examples (or perhaps briefly mention the solution sets for both cases ? This would help clarify the fact that the set is )

Page 7
- Although technically correct, the term “kernel” in the setence “the kernel space of the Hessian …” before Theorem 3.2. can be confusing especially in ML related papers. I would use nullspace instead of kernel space.
- The central part of section 3, i.e. the part that connects this section to Theorem 1.1. is Theorem 3.2. This theorem should appear way earlier or at least you should clearly explain that you derive the conditions appearing in Theorem 1.1. I would rewrite section 3 as follows. Start by clearly indicating that you will derive conditions for the second relation in Theorem 1.1. to hold (i.e. the fact that \nabla \psi_{\sigma}(x) is a O(\sigma) approximation of \nabla \psi)
- I would rewrite section 3.1. by giving more intuition on how you derive assumption 6. Just a couple sentences of explanation. (I think readability of the paper would benefit from for example, removing the definition of Lipschitz continuity and replacing it with a more elaborate explanation of the derivation of assumption 6)
- I would remove Proposition 3.1 or I would make it simpler. The proposition is not clear and neither is its connection to the rest of the paper or its explanation (i.e. Geometrically speaking, perturbations in … ).

Page 8
- I would simplify section 3.3.

---

> ### Author Response · Authors · 2023-11-20
> **Author Response**
>
> We appreciate the comprehensive feedback on our manuscript. For your major concern in the weakness, please refer to our common responses to all reviewers. Below, we address some important comments/questions raised by the reviewer.
>
> *Page 1/2 - Clarification on "a familiar task":* The reason that we described the penalty formulation as a slightly easier task is that it is essentially a single-level optimization (although technically, we are still dealing with the "min-max" structure due to $\min_z g(x,z)$. So, as you pointed out, this is still non-trivial). We have edited this paragraph to make the point clear in the revision.
>
>
> *Page 3 - Theorem 1.1:* You are right, while Theorem 1.1 makes an intuitive sense, our main contribution is the explicit quantification of the intuition: we exactly state the dependence of the bias induced by using the penalty formula.
>
>
> *Page 5 - What Section 3 Intro doing:* We are trying to explain why and when the penalty formulation is not a good approximation of the original problem. Here we also introduce some non-standard notations and concepts, as our approach starts from converting the landscape approximation question to the question of whether differentiation with respect to $x$ and $\sigma$ is commutable.
>
>
> *Page 5 - Lipschitz continuity:* While  Lipschitz continuity is a widely known concept for real-valued functions, we do not think it is the case for set-valued mappings.
>
>
> *Page 6/7 - Theorem 3.1 (new numeration, previously Theorem 3.4):* We thank the reviewer for the comments. We also wanted to state Theorem 3.1 as early as possible. However, we believe that providing appropriate explanations for the assumptions is equally important. That's why we included examples and assumptions before stating the theorem. We appreciate the reviewer's suggestion on the organization. Following the suggestion, we added the following paragraph to the introduction of Section 3 before presenting the examples.
>
> > In what follows, we derive two assumptions: one concerning solution-set continuity (Assumption 5) and another addressing the regularities of lower-level constraints (Assumption 6). Under these assumptions, we establish the main theorem of this section, Theorem 3.2. This, in turn, leads to the derivation of the second inequality in our landscape analysis result, as presented in Theorem 1.1 and as shown in Theorem 3.6.
>
> *Page 6/7 - Section 3.1:* To clarify the derivation of the assumptions in Section 3.1, we moved two examples from the beginning of Section 3 to Section 3.1. Furthermore, we added the following sentence at the beginning of Section 3.
>
> > The following two examples illustrate the obstacles encountered when claiming $\lim_{\sigma \rightarrow 0} \nabla \psi_{\sigma}(x) \rightarrow \nabla \psi(x)$ or even when simply ensuring the existence of $\nabla \psi(x).$
>
> *Page 7 - Section 3.2:* We included Proposition 3.4 (new numeration, previously Proposition 3.1) in the main text since we believe that it is important to highlight the consequences of Assumptions 5 and 6. To clarify this point, we rearrange Section 3.2 as follows. We explain our main results (Theorem 3.1) and the remarks first. Then, we present Proposition 3.4. We believe that this gives an idea of the proof of Theorem 3.1.
>
> We greatly appreciate many other thoughtful comments and suggestions by the reviewer. We will reflect them as much as possible properly in our revision.

---

### Official Review · Reviewer_VYCb · 2023-11-08

**Soundness:** 4 excellent
**Presentation:** 3 good
**Contribution:** 4 excellent
**Rating:** 8
**Confidence:** 4

**Summary:**

This paper addresses bilevel optimization problems that can encompass constraints and nonconvex lower-level problems. It establishes an $O(\sigma)$-closeness relationship between the hyper-objective $\psi(x)$ and the penalized hyper-objective $\psi_{\sigma}(x)$ under the proximal-error bound condition when the errors are small. Using the penalty formulation, the authors develop fully first-order stochastic algorithms for finding a stationary point of $\psi_{\sigma}(x)$ with comprehensive non-asymptotic convergence guarantees.

**Strengths:**

S1. The work is well motivated. Finding a simple yet effective method for large-scale lower-level constrained bilevel optimization problems is both intriguing and significant.

S2. The paper is well written and easy to follow. The illustrations in Example 1 and Example 2 are helpful in understanding the main difficulties.

S3. The algorithm design is novel, suitable for a broader range of bilevel optimization problems that may involve constraints and nonconvex lower-level problems. Additionally, it provides a comprehensive non-asymptotic convergence analysis.

**Weaknesses:**

W1. I find the algorithm developed in this paper to be both interesting and important. It seems that the landscape analysis section is quite lengthy. Would it be possible to condense the content and incorporate the first-order algorithm into the main text?

W2. There is a closely related paper that addresses general constrained bilevel optimization problems while assuming the strong convexity of the lower-level objective.

[1]Siyuan Xu, Minghui Zhu. “Efficient Gradient Approximation Method for Constrained Bilevel Optimization.” AAAI 2023.

W3. No numerical experiments are provided.

**Questions:**

Q1. What is the relationship between the $\epsilon$-stationary point of $\psi_{\sigma}(x)$ and the $\epsilon$-KKT solution of the constrained single-level problem $P_{\mathrm{con}}$? Theorem E.5 establishes a one-directional relation. Is the converse of Theorem E.5 also valid? Note that in Theorem E.5, the $\epsilon$-KKT solution of $P_{\mathrm{con}}$ should be a pair $(x, y)$.

Q2. Could the requirement of boundedness for $\max_{x\in\mathcal{X},y\in\mathcal{Y}} |f(x,y)|=O(1)$ be relaxed or made less restrictive?

Minor Comments:

(1)On page 3, Theorem 1: It would be beneficial to clarify the meaning of $y^*(\sigma)$.

(2)On page 4, Assumption 3: Is it redundant to consider the coercivity of both $f(x,y)$ and $g(x,y)$ as $|y|\rightarrow\infty$, given that $\mathcal{Y}$ is bounded according to Assumption 4?

(3)On page 7, Definition 7: Does $\lambda_{\mathcal{I}}$ implicitely depend on $y$ in Equation (8)? What is the domain of $\mathcal{L}_{\mathcal{I}}$?

(4)On page 7, Equation (9): There is a period missing at the end of Equation (9).

(5)On page 9, Assumption 8: Does Assumption 8 implicitly assume that the active set does not change?

(6)On page 17, Equation (13): There is a period missing at the end of Equation (13).

(7)On page 18: check the definition of $g_{xy}^{k,m}$ and $g_{xz}^{k,m}$.

(8)On page 19, C.1, Definition of $\Delta_k^z$: “$\rho g(x, \cdot)$” should be “$\rho g(x_k, \cdot)$”.

(9)On page 22, Equation (21): “$D_v \ell^*(x)$” should be “$D_v \ell^*(w)$”.

(10)On page 27, Proof of Theorem 3.6, Line 8 from below: Should $\sigma C_f$ be replaced with $2\sigma C_f$?

(11)On page 27, Proof of Theorem 3.6, Line 4 from below: $\mu_g$ should be replaced with $\mu$.

(12)On page 30, Proof of Theorem E.5: It appears that in the last step of the proof, $g(x,y) - g^*(x)$ is bounded by $O(\sigma^2)$. Would this observation be useful for improving the convergence rate?

(13)On page 30, Equation (28): There is a period missing at the end of Equation (28).

---

> ### Author Response · Authors · 2023-11-20
> **Author Responses**
>
> We greatly appreciate the reviewer's acknowledgment that our work is well-motivated, well-written, and easy to follow. Furthermore, we are grateful for the insightful comments that have contributed to the enhancement of our manuscript.
>
> In addition, we would like to thank you for your interest in our algorithm. Regarding the comment W1, we also believe that both landscape analysis and algorithms are equally important, but we choose to leave only the landscape analysis in the main text. Please see the general comments to all reviewers for further details.
>
> *Q. What is the relationship... Is the converse of Theorem E.5 also valid?:* Thank you for this question! In general, the converse will not hold as there could be much more possibility of $\epsilon$-KKT solutions with different  multipliers $\lambda$. And thank you for catching the mistake, we will fix it in our revision.
>
>
>
> **Q. Minor Comments**
>
> *(3):* We appreciate your comments regarding the domain in Definition 7. To clarify this point, we define the Lagrangian restricted to a set of constraints first in Definition 7. The domain of $\mathcal{L}_I$ is added here.
> > Definition 7. $\cdots$ We also define the Lagrangian $L_I (\cdot | x,\sigma): \mathbb{R}_+^{|\mathcal{I}|} \times \mathbb{R}^{m_2} \times \mathbb{R}^{d_y} \rightarrow \mathbb{R}$ restricted to a set of constraints $\mathcal{I} \subseteq [m_1]$ $\cdots$
>
> In addition, we added the following sentence at the end of Section 3.1. We will update the manuscript accordingly to clarify this point.
> > When the context is clear, we always let $\mathcal{I}$ in (8) be $\mathcal{I}(y)$ at a given $y$.
>
>
> *(5):* Assumption 8 is about the minimum singular value of the LICQ condition in Assumption 6 and fairly standard in constrained optimization. This is about whether there are linearly dependent active constraints (so there could be multiple $\lambda^*$) at the solution $y^*$, and we do not want that to avoid further complication.
>
>
> *(12):* Thank you for pointing this out. We corrected the proof of Theorem E.5 as follows:
> >\begin{align*}
>         g(x,y) - g(x,z) = g(x,y) - g^*(x) \le \sigma^2 l_{f,0} L_T + \frac{\sigma^2 L_T^2}{2\rho} = O(\sigma^2),
> \end{align*}
>
>
> We thank for other questions and comments suggested. We will revise our manuscript accordingly. Please also see the further explanation in the general comments to all reviewers.

---

> > ### Comment · Reviewer_VYCb · 2023-11-23
> >
> > Thanks for the rebuttal. I will keep my evaluation.

---

### Author Response · Authors · 2023-11-20
**General Responses to All Reviewers**

We would like to thank all reviewers for their thoughtful comments and remarks. Here, we would like to response to two important issues raised by multiple reviewers.


**Why we choose to leave only the landscape analysis in the main text:** We thank all reviewers for acknowledging the importance of our algorithmic contribution and results that we deferred to the Appendix. As we emphasized in our introduction, our contributions are in two-fold: (a) understanding the landscape of nonconvex Bilevel optimization through the penalty formulation, and (b) proposing a general-purpose algorithm that is guaranteed to converge under the (small-error) proximal-EB assumption. We believe that both aspects are equally important, and deserve equal amount of detailed presentations.

That being said, we prioritized the landscape result over our algorithm for the following reason: to our best knowledge, our landscape analysis is **the first to explicitly quantify the relation between penalty and original formulations**. Also, we derive **the explict formula for the hypergradient which was not known in the literature** (under what we claim nearly minimal regularity assumptions). Whereas for the algorithm, we are not the first to study penalty methods, and thus, our algorithmic contribution should be viewed as an improvement over and an extension of existing methods, rather than a new idea or approach. As the authors of the paper, we were more excited about the new understanding and insights as they would be more beneficial to the community. Furthermore, we believe it is very important to provide proper contexts and justification of our assumptions used in Section 3, to properly deliver the significance of our landscape result. The only way to do that was to focus solely on one result that we believed more significant in the limited space, and defer the other result to appendix.

Nevertheless, we absolutely understand the reviewers' concerns -- perhaps **we will consider putting more pointers to our algorithmic results in the main body of the paper whenever appropriate.** We will reflect this in our final revision, as this task might take more care and time.





**On Our Assumptions 5-8:** We understand the reviewers' concerns about our assumptions. We would like to reiterate the issues that can happen without them.

Assumption 5: The solution-set continuity is the most critical requirement for both landscape and local-search algorithm to be stabilized. In general, the class of nonconvex-nonconcave min-max problem, which is a subclass of bilevel with nonconvex lower-level, is known to be PPAD-hard (Daskalakis et al., 2021). The difficulty is also illustrated in Example 1. A question would be then "is the solution-set (Lipschitz) continuity the weakest possible assumption?" -- for many the conditions considered in literature (Strong-Conveixty, PL, Morse-Bott, ...) implies the (Lipschitz) continuity of the lower-level solutions (or critical point sets). Aside from computational hardness, for the landscape of the problem to be well defined (i.e., hyperobjective is continuous and smooth), we conjecture that some kind of solution-set continuity would be necessary.

Assumption 6: LICQ is relatively standard in nonlinear constrained optimization when performing the local sensitivity analysis. We also illustrated why the strict complementary is a necessary assumption in Example 2.

Assumption 7-8: Assumption 7 and 8 can be viewed as **a non-asymptotic version of Assumption 5 and 6, and necessary for connecting the penalty method and original landscape with a nonzero penalty parameter**. In our algorithm, we gradually reduce $\sigma$ and $\epsilon$ to zero, and Assumptions 7-8 are only required for the final accuracy $\sigma = O(\epsilon) > 0$. Practical examples of such functions could be those that are strongly-convex, PL, over-parameterized strongly-convex functions (f(Ax) with a fat matrix A). Some examples that easily satisfy Assumptions 6 and 8 would be those that is essentially unconstrained (e.g., we may set a large ball as a constraint to prevent unexpected blow up of parameters), or consists only of linear equality constraints (that was studied in some recent work, e.g., Xiao et al. 2023b). A general case would be hard to characterize as it depends on where exactly the solution happens.

We conjecture that these are minimally sufficient conditions, and we actively used the assumptions in our landscape analysis. However, we will refrain from saying "minimal conditions" as it is not clear yet (as reviewers pointed out) that the assumptions are really necessary. We will work on our revision accordingly.

---

### Meta-Review · Area_Chair_cMxa · 2023-12-06

**Metareview:**

This paper achieves a better complexity than existing work on a class of bilevel optimization where the lower problem is non-convex. This problem is challenging, so I believe any theoretical improvement is significant.

The strength is the reduced complexity. The weakness is the additional assumptions the authors make.

**Justification For Why Not Higher Score:**

The additional assumptions the authors made are a little strong compared to existing work.

**Justification For Why Not Lower Score:**

This paper achieves a better complexity than existing work on a class of bilevel optimization where the lower problem is non-convex. This problem is challenging, so I believe any theoretical improvement is significant.

---

### Decision · Program_Chairs · 2024-01-16

Accept (spotlight)